# Greedy Poisson Rejection Sampling

**Gergely Flamich**
Department of Engineering
University of Cambridge
`gf332@cam.ac.uk`

## Abstract

One-shot channel simulation is a fundamental data compression problem concerned with encoding a single sample from a target distribution $Q$ using a coding distribution $P$ using as few bits as possible on average. Algorithms that solve this problem find applications in neural data compression and differential privacy and can serve as a more efficient alternative to quantization-based methods. Sadly, existing solutions are too slow or have limited applicability, preventing widespread adoption. In this paper, we conclusively solve one-shot channel simulation for one-dimensional problems where the target-proposal density ratio is unimodal by describing an algorithm with optimal runtime. We achieve this by constructing a rejection sampling procedure equivalent to greedily searching over the points of a Poisson process. Hence, we call our algorithm greedy Poisson rejection sampling (GPRS) and analyze the correctness and time complexity of several of its variants. Finally, we empirically verify our theorems, demonstrating that GPRS significantly outperforms the current state-of-the-art method, A* coding. Our code is available at `https://github.com/gergely-flamich/greedy-poisson-rejection-sampling`.

## 1   Introduction

It is a common misconception that quantization is essential to lossy data compression; in fact, it is merely a way to discard information deterministically. In this paper, we consider the alternative, that is, to discard information stochastically using *one-shot channel simulation*. To illustrate the main idea, take lossy image compression as an example. Assume we have a generative model given by a joint distribution $P_{\mathbf{x},\mathbf{y}}$ over images $\mathbf{y}$ and latent variables $\mathbf{x}$, e.g. we might have trained a variational autoencoder (VAE; Kingma & Welling, 2014) on a dataset of images. To compress a new image $\mathbf{y}$, we encode a single sample from its posterior $\mathbf{x} \sim P_{\mathbf{x}|\mathbf{y}}$ as its stochastic lossy representation. The decoder can obtain a lossy reconstruction of $\mathbf{y}$ by decoding $\mathbf{x}$ and drawing a sample $\hat{\mathbf{y}} \sim P_{\mathbf{y}|\mathbf{x}}$ (though in practice, for a VAE we normally just take the mean predicted by the generative network).

Abstracting away from our example, in this paper we will be entirely focused on *channel simulation* for a pair of correlated random variables $\mathbf{x}, \mathbf{y} \sim P_{\mathbf{x},\mathbf{y}}$: given a source symbol $\mathbf{y} \sim P_{\mathbf{y}}$ we wish to encode **a single sample** $\mathbf{x} \sim P_{\mathbf{x}|\mathbf{y}}$. A simple way to achieve this is to encode $\mathbf{x}$ with entropy coding using the marginal $P_{\mathbf{x}}$, whose average coding cost is approximately the entropy $\mathbb{H}[\mathbf{x}]$. Surprisingly, however, we can do much better by using a *channel simulation protocol*, whose average coding cost is approximately the mutual information $I[\mathbf{x}; \mathbf{y}]$ (Li & El Gamal, 2018). This is remarkable, since not only $I[\mathbf{x}; \mathbf{y}] \leq \mathbb{H}[\mathbf{x}]$, but in many cases $I[\mathbf{x}; \mathbf{y}]$ might be finite even though $\mathbb{H}[\mathbf{x}]$ is infinite, such as when $\mathbf{x}$ is continuous. Sadly, most existing protocols place heavy restrictions on $P_{\mathbf{x},\mathbf{y}}$ or their runtime scales much worse than $\mathcal{O}(I[\mathbf{x}; \mathbf{y}])$, limiting their practical applicability (Agustsson & Theis, 2020).

In this paper, we propose a family of channel simulation protocols based on a new rejection sampling algorithm, which we can apply to simulate samples from a target distribution $Q$ using a proposal distribution $P$ over an arbitrary probability space. The inspiration for our construction comes from an exciting recent line of work which recasts random variate simulation as a search problem over a set

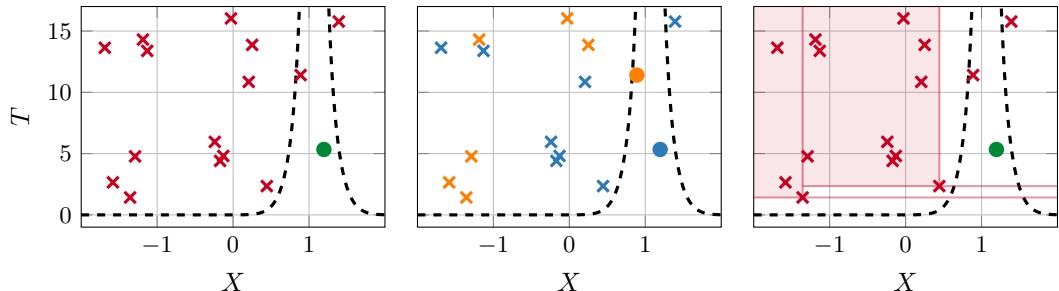

Figure 1: Illustration of three GPRS procedures for a Gaussian target $Q = \mathcal{N}(1, 0.25^2)$ and Gaussian proposal distribution $P = \mathcal{N}(0, 1)$, with the time axis truncated to the first 17 units. All three variants find the first arrival of the same $(1, P)$-Poisson process $\Pi$ under the graph of $\varphi = \sigma \circ r$ indicated by the **thick dashed black line** in each plot. Here, $r = dQ/dP$ is the target-proposal density ratio, and $\sigma$ is given by Equation (5). **Left:** Algorithm 3 sequentially searching through the points of $\Pi$. The green circle (●) shows the first point of $\Pi$ that falls under $\varphi$, and is accepted. All other points are rejected, as indicated by red crosses (✗). In practice, Algorithm 3 does not simulate points of $\Pi$ that arrive after the accepted arrival. **Middle:** Parallelized GPRS (Algorithm 4) searching through two independent $(1/2, P)$-Poisson processes $\Pi_1$ and $\Pi_2$ in parallel. Blue points are arrivals in $\Pi_1$ and orange points are arrivals in $\Pi_2$. Crosses (✗) indicate rejected, and circles (●) indicate accepted points by each thread. In the end, the algorithm accepts the earliest arrival across all processes, which in this case is marked by the blue circle (●). **Right:** Branch-and-bound GPRS (Algorithm 5), when $\varphi$ is unimodal. The shaded red areas are never searched or simulated by the algorithm since, given the first two rejections, we know points in those regions cannot fall under $\varphi$.

of randomly placed points, specifically a Poisson process (Maddison, 2016). The most well-known examples of sampling-as-search are the Gumbel-max trick and A* sampling (Maddison et al., 2014). Our algorithm, which we call *greedy Poisson rejection sampling* (GPRS), differs significantly from all previous approaches in terms of what it is searching for, which we can succinctly summarise as: "GPRS searches for the first arrival of a Poisson process $\Pi$ under the graph of an appropriately defined function $\varphi$". The first and simplest variant of GPRS is equivalent to an exhaustive search over all points of $\Pi$ in time order. Next, we show that the exhaustive search is embarrassingly parallelizable, leading to a parallelized variant of GPRS. Finally, when the underlying probability space has more structure, we develop branch-and-bound variants of GPRS that perform a binary search over the points of $\Pi$. See Figure 1 for an illustration of these three variants.

While GPRS is an interesting sampling algorithm on its own, we also show that each of its variants induces a new one-shot channel simulation protocol. That is, after we receive $\mathbf{y} \sim P_{\mathbf{y}}$, we can set $Q \leftarrow P_{\mathbf{x}|\mathbf{y}}$ and $P \leftarrow P_{\mathbf{x}}$ and use GPRS to encode a sample $\mathbf{x} \sim P_{\mathbf{x}|\mathbf{y}}$ at an average bitrate of a little more than the mutual information $I[\mathbf{x}; \mathbf{y}]$. In particular, on one-dimensional problems where the density ratio $dP_{\mathbf{x}|\mathbf{y}}/dP_{\mathbf{x}}$ is unimodal for all $\mathbf{y}$ (which is often the case in practice), branch-and-bound GPRS leads to a protocol with an average runtime of $\mathcal{O}(I[\mathbf{x}; \mathbf{y}])$, which is optimal. This is a considerable improvement over A* coding (Flamich et al., 2022), the current state-of-the-art method.

In summary, our contributions are as follows:

- We construct a new rejection sampling algorithm called *greedy Poisson rejection sampling*, which we can construe as a greedy search over the points of a Poisson process (Algorithm 3). We propose a parallelized (Algorithm 4) and a branch-and-bound variant (Algorithms 5 and 6) of GPRS. We analyze the correctness and runtime of these algorithms.

- We show that each variant of GPRS induces a one-shot channel simulation protocol for correlated random variables $\mathbf{x}, \mathbf{y} \sim P_{\mathbf{x}, \mathbf{y}}$, achieving the optimal average codelength of $I[\mathbf{x}; \mathbf{y}] + \log_2(I[\mathbf{x}; \mathbf{y}] + 1) + \mathcal{O}(1)$ bits.

- We prove that when $\mathbf{x}$ is a $\mathbb{R}$-valued random variable and the density ratio $dP_{\mathbf{x}|\mathbf{y}}/dP_{\mathbf{x}}$ is always unimodal, the channel simulation protocol based on the binary search variant of GPRS achieves $\mathcal{O}(I[\mathbf{x}; \mathbf{y}])$ runtime, which is optimal.

- We conduct toy experiments on one-dimensional problems and show that GPRS compares favourably against A* coding, the current state-of-the-art channel simulation protocol.

## 2 Background

The sampling algorithms we construct in this paper are search procedures on randomly placed points in space whose distribution is given by a Poisson process $\Pi$. Thus, in this section, we first review the necessary theoretical background for Poisson processes and how we can simulate them on a computer. Then, we formulate standard rejection sampling as a search procedure over the points of a Poisson process to serve as a prototype for our algorithm in the next section. Up to this point, our exposition loosely follows Sections 2, 3 and 5.1 of the excellent work of Maddison (2016), peppered with a few additional results that will be useful for analyzing our algorithm later. Finally, we describe the channel simulation problem, using rejection sampling as a rudimentary solution and describe its shortcomings. This motivates the development of greedy Poisson rejection sampling in Section 3.

### 2.1 Poisson Processes

A Poisson process $\Pi$ is a countable collection of random points in some mathematical space $\Omega$. In the main text, we will always assume that $\Pi$ is defined over $\Omega = \mathbb{R}^+ \times \mathbb{R}^d$ and that all objects involved are measure-theoretically well-behaved for simplicity. For this choice of $\Omega$, the positive reals represent *time*, and $\mathbb{R}^d$ represents *space*. However, most results generalize to settings when the spatial domain $\mathbb{R}^d$ is replaced with some more general space, and we give a general measure-theoretic construction in Appendix A, where the spatial domain is an arbitrary Polish space.

**Basic properties of $\Pi$:** For a set $A \subseteq \Omega$, let $\mathbf{N}(A) \stackrel{def}{=} |\Pi \cap A|$ denote the number of points of $\Pi$ falling in the set $A$, where $|\cdot|$ denotes the cardinality of a set. Then, $\Pi$ is characterized by the following two fundamental properties (Kingman, 1992). First, for two disjoint sets $A, B \subseteq \Omega, A \cap B = \emptyset$, the number of points of $\Pi$ that fall in either set are independent random variables: $\mathbf{N}(A) \perp \mathbf{N}(B)$. Second, $\mathbf{N}(A)$ is Poisson distributed with *mean measure* $\mu(A) \stackrel{def}{=} \mathbb{E}[\mathbf{N}(A)]$.

**Time-ordering the points of $\Pi$:** Since we assume that $\Omega = \mathbb{R}^+ \times \mathbb{R}^d$ has a product space structure, we may write the points of $\Pi$ as a pair of time-space coordinates: $\Pi = \{(T_n, X_n)\}_{n=1}^{\infty}$. Furthermore, we can order the points in $\Pi$ with respect to their time coordinates and *index* them accordingly, i.e. for $i < j$ we have $T_i < T_j$. Hence, we refer to $(T_n, X_n)$ as the *nth arrival* of the process.

As a slight abuse of notation, we define $\mathbf{N}(t) \stackrel{def}{=} \mathbf{N}([0, t) \times \mathbb{R}^d)$ and $\mu(t) \stackrel{def}{=} \mathbb{E}[\mathbf{N}(t)]$, i.e. these quantities measure the number and average number of points of $\Pi$ that arrive before time $t$, respectively. In this paper, we assume $\mu(t)$ has derivative $\mu'(t)$, and assume for each $t \geq 0$ there is a conditional probability distribution $P_{X|T=t}$ with density $p(x \mid t)$, such that we can write the mean measure as $\mu(A) = \int_A p(x \mid t)\mu'(t) \, dx \, dt$.

**Simulating $\Pi$:** A simple method to simulate $\Pi$ on a computer is to realize it in time order, i.e. at step $n$, simulate $T_n$ and then use it to simulate $X_n \sim P_{X|T=T_n}$. We can find the distribution of $\Pi$'s first arrival by noting that no point of $\Pi$ can come before it and hence $\mathbb{P}[T_1 \geq t] = \mathbb{P}[\mathbf{N}(t) = 0] = \exp(-\mu(t))$, where the second equality follows since $\mathbf{N}(t)$ is Poisson distributed with mean $\mu(t)$. A particularly important case is when $\Pi$ is *time-homogeneous*, i.e. $\mu(t) = \lambda t$ for some $\lambda > 0$, in which case $T_1 \sim \mathrm{Exp}(\lambda)$ is an exponential random variable with *rate* $\lambda$. In fact, all of $\Pi$'s inter-arrival times $\Delta_n = T_n - T_{n-1}$ for $n \geq 1$ share this simple distribution, where we set $T_0 = 0$. To see this, note that

**Algorithm 1:** Generating a $(\lambda, P_{X|T})$-Poisson process.

---

**Input** : Time rate $\lambda$,
            Spatial distribution $P_{X|T}$

$T_0 \leftarrow 0$
**for** $n = 1, 2, \ldots$ **do**
    $\Delta_n \sim \mathrm{Exp}(\lambda)$
    $T_n \leftarrow T_{n-1} + \Delta_n$
    $X_n \sim P_{X|T=T_n}$
    **yield** $(T_n, X_n)$
**end**

---

$$\mathbb{P}[\Delta_n \geq t \mid T_{n-1}] = \mathbb{P}\left[\mathbf{N}\left([T_{n-1}, T_{n-1} + t) \times \mathbb{R}^d\right) = 0 \mid T_{n-1}\right] = \exp(-\lambda t), \tag{1}$$

i.e. all $\Delta_n \mid T_{n-1} \sim \mathrm{Exp}(\lambda)$. Therefore, we can use the above procedure to simulate time-homogeneous Poisson processes, described in Algorithm 1. We will refer to a time-homogeneous Poisson process with time rate $\lambda$ and spatial distribution $P_{X|T}$ as a $(\lambda, P_{X|T})$-Poisson process.

**Rejection sampling using $\Pi$:** Rejection sampling is a technique to simulate samples from a *target distribution $Q$* using a *proposal distribution $P$*, assuming we can find an upper bound $M > 0$ for

| **Algorithm 2:** Standard rejection sampler. | **Algorithm 3:** Greedy Poisson rejection sampler. |
|---|---|
| **Input** : Proposal distribution $P$, Density ratio $r = dQ/dP$, Upper bound $M$ for $r$. | **Input** : Proposal distribution $P$, Density ratio $r = dQ/dP$, Stretch function $\sigma$. |
| **Output**: Sample $X \sim Q$ and its index $N$. | **Output**: Sample $X \sim Q$ and its index $N$. |
| // Generator for a $(1, P)$-Poisson process using Algorithm 1. | // Generator for a $(1, P)$-Poisson process using Algorithm 1. |
| $\Pi \leftarrow \texttt{SimulatePP}(1, P)$ | $\Pi \leftarrow \texttt{SimulatePP}(1, P)$ |
| **for** $n = 1, 2, \ldots$ **do** | **for** $n = 1, 2, \ldots$ **do** |
|   $(T_n, X_n) \leftarrow \texttt{next}(\Pi)$ |   $(T_n, X_n) \leftarrow \texttt{next}(\Pi)$ |
|   $U_n \sim \mathrm{Unif}(0, 1)$ |   **if** $T_n < \sigma\left(r(X_n)\right)$ **then** |
|   **if** $U_n < r(X_n)/M$ **then** |     **return** $X_n, n$ |
|     **return** $X_n, n$ |   **end** |
|   **end** | **end** |
| **end** | |

their density ratio $r = dQ/dP$ (technically, the Radon-Nikodym derivative). We can formulate this procedure using a Poisson process: we simulate the arrivals $(T_n, X_n)$ of a $(1, P)$-Poisson process $\Pi$, but we only keep them with probability $r(X_n)/M$, otherwise, we delete them. This algorithm is described in Algorithm 2; its correctness is guaranteed by the *thinning theorem* (Maddison, 2016).

**Rejection sampling is suboptimal:** Using Poisson processes to formulate rejection sampling highlights a subtle but crucial inefficiency: it does not make use of $\Pi$'s temporal structure and only uses the spatial coordinates. GPRS fixes this by using a rejection criterion that does depend on the time variable. As we show, this significantly speeds up sampling for certain classes of distributions.

## 2.2 Channel Simulation

The main motivation for our work is to develop a *one-shot channel simulation protocol* using the sampling algorithm we derive in Section 3. Channel simulation is of significant theoretical and practical interest. Recent works used it to compress neural network weights, achieving state-of-the-art performance (Havasi et al., 2018); to perform image compression using variational autoencoders (Flamich et al., 2020) and diffusion models with perfect realism (Theis et al., 2022); and to perform differentially private federated learning by compressing noisy gradients (Shah et al., 2022).

One-shot channel simulation is a communication problem between two parties, Alice and Bob, sharing a joint distribution $P_{\mathbf{x}, \mathbf{y}}$ over two correlated random variables $\mathbf{x}$ and $\mathbf{y}$, where we assume that Alice and Bob can simulate samples from the marginal $P_{\mathbf{x}}$. In a single round of communication, Alice receives a sample $\mathbf{y} \sim P_{\mathbf{y}}$ from the marginal distribution over $\mathbf{y}$. Then, she needs to send the minimum number of bits to Bob such that he can **simulate a single sample** from the conditional distribution $\mathbf{x} \sim P_{\mathbf{x}|\mathbf{y}}$. Note that Bob **does not want to learn** $P_{\mathbf{x}|\mathbf{y}}$; he just wants to simulate a single sample from it. Surprisingly, when Alice and Bob have access to *shared randomness*, e.g. by sharing the seed of their random number generator before communication, they can solve channel simulation very efficiently. Mathematically, in this paper we will always model this shared randomness by some time-homogeneous Poisson process (or processes) $\Pi$, since given a shared random seed, Alice and Bob can always simulate the same process using Algorithm 1. Then, the average coding cost of $\mathbf{x}$ given $\Pi$ is its conditional entropy $\mathbb{H}[\mathbf{x} \mid \Pi]$ and, surprisingly, it is always upper bounded by $\mathbb{H}[\mathbf{x} \mid \Pi] \leq I[\mathbf{x}; \mathbf{y}] + \log_2(I[\mathbf{x}; \mathbf{y}] + 1) + \mathcal{O}(1)$, where $I[\mathbf{x}; \mathbf{y}]$ is the mutual information between $\mathbf{x}$ and $\mathbf{y}$ (Li & El Gamal, 2018). This is an especially curious result, given that in many cases $\mathbb{H}[\mathbf{x}]$ is infinite while $I[\mathbf{x}; \mathbf{y}]$ is finite, e.g. when $\mathbf{x}$ is a continuous variable. In essence, this result means that given the additional structure $P_{\mathbf{x}, \mathbf{y}}$, channel simulation protocols can "offload" an infinite amount of information into the shared randomness, and only communicate the finitely many "necessary" bits.

**An example channel simulation protocol with rejection sampling:** Given $\Pi$ and $\mathbf{y} \sim P_{\mathbf{y}}$, Alice sets $Q \leftarrow P_{\mathbf{x}|\mathbf{y}}$ as the target and $P \leftarrow P_{\mathbf{x}}$ as the proposal distribution with density ratio $r = dQ/dP$, and run the rejection sampler in Algorithm 2 to find the first point of $\Pi$ that was not deleted. She counts the number of samples $N$ she had to simulate before acceptance and sends this number to Bob. He can then decode a sample $\mathbf{x} \sim P_{\mathbf{x}|\mathbf{y}}$ by selecting the spatial coordinate of the $N$th arrival of $\Pi$.

Unfortunately, this simple protocol is suboptimal. To see this, let $D_\infty[Q\|P] \overset{def}{=} \sup_{\mathbf{x}\in\Omega}\{\log_2 r(\mathbf{x})\}$ denote Rényi $\infty$-divergence of $Q$ from $P$, and recall two standard facts: (1) the best possible upper bound Alice can use for rejection sampling is $M_{opt} = \exp_2(D_\infty[Q\|P])$, where $\exp_2(x) = 2^x$, and (2), the number of samples $N$ drawn until acceptance is a geometric random variable with mean $M_{opt}$ (Maddison, 2016). We now state the two issues with rejection sampling that GPRS solves.

**Problem 1: Codelength.** By using the formula for the entropy of a geometric random variable and assuming Alice uses the best possible bound $M_{opt}$ in the protocol, we find that

$$\mathbb{H}[\mathbf{x} \mid \Pi] = \mathbb{E}_{\mathbf{y}\sim P_\mathbf{y}}[\mathbb{H}[N \mid \mathbf{y}]] \geq \mathbb{E}_{\mathbf{y}\sim P_\mathbf{y}}[D_\infty[P_{\mathbf{x}|\mathbf{y}}\|P_\mathbf{x}]] \overset{(a)}{\geq} I[\mathbf{x}; \mathbf{y}], \tag{2}$$

see Appendix I for the derivation. Unfortunately, inequality (a) can be *arbitrarily loose*, hence the average codelength scales with the expected $\infty$-divergence instead of $I[\mathbf{x}; \mathbf{y}]$, as would be optimal.

**Problem 2: Slow runtime.** We are interested in classifying the time complexity of our protocol. As we saw, for a target $Q$ and proposal $P$, Algorithm 2 draws $M_{opt} = \exp_2(D_\infty[Q\|P])$ samples on average. Unfortunately, under the computational hardness assumption RP $\neq$ NP, Agustsson & Theis (2020) showed that without any further assumptions, there is no sampler that scales polynomially in $D_{\mathrm{KL}}[Q\|P]$. However, with further assumptions, we can do much better, as we show in Section 3.1.

# 3 Greedy Poisson Rejection Sampling

We now describe GPRS; its pseudo-code is shown in Algorithm 3. This section assumes that $Q$ and $P$ are the target and proposal distributions, respectively, and $r = dQ/dP$ is their density ratio. Let $\Pi$ be a $(1, P)$-Poisson process. Our proposed rejection criterion is now embarrassingly simple: for an appropriate invertible function $\sigma : \mathbb{R}^+ \to \mathbb{R}^+$, accept the first arrival of $\Pi$ that falls under the graph of the composite function $\varphi = \sigma \circ r$, as illustrated in the left plot in Figure 1. We refer to $\sigma$ as the *stretch function* for $r$, as its purpose is to stretch the density ratio along the time-axis towards $\infty$.

**Deriving the stretch function (sketch, see Appendix A for details):** Let $\varphi = \sigma \circ r$, where for now $\sigma$ is an arbitrary invertible function on $\mathbb{R}^+$, let $U = \{(t, x) \in \Omega \mid t \leq \varphi(x)\}$ be the set of points under the graph of $\varphi$ and let $\tilde{\Pi} = \Pi \cap U$. By the restriction theorem (Kingman, 1992), $\tilde{\Pi}$ is also a Poisson process with mean measure $\tilde{\mu}(A) = \mu(A \cap U)$. Let $(\tilde{T}, \tilde{X})$ be the first arrival of $\tilde{\Pi}$, i.e. the first arrival of $\Pi$ under $\varphi$ and let $Q_\sigma$ be the marginal distribution of $\tilde{X}$. Then, as we show in Appendix A, the density ratio $dQ_\sigma/dP$ is given by

$$\frac{dQ_\sigma}{dP}(x) = \int_0^{\varphi(x)} \mathbb{P}[\tilde{T} \geq t]\, dt. \tag{3}$$

Now we pick $\sigma$ such that $Q_\sigma = Q$, for which we need to ensure that $dQ_\sigma/dP = r$. Substituting $t = \varphi(x)$ into Equation (3), and differentiating, we get $(\sigma^{-1})'(t) = \mathbb{P}[\tilde{T} \geq t]$. Since $(\tilde{T}, \tilde{X})$ falls under the graph of $\varphi$ by definition, we have $\mathbb{P}[\tilde{T} \geq t] = \mathbb{P}[\tilde{T} \geq t, r(\tilde{X}) \geq \sigma^{-1}(t)]$. By expanding the definition of the right-hand side, we obtain a time-invariant ODE for $\sigma^{-1}$:

$$(\sigma^{-1})' = w_Q(\sigma^{-1}) - \sigma^{-1} \cdot w_P(\sigma^{-1}), \quad \text{with} \quad \sigma^{-1}(0) = 0, \tag{4}$$

where $w_P(h) \overset{def}{=} \mathbb{P}_{Z\sim P}[r(Z) \geq h]$ and $w_Q$ defined analogously. Finally, solving the ODE, we get

$$\sigma(h) = \int_0^h \frac{1}{w_Q(\eta) - \eta \cdot w_P(\eta)}\, d\eta. \tag{5}$$

Remember that picking $\sigma$ according to Equation (5) ensures that GPRS is **correct by construction**. To complete the picture, in Appendix A we prove that $(\tilde{T}, \tilde{X})$ always exists and Algorithm 3 terminates with probability 1. We now turn our attention to analyzing the runtime of Algorithm 3 and surprisingly, we find that the expected runtime of GPRS matches that of standard rejection sampling; see Appendix B.2 for the proof. Note that in our analyses, we identify GPRS's time complexity with the number of arrivals of $\Pi$ it simulates. Moreover, we assume that we can evaluate $\sigma$ in $\mathcal{O}(1)$ time.

**Theorem 3.1** (Expected Runtime). *Let $Q$ and $P$ be the target and proposal distributions for Algorithm 3, respectively, and $r = dQ/dP$ their density ratio. Let $N$ denote the number of samples simulated by the algorithm before it terminates. Then,*

$$\mathbb{E}[N] = \exp_2(D_\infty[Q\|P]) \quad \text{and} \quad \mathbb{V}[N] \geq \exp_2(D_\infty[Q\|P]), \tag{6}$$

*where $\mathbb{V}[\cdot]$ denotes the variance of a random variable.*

Furthermore, using ideas from the work of Liu & Verdu (2018), we can show the following upper bound on the fractional moments of the index; see Appendix B.3 for the proof.

**Theorem 3.2** (Fractional Moments of the Index). *Let $Q$ and $P$ be the target and proposal distributions for Algorithm 3, respectively, and $r = dQ/dP$ their density ratio. Let $N$ denote the number of samples simulated by the algorithm before it terminates. Then, for $\alpha \in (0, 1)$ we have*

$$\mathbb{E}[N^\alpha] \leq \frac{1}{1-\alpha} \left( 1 + \frac{D_\infty[Q\|P]}{\log_2 e} \right) \exp_2 \left( \alpha D_{\frac{1}{1-\alpha}}[Q\|P] \right), \tag{7}$$

*where $D_\alpha[Q\|P] = \frac{1}{\alpha-1} \log_2 \mathbb{E}_{Z \sim Q}\left[ r(Z)^{\alpha-1} \right]$ is the Rényi $\alpha$-divergence of $Q$ from $P$.*

**GPRS induces a channel simulation protocol:** We use an analogous solution to the standard rejection sampling-based scheme in Section 2.2, to obtain a channel simulation protocol for a pair of correlated random variables $\mathbf{x}, \mathbf{y} \sim P_{\mathbf{x},\mathbf{y}}$. First, we assume that both the encoder and the decoder have access to the same realization of a $(1, P_{\mathbf{x}})$-Poisson process $\Pi$, which they will use as their shared randomness. In a single round of communication, the encoder receives $\mathbf{y} \sim P_{\mathbf{y}}$, sets $Q \leftarrow P_{\mathbf{x}|\mathbf{y}}$ as the target and $P \leftarrow P_{\mathbf{x}}$ as the proposal distribution, and runs Algorithm 3 to find the index $N$ of the accepted arrival in $\Pi$. Finally, following Li & El Gamal (2018), they use a zeta distribution $\zeta(n \mid s) \propto n^{-s}$ with $s^{-1} = I[\mathbf{x}; \mathbf{y}] + 2 \log_2 e$ to encode $N$ using entropy coding. Then, the decoder simulates a $P_{\mathbf{x}|\mathbf{y}}$-distributed sample by looking up the spatial coordinate $X_N$ of $\Pi$'s $N$th arrival. The following theorem shows that this protocol is optimally efficient; see Appendix B.4 for the proof.

**Theorem 3.3** (Expected Codelength). *Let $P_{\mathbf{x},\mathbf{y}}$ be a joint distribution over correlated random variables $\mathbf{x}$ and $\mathbf{y}$, and let $\Pi$ be the $(1, P_{\mathbf{x}})$-Poisson process used by Algorithm 3. Then, the above protocol induced by the the algorithm is optimally efficient up to a constant, in the sense that*

$$\mathbb{H}[\mathbf{x} \mid \Pi] \leq I[\mathbf{x}; \mathbf{y}] + \log_2 (I[\mathbf{x}; \mathbf{y}] + 1) + 6. \tag{8}$$

Note, that this result shows we can use GPRS as an alternative to the *Poisson functional representation* to prove of the *strong functional representation lemma* (Theorem 1; Li & El Gamal, 2018).

**GPRS in practice:** The real-world applicability of Algorithm 3 is limited by the following three computational obstacles: (1) GPRS requires exact knowledge of the density ratio $dQ/dP$ to evaluate $\varphi$, which immediately excludes a many practical problems where $dQ/dP$ is known only up to a normalizing constant. (2) To compute $\sigma$, we need to evaluate Equation (4) or (5), which requires us to compute $w_P$ and $w_Q$, which is usually intractable for general $dQ/dP$. Fortunately, in some special cases they can be computed analytically: in Appendix G, we provide the expressions for discrete, uniform, triangular, Gaussian, and Laplace distributions. (3) However, even when we can compute $w_P$ and $w_Q$, analytically evaluating the integral in Equation (5) is usually not possible. Moreover, solving it numerically is unstable as $\sigma$ is unbounded. Instead, we numerically solve for $\sigma^{-1}$ using Equation (4) in practice, which fortunately turns out to be stable.

However, we emphasize that GPRS's *raison d'être* is to perform channel simulation using the protocol we outline above or the ones we present in Section 3.1, not general-purpose random variate simulation. Practically relevant channel simulation problems, such as encoding a sample from the latent posterior of a VAE (Flamich et al., 2020) or encoding variational implicit neural representations (Guo et al., 2023; He et al., 2023) usually involve simple distributions such as Gaussians. In these cases, we have readily available formulae for $w_P$ and $w_Q$ and can cheaply evaluate $\sigma^{-1}$ via numerical integration.

**GPRS as greedy search:** The defining feature of GPRS is that its acceptance criterion at each step is *local*, since if at step $n$ the arrival $(T_n, X_n)$ in $\Pi$ falls under $\varphi$, the algorithm accepts it. Thus it is *greedy* search procedure. This is in contrast with A* sampling (Maddison et al., 2014), whose acceptance criterion is *global*, as the acceptance of the $n$th arrival $(T_n, X_n)$ depends on all other points of $\Pi$ in the general case. In other words, using the language of Liu & Verdu (2018), GPRS is a *causal* sampler, as its acceptance criterion at each step only depends on the samples the algorithm has examined in previous steps. On the other hand, A* sampling is *non-causal* as its acceptance criterion in each step depends on future samples as well. Surprisingly, this difference between the search criteria does not make a difference in the average runtimes and codelengths in the general case. However, as we show in the next section, GPRS can be much faster in special cases.

## 3.1 Speeding up the greedy search

This section discusses two ways to improve the runtime of GPRS. First, we show how we can utilize available parallel computing power to speed up Algorithm 3. Second, we propose an advanced search

| **Algorithm 4:** Parallel GPRS with $J$ available threads. | **Algorithm 5:** Branch-and-bound GPRS on $\mathbb{R}$ with unimodal $r$ |
|---|---|
| **Input** : Proposal distribution $P$, Density ratio $r = dQ/dP$, Stretch function $\sigma$, Number of parallel threads $J$. | **Input** : Proposal distribution $P$, Density ratio $r = dQ/dP$, Stretch function $\sigma$, Location $x^*$ of the mode of $r$. |
| **Output:** Sample $X \sim Q$ and its code $(j^*, N_{j^*})$. | **Output:** Sample $X \sim Q$ and its heap index $H$. |

**Algorithm 4:**

$T^*, X^*, j^*, N_{j^*} \leftarrow \infty, \texttt{nil}, \texttt{nil}, \texttt{nil}$
**in parallel for** $j = 1, \ldots, J$ **do**
    $\Pi_j \leftarrow \texttt{SimulatePP}(1/J, P)$
    **for** $n_j = 1, 2, \ldots$ **do**
        $\left( T_{n_j}^{(j)}, X_{n_j}^{(j)} \right) \leftarrow \texttt{next}(\Pi_j)$
        **if** $T^* < T_{n_j}^{(j)}$ **then**
            **terminate thread** $j$.
        **end**
        **if** $T_{n_j}^{(j)} < \sigma\left( r\left( X_{n_j}^{(j)} \right) \right)$ **then**
            $T^*, X^*, j^*, N_{j^*} \leftarrow T_{n_j}^{(j)}, X_{n_j}^{(j)}, j, n_j$
            **terminate thread** $j$.
        **end**
    **end**
**end**
**return** $X^*, (j^*, N_{j^*})$

**Algorithm 5:**

$T_0, H, B \leftarrow (0, 1, \mathbb{R})$
**for** $d = 1, 2, \ldots$ **do**
    $X_d \sim P|_B$
    $\Delta_d \sim \mathrm{Exp}\left( P(B) \right)$
    $T_d \leftarrow T_{d-1} + \Delta_d$
    **if** $T_d < \sigma(r(X_d))$ **then**
        **return** $X_d, H$
    **end**
    **if** $X_d \geq x^*$ **then**
        $B \leftarrow B \cap (-\infty, X_d)$
        $H \leftarrow 2H$
    **else**
        $B \leftarrow B \cap (X_d, \infty)$
        $H \leftarrow 2H + 1$
    **end**
**end**

strategy when the spatial domain $\Omega$ has more structure and obtain a **super-exponential improvement** in the runtime from $\exp_2\left( D_\infty[Q\|P] \right)$ to $\mathcal{O}(D_{\mathrm{KL}}[Q\|P])$ in certain cases.

**Parallel GPRS:** The basis for parallelizing GPRS is the *superposition theorem* (Kingman, 1992): For a positive integer $J$, let $\Pi_1, \ldots \Pi_J$ all be $(1/J, P)$-Poisson processes; then, $\Pi = \bigcup_{j=1}^{J} \Pi_j$ is a $(1, P)$-Poisson process. As shown in Algorithm 4, this result makes parallelizing GPRS very simple given $J$ parallel threads: First, we independently look for the first arrivals of $\Pi_1, \ldots, \Pi_J$ under the graph of $\varphi$, yielding $\left( T_{N_1}^{(1)}, X_{N_1}^{(1)} \right), \ldots, \left( T_{N_J}^{(J)}, X_{N_J}^{(J)} \right)$, respectively, where the $N_j$ corresponds to the index of $\Pi_j$'s first arrival under $\varphi$. Then, we select the candidate with the earliest arrival time, i.e. $j^* = \arg\min_{j \in \{1, \ldots, J\}} T_{N_j}^{(j)}$. Now, by the superposition theorem, $\left( T_{N_{j^*}}^{(j^*)}, X_{N_{j^*}}^{(j^*)} \right)$ is the first arrival of $\Pi$ under $\varphi$, and hence $X_{N_{j^*}}^{(j^*)} \sim Q$. Finally, Alice encodes the sample via the tuple $(j^*, N_{j^*})$, i.e. which of the $J$ processes the first arrival occurred in, and the index of the arrival in $\Pi_{j^*}$. See the middle graphic in Figure 1 for an example with $J = 2$ parallel threads.

Our next results show that parallelizing GPRS results in a linear reduction in both the expectation and variance of its runtime and a more favourable codelength guarantee for channel simulation.

**Theorem 3.4** (Expected runtime of parallelized GPRS)**.** *Let $Q, P$ and $r$ be defined as above, and let $\nu_j$ denote the random variable corresponding to the number of samples simulated by thread $j$ in Algorithm 4 using $J$ threads. Then, for all $j$,*

$$\mathbb{E}[\nu_j] = \left( \exp_2\left( D_\infty[Q\|P] \right) - 1 \right) / J + 1 \quad and \quad \mathbb{V}[\nu_j] \geq \left( \exp_2\left( D_\infty[Q\|P] \right) - 1 \right) / J + 1. \quad (9)$$

**Theorem 3.5** (Expected codelength of parallelized GPRS)**.** *Let $P_{\mathbf{x},\mathbf{y}}$ be a joint distribution over correlated random variables $\mathbf{x}$ and $\mathbf{y}$, and let $\Pi_1, \ldots, \Pi_J$ be $(1/J, P_{\mathbf{x}})$-Poisson processes. Then, assuming $\log_2 J \leq I[\mathbf{x}; \mathbf{y}]$, parallelized GPRS induces a channel simulation protocol such that*

$$\mathbb{H}[\mathbf{x} \mid \Pi_1, \ldots, \Pi_J] \leq I[\mathbf{x}; \mathbf{y}] + \log_2\left( I[\mathbf{x}; \mathbf{y}] - \log_2 J + 1 \right) + 8. \quad (10)$$

See Appendix C for the proofs. Note that we can use the same parallelisation argument with the appropriate modifications to speed up A$^*$ sampling / coding too.

**Branch-and-bound GPRS on $\mathbb{R}$:** We briefly restrict our attention to problems on $\mathbb{R}$ when the density ratio $r$ is unimodal, as we can exploit this additional structure to more efficiently search for $\Pi$'s first

arrival under $\varphi$. Consider the example in the right graphic in Figure 1: we simulate the first arrival $(T_1, X_1)$, and reject it, since $T_1 > \varphi(X_1)$. Since $r$ is unimodal by assumption and $\sigma$ is increasing, $\varphi$ is also unimodal. Hence, for $x < X_1$ we must have $\varphi(x) < \varphi(X_1)$, while the arrival time of any of the later arrivals will be larger than $T_1$. Therefore, none of the arrivals to the left of $X_1$ will fall under $\varphi$ either! Hence, it is sufficient to simulate $\Pi$ to the right of $X_1$, which we can easily do using generalized inverse transform sampling.[1] We repeat this argument for all further arrivals to obtain an efficient search procedure for finding $\Pi$'s first arrival under $\varphi$, described in Algorithm 5: We simulate the next arrival $(T_B, X_B)$ of $\Pi$ within some bounds $B$, starting with $B = \Omega$. If $T_B \leq \varphi(X_B)$ we accept; otherwise, we truncate the bound to $B \leftarrow B \cap (-\infty, X_B)$, or $B \leftarrow B \cap (X_B, \infty)$ based on where $X_B$ falls relative to $\varphi$'s mode. We repeat these two steps until we find the first arrival.

Since Algorithm 5 does not simulate every point of $\Pi$, we cannot use the index $N$ of the accepted arrival to obtain a channel simulation protocol as before. Instead, we encode the *search path*, i.e. whether we chose the left or the right side of our current sample at each step. Similarly to A* coding, we encode the path using its *heap index* (Flamich et al., 2022): the root has index $H_{root} = 1$, and for a node with index $H$, its left child is assigned index $2H$ and its right child $2H + 1$. As the following theorems show, this version of GPRS is, in fact, optimally efficient; see Appendix E for proofs.

**Theorem 3.6** (Expected Runtime of GPRS with binary search). *Let $Q, P$ and $r$ be defined as above, and let $D$ denote the number of samples simulated by Algorithm 5 and $\lambda = 1/\log_2(4/3)$ Then,*

$$\mathbb{E}[D] \leq \lambda \cdot D_{\mathrm{KL}}[Q\|P] + \mathcal{O}(1). \tag{11}$$

**Theorem 3.7** (Expected Codelength of GPRS with binary search). *Let $P_{\mathbf{x},\mathbf{y}}$ be a joint distribution over correlated random variables $\mathbf{x}$ and $\mathbf{y}$, and let $\Pi$ be a $(1, P_{\mathbf{x}})$-Poisson process. Then, GPRS with binary search induces a channel simulation protocol such that*

$$\mathbb{H}[\mathbf{x} \mid \Pi] \leq I[\mathbf{x}; \mathbf{y}] + 2\log_2\left(I[\mathbf{x}; \mathbf{y}] + 1\right) + 11. \tag{12}$$

**Branch-and-bound GPRS with splitting functions:** With some additional machinery, Algorithm 5 can be extended to more general settings, such as $\mathbb{R}^D$ and cases where $r$ is not unimodal, by introducing the notion of a *splitting function*. For a region of space, $B \subseteq \Omega$, a splitting function `split` simply returns a binary partition of the set, i.e. $\{L, R\} = \mathtt{split}(B)$, such that $L \cap R = \emptyset$ and $L \cup R = B$. In this case, we can perform a similar tree search to Algorithm 5, captured in Algorithm 6. Recall that $(\tilde{T}, \tilde{X})$ is the first arrival of $\Pi$ under $\varphi$. Starting with the whole space $B = \Omega$, we simulate the next arrival $(T, X)$ of $\Pi$ in $B$ at each step. If we reject it, we partition $B$ into two parts, $L$ and $R$, using `split`. Then, with probability $\mathbb{P}[\tilde{X} \in R \mid \tilde{X} \in B, \tilde{T} \geq T]$, we continue searching through the arrivals of $\Pi$ only in $R$, and only in $L$ otherwise. We show the correctness of this procedure in Appendix F and describe how to compute $\mathbb{P}[\tilde{X} \in R \mid \tilde{X} \in B, \tilde{T} \geq T]$ and $\sigma$ in practice. This splitting procedure is analogous to the general version of A* sampling/coding (Maddison et al., 2014; Flamich et al., 2022), which is parameterized by the same splitting functions. Note that Algorithm 5 is a special case of Algorithm 6, where $\Omega = \mathbb{R}$, $r$ is unimodal, and at each step for an interval bound $B = (a, b)$ and sample $X \in (a, b)$ we split $B$ into $\{(a, X), (X, b)\}$.

## 4 Experiments

We compare the average and one-shot case efficiency of our proposed variants of GPRS and a couple of other channel simulation protocols in Figure 2. See the figure's caption and Appendix H for details of our experimental setup. The top two plots in Figure 2 demonstrate that our methods' expected runtimes and codelengths align well with our theorems' predictions and compare favourably to other methods. Furthermore, we find that the mean performance is a *robust statistic* for the binary search-based variants of GPRS in that it lines up closely with the median, and the interquartile range is quite narrow. However, we also see that Theorem 3.7 is slightly loose empirically. We conjecture that the coefficient of the $\log_2(I[\mathbf{x}; \mathbf{y}] + 1)$ term in Equation (12) could be reduced to 1, and that the constant term could be improved as well.

On the other hand, the bottom plot in Figure 2 demonstrates the most salient property of the binary search variant of GPRS: unlike all previous methods, its runtime scales with $D_{\mathrm{KL}}[Q\|P]$ and not $D_\infty[Q\|P]$. Thus, we can apply it to a larger family of channel simulation problems, where $D_{\mathrm{KL}}[Q\|P]$ is finite but $D_\infty[Q\|P]$ is large or even infinite, and other methods would not terminate.

---

[1]Suppose we have a real-valued random variable $X$ with CDF $F$ and inverse CDF $F^{-1}$. Then, we can simulate $X$ truncated to $[a, b]$ by simulating $U \sim \mathrm{Unif}[F(a), F(b)]$ and computing $X = F^{-1}(U)$.

**Algorithm 6:** Branch-and-bound GPRS with splitting function.

**Input** : Proposal distribution $P$,
Density ratio $r = dQ/dP$,
Stretch function $\sigma$,
Splitting function `split`.
**Output** : Sample $X \sim Q$ and its heap index $H$

$T_0, H, B \leftarrow (0, 1, \mathbb{R})$
**for** $d = 1, 2, \ldots$ **do**
    $X_d \sim P|_B$
    $\Delta_d \sim \mathrm{Exp}(P(B))$
    $T_d \leftarrow T_{d-1} + \Delta_d$
    **if** $T_d < \sigma(r(X_d))$ **then**
        **return** $X_d, H$
    **else**
        $B_0, B_1 \leftarrow \texttt{split}(B)$
        $\rho \leftarrow$
        $\quad \mathbb{P}[\tilde{X} \in B_1 \mid \tilde{X} \in B, \tilde{T} \geq T_d]$
        $\beta \leftarrow \mathrm{Bernoulli}(\rho)$
        $H \leftarrow 2H + \beta$
        $B \leftarrow B_\beta$
    **end**
**end**

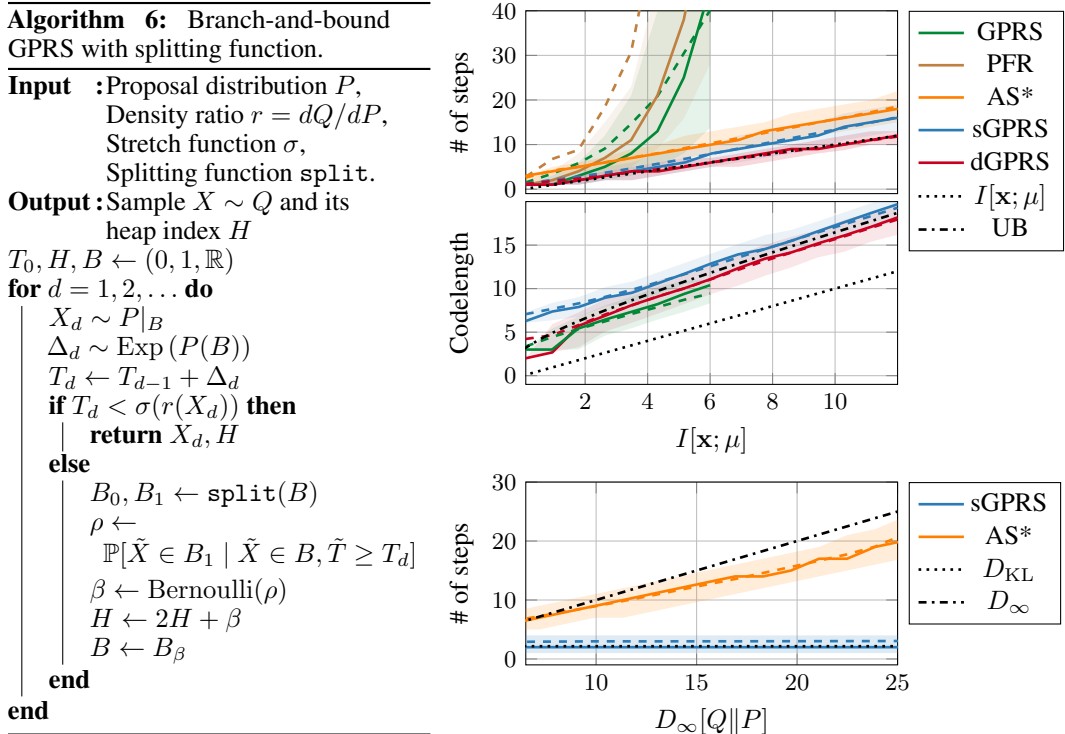

Figure 2: **Left:** Binary search GPRS with arbitrary splitting function. **Right:** Performance comparison of different channel simulation protocols. In each plot, *dashed lines* indicate the mean, *solid lines* the median and the *shaded areas* the 25 - 75 percentile region of the relevant performance metric. We computed the statistics over 1000 runs for each setting. **Top right:** Runtime comparison on a 1D Gaussian channel simulation problem $P_{\mathbf{x},\mu}$, plotted against increasing mutual information $I[\mathbf{x};\mu]$. Alice receives $\mu \sim \mathcal{N}(0, \sigma^2)$ and encodes a sample $\mathbf{x} \mid \mu \sim \mathcal{N}(\mu, 1)$ to Bob. The abbreviations in the legend are: *GPRS* – Algorithm 3; *PFR* – Poisson functional representation / Global-bound A* coding (Li & El Gamal, 2018; Flamich et al., 2022), *AS** – Split-on-sample A* coding (Flamich et al., 2022); *sGPRS* – split-on-sample GPRS (Algorithm 5); *dGPRS* – GPRS with dyadic `split` (Algorithm 6). **Middle right:** Average codelength comparison of our proposed algorithms on the same channel simulation problem as above. *UB* in the legend corresponds to an *upper bound* of $I[\mathbf{x};\mu] + \log_2(I[\mathbf{x};\mu] + 1) + 2$ bits. We estimate the algorithms' expected codelengths by encoding the indices returned by GPRS using a Zeta distribution $\zeta(n \mid \lambda) \propto n^{-\lambda}$ with $\lambda = 1 + 1/I[\mathbf{x};\mathbf{y}]$ in each case, which is the optimal maximum entropy distribution for this problem setting (Li & El Gamal, 2018). **Bottom right:** One-shot runtime comparison of sGPRS with AS* coding. Alice encodes samples from a target $Q = \mathcal{N}(m, s^2)$ using $P = \mathcal{N}(0, 1)$ as the proposal. We computed $m$ and $s^2$ such that $D_{\mathrm{KL}}[Q\|P] = 2$ bits for each problem, but $D_\infty[Q\|P]$ increases. GPRS' runtime stays fixed as it scales with $D_{\mathrm{KL}}[Q\|P]$, while the runtime of A* keeps increasing.

## 5   Related Work

**Poisson processes for channel simulation:** Poisson processes were introduced to the channel simulation literature by Li & El Gamal (2018) via their construction of the *Poisson functional representation* (PFR) in their proof of the strong functional representation lemma. Flamich et al. (2022) observed that the PFR construction is equivalent to a certain variant of A* sampling (Maddison et al., 2014; Maddison, 2016). Thus, they proposed an optimized version of the PFR called A* coding, which achieves $\mathcal{O}(D_\infty[Q\|P])$ runtime for one-dimensional unimodal distributions. GPRS was mainly inspired by A* coding, and they are *dual* constructions to each other in the following sense: *Depth-limited A* coding* can be thought of as an *importance sampler*, i.e. a Monte Carlo algorithm that returns an approximate sample in fixed time. On the other hand, GPRS is a *rejection sampler*, i.e. a Las Vegas algorithm that returns an exact sample in random time.

**Channel simulation with dithered quantization:** *Dithered quantization* (DQ; Ziv, 1985) is an alternative to rejection and importance sampling-based approaches to channel simulation. DQ exploits that for any $c \in \mathbb{R}$ and $U, U' \sim \text{Unif}(-1/2, 1/2)$, the quantities $\lfloor c - U \rceil + U$ and $c + U'$ are equal in distribution. While DQ has been around for decades as a tool to model and analyze quantization error, Agustsson & Theis (2020) reinterpreted it as a channel simulation protocol and used it to develop a VAE-based neural image compression algorithm. Unfortunately, basic DQ only allows uniform target distributions, limiting its applicability. As a partial remedy, Theis & Yosri (2022) showed DQ could be combined with other channel simulation protocols to speed them up and thus called their approach hybrid coding (HQ). Originally, HQ required that the target distribution be compactly supported, which was lifted by Flamich & Theis (2023), who developed an adaptive rejection sampler using HQ. In a different vein, Hegazy & Li (2022) generalize and analyze a method proposed in the appendix of Agustsson & Theis (2020) and show that DQ can be used to realize channel simulation protocols for one-dimensional symmetric, unimodal distributions.

**Greedy Rejection Coding:** Concurrently to our work, Flamich et al. (2023) introduce greedy rejection coding (GRC) which generalizes Harsha et al.'s rejection sampling algorithm to arbitrary probability spaces and arbitrary splitting functions, extending the work of Flamich & Theis (2023). Furthermore, they also utilize a space-partitioning procedure to speed up the convergence of their sampler and prove that a variant of their sampler also achieves optimal runtime for one-dimensional problems where $dQ/dP$ is unimodal. However, the construction of their method differs significantly from ours. GRC is a direct generalization of Harsha et al. (2007)'s algorithm and, thus, a more "conventional" rejection sampler, while we base our construction on Poisson processes. Thus, our proof techniques are also significantly different. It is an interesting research question whether GRC could be formulated using Poisson processes, akin to standard rejection sampling in Algorithm 2, as this could be used to connect the two algorithms and improve both.

# 6 Discussion and Future Work

Using the theory of Poisson processes, we constructed greedy Poisson rejection sampling. We proved the correctness of the algorithm and analyzed its runtime, and showed that it could be used to obtain a channel simulation protocol. We then developed several variations on it, analyzed their runtimes, and showed that they could all be used to obtain channel simulation protocols. As the most significant result of the paper, we showed that using the binary search variant of GPRS we can achieve $\mathcal{O}(D_{\text{KL}}[Q\|P])$ runtime for arbitrary one-dimensional, unimodal density ratios, significantly improving upon the previous best $\mathcal{O}(D_{\infty}[Q\|P])$ bound by A* coding.

There are several interesting directions for future work. From a practical perspective, the most pressing question is whether efficient channel simulation algorithms exist for multivariate problems; finding an efficient channel simulation protocol for multivariate Gaussians would already have far-reaching practical consequences. We also highlight an interesting theoretical question. We found that the most general version of GPRS needs to simulate exactly $\exp_2(D_{\infty}[Q\|P])$ samples in expectation, which matches A* sampling. From Agustsson & Theis (2020), we know that for general problems, the average number of simulated samples has to be at least on the order of $\exp_2(D_{\text{KL}}[Q\|P])$. Hence we ask whether this lower bound can be tightened to match the expected runtime of GPRS and A* sampling or whether there exists an algorithm that achieves a lower runtime.

# 7 Acknowledgements

We are grateful to Lennie Wells for the many helpful discussions throughout the project, particularly for suggesting the nice argument for Lemma D.2; Peter Wildemann for his advice on solving some of the differential equations involved in the derivation of GPRS and Stratis Markou for his help that aided our understanding of some high-level concepts so that we could derive Algorithm 6. We would also like to thank Lucas Theis for the many enjoyable early discussions about the project and for providing helpful feedback on the manuscript; and Cheuk Ting Li for bringing Liu & Verdu's work to our attention. Finally, we would like to thank Daniel Goc for proofreading the manuscript and Lorenzo Bonito for helping to design Figures 1 and 2 and providing feedback on the early version of the manuscript. GF acknowledges funding from DeepMind.

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

# A  Measure-theoretic Construction of Greedy Poisson Rejection Sampling

In this section, we provide a construction of greedy Poisson rejection sampling (GPRS) in its greatest generality. However, we first establish the notation for the rest of the appendix.

**Notation:** In what follows, we will always denote GPRS's target distribution as $Q$ and its proposal distribution as $P$. We assume that $Q$ and $P$ are Borel probability measures over some Polish space $\Omega$ with Radon-Nikodym derivative $r \stackrel{def}{=} dQ/dP$. We denote the standard Lebesgue measure on $\mathbb{R}^n$ by $\lambda$. All logarithms are assumed to be to the base 2 denoted by $\log_2$. Similarly, we use the less common notation of $\exp_2$ to denote exponentiation with base 2, i.e. $\exp_2(x) = 2^x$. The relative entropy of $Q$ from $P$ is defined as $D_{\mathrm{KL}}[Q\|P] \stackrel{def}{=} \int_{\Omega} \log_2 r(x)\, dP(x)$ and the Rényi $\infty$-divergence as $D_{\infty}[Q\|P] \stackrel{def}{=} \mathrm{ess\,sup}_{x\in\Omega} \{\log r(x)\}$, where the essential supremum is taken with respect to $P$.

**Restricting a Poisson process "under the graph of a function":** We first consider the class of functions "under which" we will be restricting our Poisson processes. Let $\varphi : \Omega \to \mathbb{R}^+$ be a measurable function, such that $|\varphi(\Omega)| = |r(\Omega)|$, i.e. the image of $\varphi$ has the same cardinality as the image of $r = dQ/dP$. This implies that there exist bijections between $r(\Omega)$ and $\varphi(\Omega)$. Since both images are subsets of $\mathbb{R}^+$, both images have a linear order. Hence, a monotonically increasing bijection $\tilde{\sigma}$ exists such that $\varphi = \tilde{\sigma} \circ r$. Since $\tilde{\sigma}$ is monotonically increasing, we can extend it to a monotonically increasing continuous function $\sigma : \mathbb{R}^+ \to \mathbb{R}^+$. This fact is significant, as it allows us to reduce questions about functions of interest from $\Omega$ to $\mathbb{R}^+$ to invertible functions on the positive reals. Since $\sigma$ is continuous and monotonically increasing on the positive reals, it is invertible on the extended domain. We now consider restricting a Poisson process under the graph of $\varphi$, and work out the distribution of the spatial coordinate of the first arrival.

Let $\Pi = \{(T_n, X_n)\}_{n=1}^{\infty}$ be a Poisson process over $\mathbb{R}^+ \times \Omega$ with mean measure $\mu = \lambda \times P$. Let $U \stackrel{def}{=} \{(t, x) \in \Omega \mid t \le \varphi(x)\}$ be the set of points "under the graph" of $\varphi$. By the restriction theorem (Kingman, 1992), $\tilde{\Pi} \stackrel{def}{=} \Pi \cap U$ is a Poisson process with mean measure $\tilde{\mu}(A) = \mu(A \cap U)$ for an arbitrary Borel set $A$. As a slight abuse of notation, we define the shorthand $\tilde{\mu}(t) = \tilde{\mu}([0, t) \times \Omega)$ for the average number of points in $\tilde{\Pi}$ up to time $t$. Let $\mathbf{N}_{\tilde{\Pi}}$ denote the counting measure of $\tilde{\Pi}$ and let $(\tilde{T}, \tilde{X})$ be the first arrival of $\tilde{\Pi}$, i.e. the first point of the original process $\Pi$ that falls under the graph of $\varphi$, *assuming that it exists*. To develop GPRS, we first work out the distribution of $\tilde{X}$ for an arbitrary $\varphi$.

**Remark:** Note, that the construction below only holds if the first arrival is guaranteed to exist, otherwise the quantities below do not make sense. From a formal point of view, we would have to always condition on the event $\mathbf{N}_{\tilde{\Pi}}(\infty) \stackrel{def}{=} \lim_{t\to\infty} \mathbf{N}_{\tilde{\Pi}}(t) > 0$, e.g. we would need to write $\mathbb{P}[\tilde{T} \ge t \mid \mathbf{N}_{\tilde{\Pi}}(\infty) > 0]$. However, we make a "leap of faith" instead and assume that our construction is valid to obtain our "guesses" for the functions $\sigma$ and $\varphi$. Then, we show that for our specific guesses the derivation below is well-defined, i.e. namely that $\mathbb{P}[\mathbf{N}_{\tilde{\Pi}}(\infty) > 0] = 1$, which then implies $\mathbb{P}[\tilde{T} \ge t \mid \mathbf{N}_{\tilde{\Pi}}(\infty) > 0] = \mathbb{P}[\tilde{T} \ge t]$. To do this, note that

$$\mathbb{P}[\mathbf{N}_{\tilde{\Pi}}(\infty) > 0] = 1 - \mathbb{P}[\mathbf{N}_{\tilde{\Pi}}(\infty) = 0] \tag{13}$$
$$= 1 - \lim_{t\to\infty} e^{-\tilde{\mu}(t)}. \tag{14}$$

Hence, for the well-definition of our construction it is enough to show that for our choice of $\sigma$, $\tilde{\mu}(t) \to \infty$ as $t \to \infty$.

**Constructing $\sigma$:** We wish to derive

$$\frac{d\mathbb{P}[\tilde{X} = x]}{dP} = \int_0^{\infty} \frac{d\mathbb{P}[\tilde{T} = t, \tilde{X} = x]}{d(\lambda \times P)}\, dt. \tag{15}$$

To do this, recall that we defined $\mathbf{N}_{\tilde{\Pi}}(t) \overset{def}{=} \mathbf{N}_{\tilde{\Pi}}([0,t) \times \Omega)$ and define $\mathbf{N}_{\tilde{\Pi}}(s,t) \overset{def}{=} \mathbf{N}_{\tilde{\Pi}}([s,t] \times \Omega)$. Similarly, let $\tilde{\mu}(t) \overset{def}{=} \mathbb{E}[\mathbf{N}_{\tilde{\Pi}}(t)]$ and $\tilde{\mu}(s,t) \overset{def}{=} \mathbb{E}[\mathbf{N}_{\tilde{\Pi}}(s,t)] = \tilde{\mu}(t) - \tilde{\mu}(s)$. Then,

$$\mathbb{P}[\tilde{X} \in A, \tilde{T} \in dt] = \lim_{s \to t} \frac{\mathbb{P}[\tilde{X} \in A, \tilde{T} \in [s,t]]}{t-s} \tag{16}$$

$$= \lim_{s \to t} \frac{\mathbb{P}[\tilde{X} \in A, \mathbf{N}_{\tilde{\Pi}}(s,t) = 1, \mathbf{N}_{\tilde{\Pi}}(s) = 0]}{t-s} \tag{17}$$

$$= \lim_{s \to t} \frac{\mathbb{P}[\tilde{X} \in A \mid \mathbf{N}_{\tilde{\Pi}}(s,t) = 1]\mathbb{P}[\mathbf{N}_{\tilde{\Pi}}(s,t) = 1]\mathbb{P}[\mathbf{N}_{\tilde{\Pi}}(s) = 0]}{t-s}, \tag{18}$$

where the last equality holds, because $\mathbf{N}_{\tilde{\Pi}}(s) \perp \mathbf{N}_{\tilde{\Pi}}(s,t)$, since $[0,s) \times \Omega$ and $[s,t] \times \Omega$ are disjoint. By Lemma 3 from Maddison (2016),

$$\mathbb{P}[\tilde{X} \in A \mid \mathbf{N}_{\tilde{\Pi}}(s,t) = 1] = \mathbb{P}[(\tilde{T}, \tilde{X}) \in [s,t] \times A \mid \mathbf{N}_{\tilde{\Pi}}(s,t) = 1] \tag{19}$$

$$= \frac{\tilde{\mu}([s,t] \times A)}{\tilde{\mu}(s,t)}. \tag{20}$$

Substituting this back into Equation (18) and using the fact that $\mathbf{N}_{\tilde{\Pi}}$ is Poisson, we find

$$\mathbb{P}[\tilde{X} \in A, \tilde{T} \in dt] = \lim_{s \to t} \left( \frac{\tilde{\mu}([s,t] \times A)}{\tilde{\mu}(s,t)} \cdot \tilde{\mu}(s,t)e^{-\tilde{\mu}(s,t)} \cdot e^{-\tilde{\mu}(s)} \right) \Big/ (t-s) \tag{21}$$

$$= \lim_{s \to t} \left( \tilde{\mu}([s,t] \times A) \cdot e^{-(\tilde{\mu}(t) - \tilde{\mu}(s))} \cdot e^{-\tilde{\mu}(s)} \right) \Big/ (t-s) \tag{22}$$

$$= e^{-\tilde{\mu}(t)} \cdot \lim_{s \to t} \frac{\tilde{\mu}([s,t] \times A)}{t-s} \tag{23}$$

$$= e^{-\tilde{\mu}(t)} \cdot \int_A \mathbf{1}[t \le \varphi(x)] \, dP(x), \tag{24}$$

where the last equation holds by the definition of the derivative and the fundamental theorem of calculus. From this, by inspection we find

$$\frac{d\mathbb{P}[\tilde{T} = t, \tilde{X} = x]}{d(\lambda \times P)} = \mathbf{1}[t \le \varphi(x)]e^{-\tilde{\mu}(t)} = \mathbf{1}[t \le \varphi(x)]\mathbb{P}[\tilde{T} \ge t]. \tag{25}$$

Finally, by marginalizing out the first arrival time, we find that the spatial distribution is

$$\frac{d\mathbb{P}[\tilde{X} = x]}{dP} = \int_0^\infty \frac{d\mathbb{P}[\tilde{T} = t, \tilde{X} = x]}{d(\lambda \times P)} \, dt \tag{26}$$

$$= \int_0^{\varphi(x)} \mathbb{P}[\tilde{T} \ge t] \, dt. \tag{27}$$

**Deriving $\varphi$ by finding $\sigma$:** Note that Equation (27) holds for any $\varphi$. However, to get a correct algorithm, we need to set it such that $\frac{d\mathbb{P}[\tilde{X}=x]}{dP} = \frac{dQ}{dP} = r$. Since we can write $\varphi = \sigma \circ r$ by our earlier argument, this problem is reduced to finding an appropriate invertible continuous function $\sigma : \mathbb{R}^+ \to \mathbb{R}^+$. Thus, we wish to find $\sigma$ such that

$$\forall x \in \Omega, \quad r(x) = \int_0^{\sigma(r(x))} \mathbb{P}[\tilde{T} \ge t] \, dt. \tag{28}$$

Now, introduce $\tau = \sigma(r(x))$, so that $\sigma^{-1}(\tau) = r(x)$. Note that this substitution only makes sense for $\tau \in r(\Omega)$. However, since we are free to extend $\sigma^{-1}$ to $\mathbb{R}^+$ in any we like so long as it is monotone, we may require that this equation hold for all $\tau \in \mathbb{R}^+$. Then, we find

$$\sigma^{-1}(\tau) = \int_0^\tau \mathbb{P}[\tilde{T} \ge t] \, dt \tag{29}$$

$$\Rightarrow \left(\sigma^{-1}\right)'(\tau) = \mathbb{P}[\tilde{T} \ge \tau] \tag{30}$$

with $\sigma^{-1}(0) = 0$. Before we solve for $\sigma$, we define

$$w_P(h) \stackrel{def}{=} \mathbb{P}_{Y \sim P}[r(Y) \geq h] \tag{31}$$

$$w_Q(h) \stackrel{def}{=} \mathbb{P}_{Y \sim Q}[r(Y) \geq h]. \tag{32}$$

Note that $w_P$ and $w_Q$ are supported on $[0, r^*)$, where $r^* \stackrel{def}{=} \text{ess sup}_{x \in \Omega}\{r(x)\} = \exp_2(D_\infty[Q\|P])$. Now, we can rewrite Equation (30) as

$$\left(\sigma^{-1}\right)'(t) = \mathbb{P}[\tilde{T} \geq t] \tag{33}$$

$$= \mathbb{P}[\tilde{T} \geq t, \varphi(\tilde{X}) \geq t] + \underbrace{\mathbb{P}[\tilde{T} \geq t, \varphi(\tilde{X}) < t]}_{=0 \text{ due to mutual exclusivity}} \tag{34}$$

$$= \underbrace{\mathbb{P}[\tilde{T} \geq 0, \varphi(\tilde{X}) \geq t]}_{=\mathbb{P}[\varphi(\tilde{X}) \geq t], \text{ since } \tilde{T} \geq 0 \text{ always}} - \mathbb{P}[\varphi(\tilde{X}) \geq t \geq \tilde{T}] \tag{35}$$

$$= \mathbb{P}[r(\tilde{X}) \geq \sigma^{-1}(t)] - \mathbb{P}[r(\tilde{X}) \geq \sigma^{-1}(t) \geq \sigma^{-1}(\tilde{T})] \tag{36}$$

$$= w_Q(\sigma^{-1}(t)) - \sigma^{-1}(t)w_P(\sigma^{-1}(t)) \tag{37}$$

where the second term in the last equation follows by noting that

$$\mathbb{P}[r(\tilde{X}) \geq \sigma^{-1}(t) \geq \sigma^{-1}(\tilde{T})] = \int_\Omega \int_0^t \underbrace{\mathbf{1}[r(x) \geq \sigma^{-1}(t)]\mathbf{1}[r(x) \geq \sigma^{-1}(\tau)]}_{=\mathbf{1}[r(x) \geq \sigma^{-1}(t)], \text{ since } \tau < t} \mathbb{P}[\tilde{T} \geq \tau] \, d\tau \, dP(x)$$

$$\tag{38}$$

$$= \int_\Omega \mathbf{1}[r(x) \geq \sigma^{-1}(t)] \underbrace{\int_0^t \mathbb{P}[\tilde{T} \geq \tau] \, d\tau}_{=\sigma^{-1}(t), \text{by eq. (29)}} \, dP(x) \tag{39}$$

$$= \sigma^{-1}(t)w_P(\sigma^{-1}(t)). \tag{40}$$

Now, by the inverse function theorem, we get

$$\sigma'(h) = \frac{1}{w_Q(h) - h \cdot w_P(h)}, \tag{41}$$

thus we finally find

$$\sigma(h) = \int_0^h \frac{1}{w_Q(\eta) - \eta \cdot w_P(\eta)} \, d\eta. \tag{42}$$

Thus, to recapitulate, setting $\varphi = \sigma \circ r$, where $\sigma$ is given by Equation (42) will ensure that the spatial distribution of the first arrival of $\Pi$ under $\varphi$ is the target distribution $Q$.

**An alternate form for $\mathbb{P}[\tilde{T} \geq t]$:** We make use of the following integral representation of the indicator function to present an alternative form of the survival function of $\tilde{T}$:

$$\text{for } a > 0: \quad \mathbf{1}[a \leq x] = \int_0^x \delta(y - a) \, dy, \tag{43}$$

where $\delta$ is the Dirac delta function. Then, we have

$$w_Q(h) = \int_\Omega \mathbf{1}[r(x) \geq h] \, dQ(x) \tag{44}$$

$$= 1 - \int_\Omega \int_0^h r(x)\delta(y - r(x)) \, dy \, dP(x) \tag{45}$$

$$= 1 - \int_0^h y \int_\Omega \delta(y - r(x)) \, dP(x) \, dy \tag{46}$$

$$= 1 - [y(1 - w_P(y))]_0^h + \int_0^h (1 - w_P(y)) \, dy \tag{47}$$

$$= 1 + h \cdot w_P(h) - \int_0^h w_P(y) \, dy. \tag{48}$$

Then, from this identity, we find that

$$\mathbb{P}[\tilde{T} \geq t] = w_Q(\sigma^{-1}(t)) - \sigma^{-1}(t)w_P(\sigma^{-1}(t)) \tag{49}$$

$$= 1 - \int_0^{\sigma^{-1}(t)} w_P(y)\,dy. \tag{50}$$

**Well-definition of GPRS and Algorithm 3 terminates with probability** 1**:** To ensure that our construction is useful, we need to show that the first arrival under the graph $\tilde{T}$ exists and is almost surely finite, so that Algorithm 3 terminates with probability 1. First, note that for any $t > 0$, we have $\tilde{\mu}(t) \leq \mu(t) = t < \infty$. Since $\mathbb{P}[\mathbf{N}_{\tilde{\Pi}}(t) = 0] = e^{-\tilde{\mu}(t)}$, this implies that for any finite time $t$, $\mathbb{P}[\mathbf{N}_{\tilde{\Pi}}(t) = 0] > 0$. Second, note that for $h \in [0, r^*]$,

$$w_Q(h) - h \cdot w_P(h) = \int_\Omega \mathbf{1}[r(x) \geq h](r(x) - h)\,dP(x), \tag{51}$$

from which

$$w_Q(r^*) - r^* w_P(r^*) = \int_\Omega \mathbf{1}[r(x) \geq r^*](r(x) - r^*)\,dP(x) \tag{52}$$

$$= \int_\Omega \mathbf{1}[r(x) = r^*](r(x) - r^*)\,dP(x) \tag{53}$$

$$= 0. \tag{54}$$

Thus, in particular, we find that

$$\lim_{h \to r^*} e^{-\tilde{\mu}(\sigma(h))} = \lim_{h \to r^*} \mathbb{P}[\mathbf{N}_{\tilde{\Pi}}(\sigma(h)) = 0] \stackrel{\text{eq. (37)}}{=} \lim_{h \to r^*} (w_Q(h) - h \cdot w_P(h)) = 0. \tag{55}$$

By continuity, this can only hold if $\tilde{\mu}(\sigma(h)) \to \infty$ as $h \to r^*$. Since $\tilde{\mu}(t)$ is finite for all $t > 0$ and $\sigma$ is monotonically increasing, this also implies that $\sigma(h) \to \infty$ as $h \to r^*$. Thus, we have shown a couple of facts:

- $\sigma(h) \to \infty$ as $h \to r^*$, hence $\varphi$ is always unbounded at points in $\Omega$ that achieve the supremum $r^*$. Furthermore, this implies that

$$\sigma^{-1}(t) \to r^* \quad \text{as} \quad t \to \infty. \tag{56}$$

  Since $\sigma^{-1}$ is increasing, this implies that it is bounded from above by $r^*$.

- $\tilde{\mu}(t) < \infty$ for all $t > 0$, but $\tilde{\mu}(t) \to \infty$ as $t \to \infty$. Hence, by Equation (14) we have $\mathbb{P}[\mathbf{N}_{\tilde{\Pi}}(\infty) > 0] = 1$. Thus, the first arrival of $\tilde{\Pi}$ exists almost surely, and our construction is well-defined. In particular, it is meaningful to write $\mathbb{P}[\tilde{T} \geq t]$.

- $\mathbb{P}[\tilde{T} \geq t] = \mathbb{P}[\mathbf{N}_{\tilde{\Pi}}(t) = 0] \to 0$ as $t \to \infty$, which shows that the first arrival time is finite with probability 1. In turn, this implies that Algorithm 3 will terminate with probability one, as desired.

Note, that $w_P$ and $w_Q$ can be computed in many practically relevant cases, see Appendix G.

## B   Analysis of Greedy Poisson Rejection Sampling

Now that we constructed a correct sampling algorithm in Appendix A, we turn our attention to deriving the expected first arrival time $\tilde{T}$ in the restricted process $\tilde{\Pi}$, the expected number of samples $N$ before Algorithm 3 terminates and bound on $N$'s variance, and an upper bound on its coding cost. The proofs below showcase the true advantage of formulating the sampling algorithm using the language of Poisson processes: the proofs are all quite short and elegant.

### B.1   The Expected First Arrival Time

At the end of the previous section, we showed that $\tilde{T}$ is finite with probability one, which implies that $\mathbb{E}[\tilde{T}] < \infty$. Now, we derive the value of this expectation exactly. Since $\tilde{T}$ is a positive random variable, by the Darth Vader rule (Muldowney et al., 2012), we may write its expectation as

$$\mathbb{E}\left[\tilde{T}\right] = \int_0^\infty \mathbb{P}[\tilde{T} \geq t]\,dt \stackrel{\text{eq. (29)}}{=} \lim_{t \to \infty} \sigma^{-1}(t) \stackrel{\text{eq. (56)}}{=} r^*. \tag{57}$$

## B.2 The Expectation and Variance of the Runtime

**Expectation of $N$:** First, let $\tilde{\Pi}^C \stackrel{def}{=} \Pi \setminus \tilde{\Pi}$ be the set of points in $\Pi$ above the graph of $\varphi$. Since $\tilde{\Pi}$ and $\tilde{\Pi}^C$ are defined on complementary sets, they are independent. Since by definition $\tilde{T}$ is the first arrival of $\tilde{\Pi}$ and the $N$th arrival of $\Pi$, it must mean that the first $N-1$ arrivals of $\Pi$ occurred in $\tilde{\Pi}^C$. Thus, conditioned on $\tilde{T}$, we have $N - 1 = |\{(T, X) \in \tilde{\Pi}^C \mid T < \tilde{T}\}|$ is Poisson distributed with mean

$$\mathbb{E}\left[N - 1 \mid \tilde{T}\right] = \int_0^{\tilde{T}} \int_\Omega \mathbf{1}[t \geq \varphi(x)] \, dP(x) \, dt \tag{58}$$

$$= \int_0^{\tilde{T}} \int_\Omega 1 - \mathbf{1}[t < \varphi(x)] \, dP(x) \, dt \tag{59}$$

$$= \tilde{T} - \tilde{\mu}\left(\tilde{T}\right). \tag{60}$$

By the law of iterated expectations,

$$\mathbb{E}[N - 1] = \mathbb{E}_{\tilde{T}}\left[\mathbb{E}\left[N - 1 \mid \tilde{T}\right]\right] \tag{61}$$

$$= \mathbb{E}\left[\tilde{T}\right] - \mathbb{E}\left[\tilde{\mu}\left(\tilde{T}\right)\right]. \tag{62}$$

Focusing on the second term, we find

$$\mathbb{E}\left[\tilde{\mu}\left(\tilde{T}\right)\right] = \int_0^\infty \tilde{\mu}(t) \cdot \mathbb{P}[\tilde{T} \in dt] \, dt \tag{63}$$

$$= \int_0^\infty \tilde{\mu}(t) \cdot \tilde{\mu}'(t) e^{-\tilde{\mu}(t)} \, dt \tag{64}$$

$$= \int_0^\infty u e^{-u} \, du = 1, \tag{65}$$

where the third equality follows by substituting $u = \tilde{\mu}(t)$. Finally, plugging the above and Equation (57) into Equation (62), we find

$$\mathbb{E}[N] = 1 + \mathbb{E}[N - 1] = 1 + r^* - 1 = r^*. \tag{66}$$

**Variance of $N$:** We now show that the distribution of $N$ is *super-Poissonian*, i.e. $\mathbb{E}[N] \leq \mathbb{V}[N]$. Similarly to the above, we begin with the law of iterated variances to find

$$\mathbb{V}[N] = \mathbb{V}[N - 1] = \mathbb{E}_{\tilde{T}}[\mathbb{V}[N - 1 \mid \tilde{T}]] + \mathbb{V}_{\tilde{T}}[\mathbb{E}[N - 1 \mid \tilde{T}]] \tag{67}$$

$$= \mathbb{E}\left[\tilde{T}\right] - \mathbb{E}\left[\tilde{\mu}\left(\tilde{T}\right)\right] + \mathbb{V}\left[\tilde{T}\right] + \mathbb{V}\left[\tilde{\mu}\left(\tilde{T}\right)\right], \tag{68}$$

where the second equality follows from the fact that the variance of $N - 1$ matches its mean conditioned on $\tilde{T}$, since it is a Poisson random variable. Focussing on the last term, we find

$$\mathbb{V}\left[\tilde{\mu}\left(\tilde{T}\right)\right] = \mathbb{E}\left[\tilde{\mu}\left(\tilde{T}\right)^2\right] - \mathbb{E}\left[\tilde{\mu}\left(\tilde{T}\right)\right]^2 \tag{69}$$

$$= \int_0^\infty \tilde{\mu}(t)^2 \cdot \mathbb{P}[\tilde{T} \in dt] \, dt - 1 \tag{70}$$

$$= \int_0^\infty u^2 e^{-u} \, dt - 1 = 1, \tag{71}$$

where the third equality follows from a similar $u$-substitution as in Equation (65). Thus, putting everything together, we find

$$\mathbb{V}[N] = r^* - 1 + \mathbb{V}\left[\tilde{T}\right] + 1 = r^* + \mathbb{V}\left[\tilde{T}\right] > \mathbb{E}[N]. \tag{72}$$

## B.3 The Fractional Moments of the Index

In this section, we follow the work of Liu & Verdu (2018) and analyse the fractional moments of the index $N$ and prove Theorem 3.2. As before, set $Q$ and $P$ as GPRS's target and proposal distribution, respectively, and let $r = \frac{dQ}{dP}$. To begin, we define the Rényi $\alpha$-divergence between two distributions for $\alpha \in (0,1) \cup (1, \infty)$ as

$$D_\alpha[Q\|P] = \frac{1}{\alpha - 1} \log_2 \mathbb{E}_{Z \sim Q} \left[ r(Z)^{\alpha - 1} \right]. \tag{73}$$

By taking limits, we can extend the definiton to $\alpha = 1, \infty$ as

$$D_1[Q\|P] \stackrel{def}{=} \lim_{\alpha \to 1} D_\alpha[Q\|P] = D_{\mathrm{KL}}[Q\|P] \tag{74}$$

$$D_\infty[Q\|P] \stackrel{def}{=} \lim_{\alpha \to \infty} D_\alpha[Q\|P] = \operatorname*{ess\,sup}_{x \in \Omega} \left\{ \log_2 r(x) \right\}. \tag{75}$$

Now fix $\alpha \in (0,1)$ and $x \in \Omega$ and note that by Jensen's inequality,

$$\mathbb{E}[N^\alpha \mid \tilde{X} = x] \leq \mathbb{E}[N \mid \tilde{X} = x]^\alpha. \tag{76}$$

Now, to continue, we will show the following bound:

$$\mathbb{E}[N \mid \tilde{X} = x] \leq \frac{1}{w_P(r(x))} \tag{77}$$

Let $\Pi$ be the $(1, P)$-Poisson process realized by Algorithm 3, and let

$$M = \min\{m \in \mathbb{N} \mid (T_m, X_m) \in \Pi, r(X_m) \geq r(x)\}, \tag{78}$$

i.e. the index of the first arrival in $\Pi$ such that the density ratio of the spatial coordinate exceeds $r(x)$. Since $X_i \sim P$ are i.i.d.,

$$M \sim \mathrm{Geom}\left( \mathbb{P}_{Z \sim P}[r(Z) \geq r(x)] \right), \tag{79}$$

hence

$$\mathbb{E}[M] = \frac{1}{\mathbb{P}_{Z \sim P}[r(Z) \geq r(x)]} = \frac{1}{w_P(r(x))}. \tag{80}$$

Now, since geometrically distributed random variables are memoryless, we find

$$\mathbb{E}[M] = \mathbb{E}[M - N \mid M > N] \tag{81}$$
$$\leq \mathbb{E}[M \mid M > N], \tag{82}$$

which then implies

$$\mathbb{E}[M \mid M \leq N] \leq \mathbb{E}[M]. \tag{83}$$

Before we proceed, we first require the following result

**Lemma B.1.** *Let all quantities be as defined above and let $x, y \in \Omega$ such that $r(x) \leq r(y)$. Then, $N \mid \tilde{X} = y$ stochastically dominates $N \mid \tilde{X} = x$, i.e. for all positive integers $n$ we have*

$$\mathbb{P}[N \geq n \mid \tilde{X} = x] \leq \mathbb{P}[N \geq n \mid \tilde{X} = y]. \tag{84}$$

*Proof.* The proof follows via direct calculation. Concretely,

$$\mathbb{P}[N \leq n \mid \tilde{X} = x] = \mathbb{P}[N - 1 \leq n - 1 \mid \tilde{X} = x] \tag{85}$$

$$= \mathbb{E}_{\tilde{T} \sim P_{\tilde{T}|\tilde{X}=x}} \left[ \mathbb{P}[N - 1 \leq n - 1 \mid \tilde{X} = x, \tilde{T}] \right] \tag{86}$$

$$= \frac{1}{(n-1)!} \mathbb{E}_{\tilde{T} \sim P_{\tilde{T}|\tilde{X}=x}} \left[ \Gamma(n, \tilde{T} - \tilde{\mu}(\tilde{T})) \right], \tag{87}$$

where $\Gamma(s, x)$ is the upper incomplete $\Gamma$-function defined as

$$\Gamma(s, x) \stackrel{def}{=} \int_x^\infty t^{s-1} e^{-t} \, dt. \tag{88}$$

Equation (87) follows from the fact that conditioned on $\tilde{T}$ we have that $N - 1$ is a Poisson random variable with mean $\tilde{T} - \tilde{\mu}(\tilde{T})$. Now, focussing on the expectation term in Equation (87), we let

$$\varepsilon(r(x)) \stackrel{def}{=} \mathbb{E}_{\tilde{T} \sim P_{\tilde{T}|\tilde{X}=x}} \left[ \Gamma(n, \tilde{T} - \tilde{\mu}(\tilde{T})) \right] = \frac{1}{r(x)} \int_0^{\sigma(r(x))} \Gamma(n, t - \tilde{\mu}(t)) \left( \sigma^{-1} \right)'(t) \, dt. \quad (89)$$

Substituting $h = r(x)$, we now show that $\epsilon$ is a decreasing function, which proves the lemma. To this end, regardless of the image of the density ratio $r(\Omega)$, extend the definition of $\epsilon$ to the entirety of $(0, \mathrm{ess\,sup}\, r(x))$. Now, observe, that

$$\frac{d\varepsilon}{dh} = \frac{1}{h^2} \left( h \cdot \Gamma(n, \sigma(h) - \tilde{\mu}(\sigma(h))) \cdot \overbrace{\left( \sigma^{-1} \right)'(\sigma(h)) \cdot \sigma'(h)}^{=1 \text{ by inverse function theorem}} - \int_0^{\sigma(h)} \Gamma(n, t - \tilde{\mu}(t)) \left( \sigma^{-1} \right)'(t) \, dt \right). \quad (90)$$

Focussing on the integral, via integration by parts we find

$$\int_0^{\sigma(h)} \Gamma(n, t - \tilde{\mu}(t)) \left( \sigma^{-1} \right)'(t) \, dt = \quad (91)$$

$$h \cdot \Gamma(n, \sigma(h) - \tilde{\mu}(\sigma(h))) + \int_0^{\sigma(h)} (t - \tilde{\mu}(t))^{(n-1)} e^{-(t-\tilde{\mu}(t))} (1 - w_P(\sigma^{-1}(t))) \sigma^{-1}(t) \, dt. \quad (92)$$

Plugging this back into the above calculation, we get

$$\frac{d\varepsilon}{dh} = -\frac{1}{h^2} \int_0^{\sigma(h)} (t - \tilde{\mu}(t))^{(n-1)} e^{-(t-\tilde{\mu}(t))} (1 - w_P(\sigma^{-1}(t))) \sigma^{-1}(t) \, dt. \quad (93)$$

Note, that since the integrand is positive and $h$ was arbitrary, $\frac{d\epsilon}{dh}$ is negative for every $h$ in its domain, which proves the lemma. $\qquad \square$

Returning to Equation (83), we now examine the two cases $M < N$ and $M = N$ separately.

**Case $M < N$:** In this case, we have the following inequalities:

- $r(x) \leq r(X_M)$ by the definition of $M$. Applying $\sigma$ to both sides we find $\varphi(x) \leq \varphi(X_M)$.
- $\varphi(X_M) < T_M$, since $M < N$ and by definition $N$ is the index of the first arrival of $\Pi$ under the graph of $\varphi$.

Now, note that since $M < N$ conditioned on $T_M$, the first $M - 1$ points of $\Pi$ are all rejected and hence are all in $\tilde{\Pi}^C = \Pi \setminus \tilde{\Pi}$. Thus, $M - 1 \mid (M < N, T_M)$ is Poisson distributed with mean $T_M - \tilde{\mu}(T_M)$. Hence,

$$\mathbb{E}[M \mid M < N] = 1 + \mathbb{E}[M - 1 \mid M < N] \quad (94)$$

$$= 1 + \mathbb{E}_{T_M \sim P_{T_M|M<N}} \left[ \mathbb{E}[M - 1 \mid M < N, T_M] \right] \quad (95)$$

$$= 1 + \mathbb{E}_{T_M \sim P_{T_M|M<N}} \left[ T_M - \tilde{\mu}(T_M) \right] \quad (96)$$

$$\geq 1 + \mathbb{E}_{T_M \sim P_{T_M|M<N}} \left[ \varphi(x) - \tilde{\mu}(\varphi(x)) \right] \quad (97)$$

$$= 1 + \varphi(x) - \tilde{\mu}(\varphi(x)). \quad (98)$$

On the other hand, since the $N$-th point is the index of first arrival of $\Pi$ under the graph of $\varphi$, hence $\tilde{T} \leq \varphi(\tilde{X})$. Furthermore, conditioning on $\tilde{X} = x$ we get $\tilde{T} \leq \varphi(x)$. Similarly to the above case $N - 1 \mid \tilde{T}$ is Poisson distributed with mean $\tilde{T} - \tilde{\mu}(\tilde{T})$. Hence, we get

$$\mathbb{E}[N \mid \tilde{X} = x] = 1 + \mathbb{E}_{\tilde{T}|\tilde{X}=x} \left[ \mathbb{E}[N - 1 \mid \tilde{X} = x, \tilde{T}] \right] \quad (99)$$

$$= 1 + \mathbb{E}_{\tilde{T}|\tilde{X}=x} \left[ \tilde{T} - \tilde{\mu}(\tilde{T}) \right] \quad (100)$$

$$\geq 1 + \mathbb{E}_{\tilde{T}|\tilde{X}=x} \left[ \varphi(x) - \tilde{\mu}(\varphi(x)) \right] \quad (101)$$

$$= 1 + \varphi(x) - \tilde{\mu}(\varphi(x)). \quad (102)$$

Putting the above two inequalities, we find

$$\mathbb{E}[N \mid \tilde{X} = x] \leq 1 + \varphi(x) - \tilde{\mu}(\varphi(x)) \leq \mathbb{E}[M \mid M < N]. \tag{103}$$

**Case $M = N$:** In this case, we have

$$\mathbb{E}[M \mid N = M] = \mathbb{E}[N \mid r(X_N) \geq r(x)] \tag{104}$$
$$\geq \mathbb{E}[N \mid X_N = \tilde{X} = x], \tag{105}$$

where Equation (105) follows by applying Lemma B.1.

Finally, putting these two cases together we find

$$\frac{1}{w_P(r(x))} = \mathbb{E}[M] \tag{106}$$
$$\geq \mathbb{E}[M \mid N \geq M] \tag{107}$$
$$\geq \mathbb{E}[N \mid \tilde{X} = x], \tag{108}$$

which establishes Equation (77). Now, taking expectation over $\tilde{X}$, we get

$$\mathbb{E}[N^\alpha] \leq \int_\Omega \frac{dQ(x)}{w_P(r(x))^\alpha} \tag{109}$$
$$= \int_\Omega \frac{r(x)}{w_P(r(x))^\alpha} \, dP(x) \tag{110}$$
$$= \int_0^\infty \frac{y}{w_P(y)^\alpha} \int_\Omega \delta(y - r(x)) \, dP(x) \, dy \tag{111}$$
$$= \int_0^\infty \frac{y}{w_P(y)^\alpha} \int_\Omega \delta(y - r(x)) \, dP(x) \, dy \tag{112}$$
$$= \left[ -\frac{y \cdot w_P(y)^{1-\alpha}}{1-\alpha} \right]_0^\infty + \frac{1}{1-\alpha} \int_0^\infty w_P(y)^{1-\alpha} \, dy \tag{113}$$
$$= \frac{1}{1-\alpha} \int_0^\infty w_P(y)^{1-\alpha} \, dy. \tag{114}$$

In the above, Equation (113) follows by integration by parts and from the fact that

$$w_P(h) = \int_\Omega \mathbf{1}[r(x) \geq h] \, dP(x) \tag{115}$$
$$= 1 - \int_0^h \int_\Omega \delta(\eta - r(x)) \, dP(x) \, d\eta. \tag{116}$$

We now note that we can bound $w_P$ using Markov's inequality:

$$w_P(h) = \mathbb{P}_{Z \sim P}[r(Z) \geq h] \tag{117}$$
$$= \mathbb{P}_{Z \sim P}\left[ r(Z)^{\frac{1}{1-\alpha}} \geq h^{\frac{1}{1-\alpha}} \right] \tag{118}$$
$$\leq h^{-\frac{1}{1-\alpha}} \cdot \mathbb{E}_{Z \sim P}\left[ r(Z)^{\frac{1}{1-\alpha}} \right] \tag{119}$$
$$= h^{-\frac{1}{1-\alpha}} \cdot \mathbb{E}_{Z \sim Q}\left[ r(Z)^{\frac{1}{1-\alpha} - 1} \right] \tag{120}$$
$$= h^{-\frac{1}{1-\alpha}} \cdot \exp_2\left( \frac{\alpha}{1-\alpha} D_{\frac{1}{1-\alpha}}[Q\|P] \right). \tag{121}$$

Putting the above in Equation (114), we get

$$\mathbb{E}[N^\alpha] \leq \frac{1}{1-\alpha} \int_0^\infty w_P(y)^{1-\alpha}\, dy \tag{122}$$

$$\leq \frac{1}{1-\alpha} \left( 1 + \int_1^{\exp_2(D_\infty[Q\|P])} w_P(y)^{1-\alpha}\, dy \right) \tag{123}$$

$$\leq \frac{1}{1-\alpha} \left( 1 + \int_1^{\exp_2(D_\infty[Q\|P])} \frac{1}{y}\, dy \cdot \exp_2\left( \alpha D_{\frac{1}{1-\alpha}}[Q\|P] \right) \right) \tag{124}$$

$$\leq \frac{1}{1-\alpha} \left( 1 + \frac{D_\infty[Q\|P]}{\log_2 e} \right) \cdot \exp_2\left( \alpha D_{\frac{1}{1-\alpha}}[Q\|P] \right). \tag{125}$$

## B.4 The Codelength of the Index

As discussed in Section 3, Alice and Bob can realize a one-shot channel simulation protocol using GPRS for a pair of correlated random variables $\mathbf{x}, \mathbf{y} \sim P_{\mathbf{x},\mathbf{y}}$ if they have access to shared randomness and can simulate samples from $P_{\mathbf{x}}$. In particular, after receiving $\mathbf{y} \sim P_{\mathbf{y}}$, Alice runs GPRS with proposal distribution $P_{\mathbf{x}}$ and target $P_{\mathbf{x}|\mathbf{y}}$, and the shared randomness to simulate the Poisson process $\Pi$. She then encodes the index $N$ of the first arrival of $\Pi$ under the graph of $\varphi$. Bob can decode Alice's sample $\mathbf{x} \sim P_{\mathbf{x}|\mathbf{y}}$ by simulating the same $N$ samples from $\Pi$ using the shared randomness. The question is, how efficiently can Alice encode $N$? We answer this question by following the approach of Li & El Gamal (2018). Namely, we first bound the conditional expectation $\mathbb{E}[\log_2 N \mid \mathbf{y} = y]$, after which we average over $\mathbf{y}$ to bound $\mathbb{E}[\log_2 N]$. Then, we use the maximum entropy distribution subject to the constraint of fixed $\mathbb{E}[\log_2 N]$ to bound $\mathbb{H}[N]$. Finally, noting that $\mathbf{x}$ is a function of $\Pi$ and $N$, we get

$$\mathbb{H}[\mathbf{x}, N \mid \Pi] = \underbrace{\mathbb{H}[\mathbf{x} \mid N, \Pi]}_{=0} + \mathbb{H}[N \mid \Pi] \tag{126}$$

$$= \underbrace{\mathbb{H}[N \mid \mathbf{x}, \Pi]}_{=0} + \mathbb{H}[\mathbf{x} \mid \Pi], \tag{127}$$

from which

$$\mathbb{H}[\mathbf{x} \mid \Pi] = \mathbb{H}[N \mid \Pi] \leq \mathbb{H}[N], \tag{128}$$

which will finish the proof.

**Bound on the conditional expectation:** Fix $\mathbf{y}$ and set $Q = P_{\mathbf{x}|\mathbf{y}}$ and $P = P_{\mathbf{x}}$ as GPRS's target and proposal distribution, respectively. Let $(t, x) \in \mathbb{R}^+ \times \Omega$ be a point under the graph of $\varphi$, i.e. $\sigma^{-1}(t) \leq r(x)$. Then,

$$r(x) \geq \sigma^{-1}(t) \tag{129}$$

$$\overset{\text{eq. (30)}}{=} \int_0^t \mathbb{P}[\tilde{T} \geq \tau]\, d\tau \tag{130}$$

$$\geq t \cdot \inf_{\tau \in [0,t]} \left\{ \mathbb{P}[\tilde{T} \geq \tau] \right\} \tag{131}$$

$$= t \cdot \mathbb{P}[\tilde{T} \geq t]. \tag{132}$$

From this, we get

$$t + 1 \leq \frac{r(x)}{\mathbb{P}[\tilde{T} \geq t]} + 1 \leq \frac{r(x) + 1}{\mathbb{P}[\tilde{T} \geq t]}. \tag{133}$$

Next, conditioning on the first arrival $(\tilde{T}, \tilde{X})$ we get

$$\mathbb{E}[\log_2 N \mid \tilde{T} = t, \tilde{X} = x] \leq \log_2 \left( \mathbb{E}[N - 1 \mid \tilde{T} = t, \tilde{X} = x] + 1 \right) \tag{134}$$

$$\overset{\text{eq. (60)}}{=} \log_2 \left( t - \tilde{\mu}(t) + 1 \right) \tag{135}$$

$$\leq \log_2 \left( t + 1 \right) \tag{136}$$

$$\overset{\text{eq. (133)}}{\leq} \log_2 \left( r(x) + 1 \right) - \log_2 \mathbb{P}[\tilde{T} \geq t] \tag{137}$$

$$= \log_2 \left( r(x) + 1 \right) + \tilde{\mu}(t) \cdot \log_2 e, \tag{138}$$

where the first inequality follows by Jensen's inequality.

Now, by the law of iterated expectations, we find

$$\mathbb{E}[\log_2 N] = \mathbb{E}_{\tilde{X}}\left[\mathbb{E}_{\tilde{T}|\tilde{X}}\left[\mathbb{E}[\log_2 N \mid \tilde{T}, \tilde{X}]\right]\right] \tag{139}$$

$$\overset{\text{eq. (138)}}{\leq} \mathbb{E}_{\tilde{X}}\left[\log_2\left(r\left(\tilde{X}\right)+1\right)\right] + \underbrace{\mathbb{E}_{\tilde{T}}\left[\tilde{\mu}\left(\tilde{T}\right)\right]}_{=1 \text{ by eq. (65)}}\cdot\log_2 e \tag{140}$$

$$= D_{\mathrm{KL}}[Q\|P] + \mathbb{E}_{\tilde{X}}\left[\log_2\left(1+\frac{1}{r\left(\tilde{X}\right)}\right)\right] + \log_2 e \tag{141}$$

$$\leq D_{\mathrm{KL}}[Q\|P] + \mathbb{E}_{\tilde{X}}\left[\log_e\left(e^{\frac{1}{r(\tilde{X})}}\right)\right]\cdot\log_2 e + \log_2 e \tag{142}$$

$$= D_{\mathrm{KL}}[Q\|P] + 2\cdot\log_2 e, \tag{143}$$

where the second inequality follows from switching to the natural base and using a first-order Taylor approximation for $e^{\frac{1}{x}}$.

**Bound on the marginal expectation and entropy:** Equation (143) is a one-shot bound, which yields

$$\mathbb{E}[\log_2 N \mid \mathbf{y}] \leq D_{\mathrm{KL}}[P_{\mathbf{x}|\mathbf{y}}\|P_{\mathbf{x}}] + 2\cdot\log_2 e. \tag{144}$$

Taking expectation over $\mathbf{y}\sim P_{\mathbf{y}}$, we get

$$\mathbb{E}[\log_2 N] \leq I[\mathbf{x};\mathbf{y}] + 2\cdot\log_2 e. \tag{145}$$

Finally, following Proposition 4 in Li & El Gamal (2018), the maximum entropy distribution for $N$ subject to a constraint on $\mathbb{E}[\log_2 N]$ obeys

$$\mathbb{H}[N] \leq \mathbb{E}[\log_2 N] + \log\left(\mathbb{E}[\log_2 N] + 1\right) + 1. \tag{146}$$

Plugging in Equation (145) into the above, we get

$$\mathbb{H}[N] \leq I[\mathbf{x};\mathbf{y}] + \log_2(I[\mathbf{x};\mathbf{y}] + 2\log_2 e + 1) + 2\log_2 e + 1 \tag{147}$$

$$\leq I[\mathbf{x};\mathbf{y}] + \log_2(I[\mathbf{x};\mathbf{y}] + 1) + 2\log_2 e + 1 + \log_2(2\log_2 e + 1) \tag{148}$$

$$\approx I[\mathbf{x};\mathbf{y}] + \log_2(I[\mathbf{x};\mathbf{y}] + 1) + 5.84 \tag{149}$$

$$< I[\mathbf{x};\mathbf{y}] + \log_2(I[\mathbf{x};\mathbf{y}] + 1) + 6. \tag{150}$$

Similarly to Li & El Gamal (2018), we can encode $N$ using a Zeta distribution $\zeta(n \mid s) \propto n^{-s}$ with

$$s \overset{def}{=} \frac{1}{I[\mathbf{x};\mathbf{y}] + 2\log_2 e}. \tag{151}$$

With this choice of the coding distribution, the expected codelength of $N$ is upper bounded by $I[\mathbf{x};\mathbf{y}] + \log_2(I[\mathbf{x};\mathbf{y}] + 1) + 7$ bits.

## C  Analysis of Parallel GPRS

Parallel GPRS (PGPRS) is a general technique to accelerate GPRS when parallel computing power is available and is presented in Section 3.1. Assuming that $J$ parallel threads are available, for target distribution $Q$ and proposal $P$, PGPRS simulates $J$ Poisson processes $\Pi_1,\ldots,\Pi_J$ in parallel, all of them with mean measure $\mu_i \overset{def}{=} \frac{\lambda}{J}\times P$. Assuming $\left\{\left(T_{N_1}^{(1)}, X_{N_1}^{(1)}\right),\ldots,\left(T_{N_J}^{(J)}, X_{N_J}^{(J)}\right)\right\}$ are the first arrivals of $\Pi_1,\ldots,\Pi_J$ under $\varphi$, respectively, PGPRS selects the arrival with the overall smallest arrival time $J^* = \arg\min_{j\in\{1,\ldots,J\}}\left\{T_{N_j}^{(j)}\right\}$. By the superposition theorem (Kingman, 1992), $\cup_{j=1}^J\Pi_j = \Pi$ is a Poisson process with mean measure $\mu = \lambda\times P$, hence $\left(T_{N_{J^*}}^{(J^*)}, X_{N_{J^*}}^{(J^*)}\right)$ will be the first arrival of $\Pi$ under the graph of $\varphi$ and the theoretic construction in Appendix A therefore guarantees the correctness of PGPRS.

**Expected runtime:** We will proceed similarly to the analysis in Appendix B.2. Concretely, let $\tilde{T} = T_{N_{J^*}}^{(J^*)}$ be the first arrival time of $\Pi$ under the graph of $\varphi$. Let

$$\nu_j \stackrel{def}{=} \left| \left\{ (T, X) \in \Pi_j \mid T < \tilde{T}, T > \varphi(X) \right\} \right| \tag{152}$$

be the number of points in $\Pi_j$ that are rejected before the global first arrival occurs. By an independence argument analogous to the one given in Appendix B.2, the $\nu_1, \ldots, \nu_J$ are independent Poisson random variables, each distributed with mean

$$\mathbb{E}\left[ \nu_j \mid \tilde{T} \right] = \frac{1}{J} \left( \tilde{T} - \tilde{\mu}\left( \tilde{T} \right) \right). \tag{153}$$

Hence, by the law of iterated expectations, we find that we get

$$\mathbb{E}\left[ \nu_j \right] \stackrel{\text{eq. (62)}}{=} \frac{r^* - 1}{J} \tag{154}$$

rejections in each of the $J$ threads on average before the global first arrival occurs. Now, a thread $j$ terminates either when the global first arrival occurs in it or when the thread's current arrival time $T_{n_j}^{(j)}$ provably exceeds the global first arrival time. Thus, on average, each thread will have one more rejection compared to Equation (154). Hence the average runtime of a thread is

$$\mathbb{E}[\nu_j + 1] = \frac{r^* - 1}{J} + 1, \tag{155}$$

and the average number of samples simulated by PGPRS across all its threads is

$$J \cdot \left( \frac{r^* - 1}{J} + 1 \right) = r^* + J - 1. \tag{156}$$

**Variance of Runtime:** Once again, similarly to Appendix B.2, we find by the law of iterated variances that the variance of the runtime in each of the $j$ threads is

$$\mathbb{V}[\nu_j + 1] = \mathbb{V}[\nu_j] = \frac{r^* - 1}{J} + \frac{1}{J^2} \left( \mathbb{V}\left[ \tilde{T} \right] + 1 \right), \tag{157}$$

meaning that we make an $\mathcal{O}(1/J)$ reduction in the variance of the runtime compared to regular GPRS.

**Codelength:** PGPRS can also realize a one-shot channel simulation protocol for a pair of correlated random variables $\mathbf{x}, \mathbf{y} \sim P_{\mathbf{x},\mathbf{y}}$. For a fixed $\mathbf{y} \sim P_{\mathbf{y}}$, Alice applies PGPRS to the target $Q = P_{\mathbf{x}|\mathbf{y}}$ and proposal $P = P_{\mathbf{x}}$, and encodes the two-part code $(J^*, N_{J^*})$. Bob can then simulate $N_{J^*}$ samples from $\Pi_{J^*}$ and recover Alice's sample.

**Encoding $J^*$:** By the symmetry of the setup, the global first arrival will occur with equal probability in each subprocess $\Pi_j$. Hence $J^*$ follows a uniform distribution on $\{1, \ldots J\}$. Therefore, Alice can encode $J^*$ optimally using $\lceil \log_2 J \rceil$ bits.

**Encoding $N_{J^*}$:** We can develop a bound using an almost identical argument to the one in Appendix B.4. In particular, by adapting the conditional bound in Equation (138) appropriately using Equation (155), we get

$$\mathbb{E}\left[ \log_2 N_{J^*} \mid \tilde{T} = t, \tilde{X} = x, J^* \right] \le \log_2(r(x) + 1) + \tilde{\mu}(t) \cdot \log_2 e - \log_2 J. \tag{158}$$

Then, using this conditional bound and adapting Equation (143), we find

$$\mathbb{E}\left[ \log_2 N_{J^*} \mid \mathbf{y}, J^* \right] \le D_{\mathrm{KL}}[P_{\mathbf{x}|\mathbf{y}} \| P_{\mathbf{x}}] + 2 \cdot \log_2 e - \log_2 J \tag{159}$$

to obtain a one-shot bound. Taking expectation over $\mathbf{y} \sim P_{\mathbf{y}}$, we get

$$\mathbb{E}\left[ \log_2 N_{J^*} \mid J^* \right] \le I[\mathbf{x}; \mathbf{y}] + 2 \cdot \log_2 e - \log_2 J, \tag{160}$$

hence, by adapting the maximum entropy bound in Equation (150), we find

$$\mathbb{H}[N_{J^*} \mid J^*] < I[\mathbf{x}; \mathbf{y}] - \log_2 J + \log_2(I[\mathbf{x}; \mathbf{y}] - \log_2 J + 1) + 6. \tag{161}$$

Thus, we finally find that the entropy of the two-part code $(J^*, N_{J^*})$ is upper bounded by

$$\mathbb{H}[J^*, N_{J^*}] < I[\mathbf{x}; \mathbf{y}] + \log_2(I[\mathbf{x}; \mathbf{y}] - \log_2 J + 1) + 7. \tag{162}$$

Using a Zeta distribution $\zeta(n \mid s) \propto n^{-s}$ to encode $N_{J^*} \mid J^*$ with

$$s \stackrel{def}{=} \frac{1}{I[\mathbf{x}; \mathbf{y}] + 2 \cdot \log_2 e - \log_2 J}, \tag{163}$$

we find that the expected codelength of the two-part code is upper bounded by

$$\mathbb{H}[J^*, N_{J^*}] < I[\mathbf{x}; \mathbf{y}] + \log_2(I[\mathbf{x}; \mathbf{y}] - \log_2 J + 1) + 8 \text{ bits.} \tag{164}$$

# D  Simulating Poisson Processes Using Tree-Structured Partitions

---

**Algorithm 7:** Simulating a tree construction with another

---

**Input** : Proposal distribution $P$
  Target splitting function $\texttt{split}_{\text{target}}$,
  Simulating splitting function $\texttt{split}_{\text{sim}}$,
  Target heap index $H_{\text{target}}$
**Output** : Arrival $(T, X)$ with heap index $H_{\text{target}}$ in $\mathcal{T}_{\text{target}}$, and heap index $H_{\text{sim}}$ of the arrival in $\mathcal{T}_{\text{sim}}$.
$\mathcal{P} \leftarrow \text{PriorityQueue}$
$B \leftarrow \Omega$
$K \leftarrow \lfloor \log_2 H_{\text{target}} \rfloor$
$X \sim P$
$T \sim \text{Exp}(1)$
$\mathcal{P}.\texttt{push}(T, X, \Omega, 1)$
**for** $k = 0, \ldots, K$ **do**
  **repeat**
    $T, X, C, H \leftarrow \mathcal{P}.\texttt{pop}()$
    $L, R \leftarrow \texttt{split}_{\text{sim}}(C)$
    **if** $B \cap L \neq \emptyset$ **then**
      $\Delta_L \sim \text{Exp}(P(L))$
      $T_L \leftarrow T + \Delta_L$
      $X_L \sim P|_L$
      $\mathcal{P}.\texttt{push}(T_L, X_L, L, 2H)$
    **end**
    **if** $B \cap R \neq \emptyset$ **then**
      $\Delta_R \sim \text{Exp}(P(R))$
      $T_R \leftarrow T + \Delta_R$
      $X_R \sim P|_R$
      $\mathcal{P}.\texttt{push}(T_R, X_R, R, 2H + 1)$
    **end**
  **until** $X \in B$       /* Exit loop when we find first arrival in $B$. */
  **if** $k < K$ **then**
    $\{B_0, B_1\} \leftarrow \texttt{split}_{\text{target}}(B)$
    $d \leftarrow \left\lceil \frac{H_{\text{target}}}{2^{K-k}} \right\rceil \mod 2$
    $B \leftarrow B_d$
  **end**
**end**
**return** $T, X, H$

---

In this section, we examine an advanced simulation technique for Poisson processes, which is required to formulate the binary search-based variants of GPRS. We first recapitulate the tree-based simulation technique from (Flamich et al., 2022) and some important results from their work. Then, we present Algorithm 7, using which we can finally formulate the optimally efficient variant of GPRS in Appendix E. **Note:** For simplicity, we present the ideas for Poisson processes whose spatial measure $P$ is non-atomic. These ideas can also be extended to atomic spatial measures with appropriate modifications.

**Splitting functions:** Let $\Pi$ be a spatiotemporal Poisson process on some space $\Omega$ with non-atomic spatial measure $P$. In this paragraph, we will not deal with $\Pi$ itself yet, but the space on which it is defined and the measure $P$. Now, assume that there is a function $\texttt{split}$, which for any given Borel set $B \subseteq \Omega$, produces a (possibly random) $P$-*essential partition* of $B$ consisting of two Borel sets, i.e.

$$\texttt{split}(B) = \{L, R\}, \text{ such that } L \cap R = \emptyset \text{ and } P(L \cup R) = P(B). \qquad (165)$$

The last condition simply allows $\texttt{split}$ to discard some points from $B$ that will not be included in either the left or right split. For example, if we wish to design a splitting function for a subset of

$B \subseteq \mathbb{R}^d$ to split $B$ along some hyperplane, we do not need to include the points on the hyperplane in either set. The splitting function that always exists is the *trivial* splitting function, which just returns the original set and the empty set:

$$\mathtt{split}_{\text{trivial}}(B) = \{B, \emptyset\}. \tag{166}$$

A more interesting example when $\Omega = \mathbb{R}$ is the *on-sample* splitting function, where for a $\mathbb{R}$-valued random variable $X \sim P|_B$ it splits $B$ as

$$\mathtt{split}_{\text{on-samp}}(B) = \{(-\infty, X) \cap B, (X, \infty) \cap B\}. \tag{167}$$

This function is used by Algorithm 5 and AS* coding (Flamich et al., 2022). Another example is the *dyadic* splitting function operating on a bounded interval $(a, b)$, splitting as

$$\mathtt{split}_{\text{dyad}}((a,b)) = \left\{ \left(a, \frac{a+b}{2}\right), \left(\frac{a+b}{2}, b\right) \right\}. \tag{168}$$

We call this dyadic because when we apply this splitting function to $(0, 1)$ and subsets produced by applying it, we get the set of all intervals with dyadic endpoints on $(0, 1)$, i.e. $\{(0, 1), (0, 1/2), (1/2, 1), (0, 1/4), \ldots\}$. This splitting function is used by Algorithm 6 and AD* coding. As one might imagine, there might be many more possible splitting functions than the two examples we give above, all of which might be more or less useful in practice.

**Split-induced binary space partitioning tree:** Every splitting function on a space $\Omega$ induces an infinite set of subsets by repeatedly applying the splitting function to the splits it produces. These sets can be organised into an infinite *binary space partitioning tree (BSP-tree)*, where each node in the tree is represented by a set produced by $\mathtt{split}$ and an unique index. Concretely, let the root of the tree be represented by the whole space $\Omega$ and the index $H_{\text{root}} = 1$. Now we recursively construct the rest of the tree as follows: Let $(B, H)$ be a node in the tree, with $B$ a Borel set and $H$ its index, and let $\{L, R\} = \mathtt{split}(B)$ be the left and right splits of $B$. Then, we set $(L, 2H)$ as the node's left child and $(R, 2H + 1)$ as its right child. We refer to each node's associated index as their *heap index*.

**Heap-indexing the points in $\Pi$ and a strict heap invariant:** As we saw in Section 2.1, each point in $\Pi$ can be uniquely identified by their *time index* $N$, i.e. if we time-order the points of $\Pi$, $N$ represents the $N$th arrival in the ordered list. However, we can also uniquely index each point in $\Pi$ using a splitting function-induced BSP-tree as follows.

We extend each node in the BSP-tree with a point from $\Pi$, such that the extended tree satisfies a strict *heap invariant*: First, we extend the root node $(\Omega, 1)$ by adjoining $\Pi$'s first arrival $(T_1, X_1)$ and the first arrival index 1 to get $(T_1, X_1, \Omega, 1, 1)$. Then, for every other extended node $(T, X, B, H, N)$ with parent node $(T', X', B', \lfloor H/2 \rfloor, M)$ we require that $(T, X)$ is the first arrival in the restricted process $\Pi \cap ((T', \infty) \times B)$. This restriction enforces that we always have that $T' < T$ and, therefore, $M < N$. Furthermore, it is strict because there are no other points of $\Pi$ in $B$ between those two arrival times.

**Notation for the tree structure:** Let us denote the extended BSP tree $\mathcal{T}$ on $\Pi$ induced by $\mathtt{split}$. Each node $\nu \in \mathcal{T}$ is a 5-tuple $\nu = (T, X, B, H, N)$ consisting of the *arrival time $T$*, *spatial coordinate $X$*, its *bounds $B$*, $\mathtt{split}$-*induced heap index $H$* and *time index $N$*. As a slight overload of notation, for a node $\nu$, we use it as subscript to refer to its elements, e.g. $T_\nu$ is the arrival time associated with node $\nu$. We now prove the following important result, which ensures that we do not lose any information by using $\mathcal{T}$ to represent $\Pi$. An essentially equivalent statement was first proven for Gumbel processes (Appendix, "Equivalence under *partition*" subsection; Maddison et al., 2014).

**Lemma D.1.** *Let $\Pi$, $P$, $\mathtt{split}$ and $\mathcal{T}$ be as defined above. Then, $P$-almost surely, $\mathcal{T}$ contains every point of $\Pi$.*

*Proof.* We give an inductive argument.

*Base case:* the first arrival of $\Pi$ is by definition contained in the root node of $\mathcal{T}$.

*Inductive hypothesis:* assume that the first $N$ arrivals of $\Pi$ are all contained in $\mathcal{T}$.

*Case for $N + 1$:* We begin by observing that if the first $N$ nodes are contained in $\mathcal{T}$, they must form a connected subtree $\mathcal{T}_N$. To see this, assume the contrary, i.e. that the first $N$ arrivals form a subgraph $\Gamma_N \subseteq \mathcal{T}$ with multiple disconnected components. Let $\nu \in \Gamma_N$ be a node contained in a component

of $\Gamma_N$ that is *disconnected* from the component of $\Gamma_N$ containing the root node. Since $\mathcal{T}$ is a tree, there is a unique path $\pi$ from the root node to $\nu$ in $\mathcal{T}$, and since $\nu$ and the root are disconnected in $\Gamma_N$, $\pi$ must contain a node $c \notin \Gamma_N$. However, since $c$ is an ancestor of $\nu$, by the heap invariant of $\mathcal{T}$ we must have that the time index of $c$ is $N_c < N_\nu \leq N$ hence $c \in \Gamma_N$, a contradiction.

Thus, let $\mathcal{T}_N$ represent the subtree of $\mathcal{T}$ containing the first $N$ arrivals of $\Pi$. Now, let $\mathcal{F}_N$ represent the *frontier* of $\mathcal{T}_N$, i.e. the leaf nodes' children:

$$\mathcal{F}_N \overset{def}{=} \{\nu \in \mathcal{T} \mid \nu \notin \mathcal{T}_N, \text{parent}(\nu) \in \mathcal{T}_N\}, \tag{169}$$

where $\text{parent}$ retrieves the parent of a node in $\mathcal{T}$. Let

$$\bar{\Omega}_N \overset{def}{=} \bigcup_{\nu \in \mathcal{F}_N} B_\nu \tag{170}$$

be the union of all the bounds of the nodes in the frontier. A simple inductive argument shows that for all $N$ the nodes in $\mathcal{F}_N$ provide a $P$-essential partition of $\Omega$, from which $P(\bar{\Omega}_N) = 1$. Let $T_N$ be the $N$th arrival time of $\Pi$. Now, by definition, the arrival time associated with every node in the frontier $\mathcal{F}_N$ must be later than $T_N$. Finally, consider the first arrival time across the nodes in the frontier:

$$T_N^* \overset{def}{=} \min_{\nu \in \mathcal{F}_N} T_\nu. \tag{171}$$

Then, conditioned on $\mathcal{T}_N$, $T_N^*$ is the first arrival of $\Pi$ restricted to $(T_N, \infty) \times \bar{\Omega}$, thus it is $P$-almost surely the $N + 1$st arrival in $\Pi$, as desired. $\qquad\square$

**A connection between the time index an heap index of a node:** Now that we have two ways of uniquely identifying each point in $\Pi$ it is natural to ask whether there is any relation between them. In the next paragraph we adapt an argument from Flamich et al. (2022) to show that under certain circumstances, the answer is yes.

First, we need to define two concepts, the *depth* of a node in an extended BSP-tree and a *contractive* splitting function. Thus, let $\mathcal{T}$ be an extended BSP-tree over $\Pi$ induced by `split`. Let $\nu \in \mathcal{T}$. Then, the depth $D$ of $\nu$ in $\mathcal{T}$ is defined as the distance from $\nu$ to the root. A simple argument shows that the heap index $H_\nu$ of every node $\nu$ at depth $D$ in $\mathcal{T}$ is between $2^D \leq H_\nu < 2^{D+1}$. Thus, we can easily obtain the depth of a node $\nu$ from the heap index via the formula $D_\nu = \lfloor \log_2 H_\nu \rfloor$. Next, we say that `split` is *contractive* if **all** the bounds it produces shrink on average. More formally, let $\epsilon < 1$, and for a node $\nu \in \mathcal{T}$, let $\mathcal{A}_\nu$ denote the set of ancestor nodes of $\nu$. Then, `split` is contractive if for every non-root node $\nu \in \mathcal{T}$ we have

$$\mathbb{E}_{\mathcal{T}_{\mathcal{A}_\nu}|D_\nu}[P(B_\nu)] \leq \epsilon^{D_\nu}, \tag{172}$$

where $\mathcal{T}_{\mathcal{A}_\nu}$ and denotes the subtree of $\mathcal{T}$ containing the arrivals of the ancestors of $\nu$.

Note that $\epsilon \geq 1/2$ for any `split` function. This is because if $\{L, R\} = \text{split}(B)$ for some set $B$, then $P(R) = P(B) - P(L)$. Thus, if $P(R) = \alpha P(B)$, then $P(L) = (1 - \alpha)P(B)$, and by definition $\epsilon = \max\{\alpha, 1 - \alpha\}$, which is minimized when $\alpha = 1/2$, from which $\epsilon = 1/2$.

For example, $\text{split}_{\text{dyad}}$ is contractive with $\epsilon = 1/2$, while $\text{split}_{\text{trivial}}$ is not contractive. By Lemma 1 from Flamich et al. (2022), $\text{split}_{\text{on-samp}}$ is also contractive with $\epsilon = 3/4$. We now strengthen this result using a simple argument, and show that $\text{split}_{\text{on-samp}}$ is contractive with $\epsilon = 1/2$.

**Lemma D.2.** *Let $\nu, D$ be defined as above, let $P$ be a non-atomic probability measure over $\mathbb{R}$ with CDF $F_P$. Let $\Pi$ a $(1, P)$-Poisson process and $\mathcal{T}$ be the BSP-tree over $\Pi$ induced by $\text{split}_{\text{on-samp}}$. Then,*

$$\mathbb{E}_{\mathcal{T}_{\mathcal{A}_\nu}|D_\nu}[P(B_\nu)] = 2^{-D_\nu}. \tag{173}$$

*Proof.* Fix $D_\nu = d$, and let $\mathcal{N}_d = \{n \in \mathcal{T} \mid D_n = d\}$ be the set of nodes in $\mathcal{T}$ whose depth is $d$. Note, that $|\mathcal{N}_d| = 2^d$. We will show that

$$\forall n, m \in \mathcal{N}_d : \quad P(B_n) \overset{\text{d}}{=} P(B_m), \tag{174}$$

i.e. that the distributions of the bound sizes are all the same. From this, we will immediately have $\mathbb{E}[P(B_n)] = \mathbb{E}[P(B_\nu)]$ for every $n \in \mathcal{N}_d$. Then, using the fact, that the nodes in $\mathcal{N}_d$ for a $P$-almost partition of $\Omega$, we get:

$$1 = \mathbb{E}\left[P\left(\bigcup_{n \in \mathcal{N}_d} B_n\right)\right] = \mathbb{E}\left[\sum_{n \in \mathcal{N}_d} P(B_n)\right] = \sum_{n \in \mathcal{N}_d} \mathbb{E}\left[P(B_n)\right] = |\mathcal{N}_d| \cdot \mathbb{E}\left[P(B_\nu)\right]. \quad (175)$$

Dividing the very left and very right by $|\mathcal{N}_d| = 2^d$ yields the desired result.

To complete the proof, we now show that by symmetry, Equation (174) holds. We begin by exposing the fundamental symmetry of $\mathtt{split}_{\text{on-samp}}$: for a node $\nu$ with left child $L$ and right child $R$, the left and right bound sizes are equal in distribution:

$$P(B_L) \overset{\mathrm{d}}{=} P(B_r) \mid P(B_\nu). \quad (176)$$

First, note that by definition, all involved bounds will be intervals. Namely, assume that $B_\nu = (a, b)$ for some $a < b$ and $X_\nu$ is the sample associated with $\nu$. Then, $B_L = (a, X_\nu)$ and $B_R = (X_\nu, b)$ and hence $P(B_L) = F_P(X_\nu) - F_P(a)$. Since $X_\nu \sim P|_{B_\nu}$, by the probability integral transform, we have $F(X_\nu) \sim \mathrm{Unif}(F_P(a), F_P(b))$, from which $P(B_L) \sim \mathrm{Unif}(0, F_P(b) - F_P(a)) = \mathrm{Unif}(0, P(B_\nu))$. Since $P(B_R) = P(B_\nu) - P(B_L)$, we similarly have $P(B_R) \sim \mathrm{Unif}(0, P(B_\nu))$, which establishes our claim.

Now, to show Equation (174), fix $d$ and fix $n \in \mathcal{N}_d$. Let $\mathcal{A}_n$ denote the ancestor nodes of $n$. As we saw in the paragraph above,

$$P(B_n) \mid P(B_{\mathrm{parent}(n)}) \overset{\mathrm{d}}{=} P(B_{\mathrm{parent}(n)}) \cdot U, \quad U \sim \mathrm{Unif}(0, 1), \quad (177)$$

regardless of whether $n$ is a left or a right child of its parent. We can apply this $d$ times to the ancestors of $n$ find the marginal:

$$P(B_n) \overset{\mathrm{d}}{=} \prod_{i=1}^{d} U_i, \quad U_i \sim \mathrm{Unif}(0, 1). \quad (178)$$

Since the choice of $n$ was arbitrary, all nodes in $\mathcal{N}_d$ have this distribution, which is what we wanted to show. $\qquad \square$

Now, we have the following result.

**Lemma D.3.** *Let $\mathtt{split}$ be a contractive splitting function for some $\epsilon \in [1/2, 1)$. Then, for every node $\nu$ in $\mathcal{T}$ with time index $N_\nu$ and depth $D_\nu$, we have*

$$\mathbb{E}_{D_\nu \mid N_\nu}[D_\nu] \leq -\log_\epsilon N_\nu. \quad (179)$$

*Proof.* Let us examine the case $N_\nu = 1$ first. In this case, $\nu$ is the root of $\mathcal{T}$ and has depth $D_\nu = 0$ by definition. Thus, $0 \leq -\log_\epsilon 1$ holds trivially.

Now, fix $\nu \in \mathcal{T}$ with time index $N_\nu = N > 1$. Let $\mathcal{T}_{N-1}$ be the subtree of $\mathcal{T}$ containing the first $N-1$ arrivals, $\mathcal{F}_{N-1}$ be the frontier of $\mathcal{T}_{N-1}$ and $T_{N-1}$ the $(N-1)$st arrival time. Then, as we saw in Lemma D.1, the $N$th arrival in $\Pi$ occurs in one of the nodes in the frontier $\mathcal{F}_{N-1}$, after $T_{N-1}$. In particular, conditioned on $\mathcal{T}_{N-1}$, the arrival times associated with each node $f \in \mathcal{F}_{N-1}$ will be shifted exponentials $T_f = T_{N-1} + \mathrm{Exp}(P(B_f))$, and the $N$th arrival time in $\Pi$ is the minimum of these: $T_\nu = T_N = \min_{f \in \mathcal{F}_{N-1}} T_f$. It is a standard fact (see e.g. Lemma 6 in Maddison (2016)) that the index of the minimum

$$F_N = \arg\min_{f \in \mathcal{F}_{N-1}} T_f \quad (180)$$

is independent of $T_\nu = T_{F_N}$ and $\mathbb{P}[F_N = f \mid \mathcal{T}_{N-1}] = P(B_f)$. A simple inductive argument shows that the number of nodes on the frontier $|\mathcal{F}_{N-1}| = N$. Thus, we have a simple upper bound on the entropy of $F$:

$$\mathbb{E}_{F_N \mid \mathcal{T}_{N-1}, N_\nu = N}[-\log_2 P(B_\nu)] = \mathbb{H}[F_N \mid \mathcal{T}_{N-1}, N_\nu = N] \leq \log_2 N. \quad (181)$$

Thus, taking expectation over $\mathcal{T}_{N-1}$, we find

$$\log_2 N \overset{\text{eq. (181)}}{\geq} \mathbb{E}_{F_N, \mathcal{T}_{N-1} | N_\nu = N}[-\log_2 P(B_\nu)] \tag{182}$$

$$= \mathbb{E}_{D_\nu | N_\nu = N}\left[\mathbb{E}_{F_N, \mathcal{T}_{N-1} | D_\nu, N_\nu = N}[-\log_2 P(B_\nu)]\right] \tag{183}$$

$$\geq \mathbb{E}_{D_\nu | N_\nu = N}\left[-\log_2 \mathbb{E}_{F_N, \mathcal{T}_{N-1} | D_\nu, N_\nu = N}[P(B_\nu)]\right] \tag{184}$$

$$\overset{\text{eq. (172)}}{\geq} \mathbb{E}_{D_\nu | N_\nu = N}\left[-\log_2 \epsilon^{D_\nu}\right] \tag{185}$$

$$= (-\log_2 \epsilon) \cdot \mathbb{E}_{D_\nu | N_\nu = N}[D_\nu]. \tag{186}$$

The second inequality holds by Jensen's inequality. In the third inequality, we apply Equation (172), and one might worry about conditioning on $N_\nu$ here. However, this is not an issue because

$$\mathbb{E}_{F_N, \mathcal{T}_{N-1} | D_\nu = d, N_\nu = N}[P(B_\nu)] \tag{187}$$

$$= \mathbf{1}[d \leq N-1] \cdot \sum_{f \in \mathcal{F}_{N-1}} \mathbb{E}_{\mathcal{T}_{A_f} | F_N = f, D_\nu = d}[P(B_f)] \cdot \mathbb{P}[F_N = f \mid D_f = d] \tag{188}$$

$$\overset{\text{eq. (172)}}{\leq} \mathbf{1}[d \leq N-1] \cdot \sum_{f \in \mathcal{F}_{N-1}} \epsilon^d \cdot \mathbb{P}[F_N = f \mid D_f = d] \tag{189}$$

$$= \mathbf{1}[d \leq N-1] \cdot \epsilon^d. \tag{190}$$

Thus, we finally get

$$(-\log_2 \epsilon) \cdot \mathbb{E}_{D_\nu | N_\nu = N}[D_\nu] \leq \log_2 N \quad \Rightarrow \quad \mathbb{E}_{D_\nu | N_\nu}[D_\nu] \leq -\log_\epsilon N_\nu \tag{191}$$

by dividing both sides by $-\log_2 \epsilon$ and we obtain the desired result. $\qquad\square$

**Converting between different heap indices:** Assume now that we have two splitting functions, $\mathtt{split}_{\text{target}}$ and $\mathtt{split}_{\text{sim}}$, which induce their own BSP-ordering on $\Pi$, $\mathcal{T}_{\text{target}}$ and $\mathcal{T}_{\text{sim}}$. Now, given a $\mathtt{split}_{\text{target}}$-induced heap index $H$, Algorithm 7 presents a method for simulating the appropriate node $\nu \in \mathcal{T}_{\text{target}}$ by simulating nodes from $\mathcal{T}_{\text{sim}}$. In other words, given a node with some heap index induced by a splitting function, Algorithm 7 lets us find the heap index of the same arrival induced by a different splitting function. The significance of Algorithm 7 is that it lets us develop convenient search methods using a given splitting function, but it might be more efficient to encode the heap index induced by another splitting function.

**Theorem D.4.** *Let $\Pi$, $\mathtt{split}_{\text{target}}$, $\mathtt{split}_{\text{sim}}$, $\mathcal{T}_{\text{target}}$ and $\mathcal{T}_{\text{sim}}$ be as above. Let $\nu \in \mathcal{T}_{\text{target}}$ with and let $(T_{\text{sim}}, X_{\text{sim}}, H_{\text{sim}})$ be the arrival and its heap index output by Algorithm 7 given the above as input as well as $H_\nu$ as the target index. Then, Algorithm 7 is correct, in the sense that*

$$T_\nu \overset{d}{=} T_{\text{sim}} \quad \text{and} \quad X_\nu \overset{d}{=} X_{\text{sim}}, \tag{192}$$

*and $H_{\text{sim}}$ is the heap index of $(T_{\text{sim}}, X_{\text{sim}})$ in $\mathcal{T}_{\text{sim}}$.*

*Proof.* First, observe that when $\mathtt{split}_{\text{target}} = \mathtt{split}_{\text{sim}}$, Algorithm 7 collapses onto just the extended BSP tree construction for $\Pi$ and simply returns the arrival with the given heap index $H_{\text{target}}$ in $\mathcal{T}_{\text{target}}$. In particular, the inner loop will always exit after one iteration, and every time one and only one of the conditional blocks will be executed. In other words, in this case, the algorithm becomes equivalent to Algorithm 2 in Flamich et al. (2022).

Let us now consider the case when $\mathtt{split}_{\text{target}} \neq \mathtt{split}_{\text{sim}}$. Denote the depth of the required node by $D_\nu = \lfloor \log_2 H_\nu \rfloor$. Now, we give an inductive argument for correctness.

*Base case:* Consider $D_\nu = 0$. In this case, the target bounds $B = \Omega$, and the first sample we draw $X \sim P$ is guaranteed to fall in $\Omega$. Hence, for $D_\nu = 0$ the outer loop only runs for one iteration. Furthermore, during that iteration, the inner loop will also exit after one iteration, and Algorithm 7 returns the sample $(T, X)$ sampled before the outer loop with heap index 1. Since $T \sim \text{Exp}(1)$ and $X \sim P$, this will be a correctly distributed output with the appropriate heap index.

*Inductive hypothesis:* Assume Algorithm 7 is correct heap indices with depths up to $D_\nu = d$.

*Case $D_\nu = d+1$:* Let $\rho \in \mathcal{T}_{\text{target}}$ be the parent node of $\nu$ with arrival $(T_\rho, X_\rho)$. Then, $D_\rho = d$, hence by the inductive hypothesis, Algorithm 7 will correctly simulate a branch $\mathcal{T}_{\text{target}}$ up to node $\rho$. At the end of the $d$th iteration of the outer loop Algorithm 7 sets the target bounds $B \leftarrow B_\nu$. Then, in the final, $d+1$st iteration, the inner loop simply realizes $\mathcal{T}_{\text{sim}}$ and accepts the first sample after $X$ that falls inside $B_\nu$ whose time index $T > T_\rho$. Due to the priority queue, the loop simulates the nodes of $\mathcal{T}_{\text{sim}}$ in time order; hence the accepted sample will also be the one with the earliest arrival time. Furthermore, Algorithm 7 only ever considers nodes of $\mathcal{T}_{\text{sim}}$ whose bounds intersect the target bounds $B_\nu$, hence the inner loop is guaranteed to terminate, which finishes the proof. $\qquad\square$

## E   GPRS with Binary Search

We now utilise the machinery we developed in Appendix D to analyze Algorithm 5.

**Correcntess of Algorithm 5:** Observe that Algorithm 5 constructs the extended BSP tree for the on-sample splitting function. Thus, we will now focus on performing a binary tree search using the extended BSP tree induced by the on-sample splitting function, which we denote by $\mathcal{T}$. The first step of the algorithm matches GPRS's first step (Algorithm 3). Hence it is correct for the first step. Now consider the algorithm in an arbitrary step $k$ before termination, where the candidate sample $(T, X)$ is rejected, i.e. $T > \varphi(X)$. By assumption, the density ratio $r$ is unimodal, and since $\sigma$ is monotonically increasing, $\varphi = \sigma \circ r$ is unimodal too. Thus, let $x^* \in \Omega$ be such that $r(x^*) = r^*$, where $r^* = \exp_2(D_\infty[Q\|P])$. Assume for now that $X < x^*$, the case $X > x^*$ follows by a symmetric argument. By the unimodality assumption, since $X < x^*$, it must hold that for all $y < X$, we have $\varphi(y) < \varphi(X)$. Consider now the arrival $(T_L, X_L)$ of $\Pi$ in the current node's left child. Then, we will have $T < T_L$ and $X_L < X$ by construction. Thus, finally, we get

$$\varphi(X_L) < \varphi(X) < T < T_L, \tag{193}$$

meaning that the current node's left child is also guaranteed to be rejected. This argument can be easily extended to show that any left-descendant of the current node will be rejected, and it is sufficient to search its right-descendants only. By a similar argument, when $X > x^*$, we find that it is sufficient only to check the current node's left-descendants. Finally, since both Algorithms 3 and 5 simulate $\Pi$ and search for its first arrival under $\varphi$, by the construction in Appendix A, the sample returned by both algorithms will follow the desired target distribution.

**Analyzing the expected runtime and codelength:** Since Algorithm 5 simulates a single branch of $\mathcal{T}_{\text{on-sample}}$, its runtime is equal to the depth $D$ of the branch. Furthermore, let $N$ denote the time index of the accepted arrival and $H$ its heap index in $\mathcal{T}_{\text{on-sample}}$, which means $D = \lfloor \log_2 H \rfloor$. It is tempting to use lemmas D.2 and D.3 to in our analysis of $N$, $H$ and $D$. However, they do not apply, because they assume no restriction on $D$ and $N$, while in this section, we condition on the fact that $N$, $H$ and $D$ identify the arrival in $\Pi$ that was accepted by Algorithm 5. Hence, we first prove a lemma analogous to lemma D.2 in this restricted case, based on Lemma 4 of Flamich et al. (2022). Note the additional assumptions we need for our result.

**Lemma E.1.** *Let $Q$ and $P$ be distributions over $\mathbb{R}$ with unimodal density ratio $r = dQ/dP$, given to Algorithm 5 as the target and proposal distribution as input, respectively. Assume $P$ has CDF $F_P$. Let $H$ denote the accepted sample's heap index, let $D = \lfloor \log_2 H \rfloor$, and let $B_H$ denote the bounds associated with the accepted node. Then,*

$$\mathbb{E}[P(B_H) \mid H] \leq \left(\frac{3}{4}\right)^D. \tag{194}$$

*Proof.* We prove the claim by induction. For $H = 1$, $D = 0$ the claim holds trivially, since $B_1 = \mathbb{R}$, hence $P(B_1) = 1$. Assume now that the claim holds for $D = \lfloor \log_2 H \rfloor = d - 1$ for $d \geq 1$, and we prove the statement when $D = \lfloor \log_2 H \rfloor = d$. Let $X_0, \ldots, X_{d-1}, X_d$ denote the samples simulated along the branch that leads to the accepted node, i.e. $X_0$ is the sample associated with the root node and $X_d$ is the accepted sample. By the definition of the on-sample splitting function and law of iterated expectations, we have

$$\mathbb{E}_{X_{0:d-1}}[P(B_H) \mid H] = \begin{cases} \mathbb{E}_{X_0}[P(B_H) \mid H] & \text{when } d = 1 \\ \mathbb{E}_{X_{0:d-2}}[\mathbb{E}_{X_{d-1}|X_{0:d-2}}[P(B_H) \mid H] \mid H] & \text{when } d \geq 2. \end{cases} \tag{195}$$

Below, we only examine the inner expectation in the $d \geq 2$ case; computing the expectation in the $d = 1$ case follows mutatis mutandis. First, denote the bounds of the accepted node's parent as $B_{\lfloor H/2 \rfloor} = (\alpha, \beta)$ for some real numbers $\alpha < \beta$ and define $a = F_P(\alpha), b = F_P(\beta)$ and $U = F_P(X_{d-1})$. Since $X_{d-1} \mid X_{1:d-2} \sim P|_{B_{\lfloor H/2 \rfloor}}$, by the generalized probability integral transform we have $U \sim \text{Unif}(a, b)$, where $\text{Unif}(a, b)$ denotes the uniform distribution on the interval $(a, b)$. The two possible intervals from which Algorithm 5 will choose are $(\alpha, X_{d-1})$ and $(X_{d-1}, \beta)$, whose measures are $P((\alpha, X_{d-1})) = F_P(X_{d-1}) - F_P(\alpha) = U - a$ and similarly $P((X_{d-1}, \beta)) = b - U$. Then, in the worst case, the algorithm always picks the bigger of the two intervals, thus $P(B_H) \leq \max\{U - A, B - U\}$, from which we obtain the bound

$$\mathbb{E}_{X_{d-1} \mid X_{1:d-2}}[P(B_H) \mid H] \leq \mathbb{E}_U[\max\{U - a, b - U\}] = \frac{3}{4}(b - a) = \frac{3}{4} P(B_{\lfloor H/2 \rfloor}). \quad (196)$$

Plugging this into Equation (195), we get

$$\mathbb{E}_{X_{1:d-1}}[P(B_H) \mid H] \leq \frac{3}{4} \mathbb{E}_{X_{1:d-2}}\left[P(B_{\lfloor H/2 \rfloor}) \mid H\right] \quad (197)$$

$$\leq \frac{3}{4} \cdot \left(\frac{3}{4}\right)^{d-1}, \quad (198)$$

where the second inequality follows from the inductive hypothesis, which finishes the proof. $\square$

Fortunately, lemma D.3 does not depend on this condition, so we can modify it mutatis mutandis so that it still applies in this latter case as well, and we restate it here for completeness:

**Lemma E.2.** *Let $N, D, H, B_H$ be defined as above. Then*

$$\mathbb{E}[-\log P(B_H)] \leq \log_2 N \quad (199)$$

*and*

$$\mathbb{E}_{D \mid N}[D] \leq -\log_{3/4} N. \quad (200)$$

*Proof.* Equation (199) follows via the same argument as Equation (181), and Equation (200) follows via the same argument as lemma D.3, combined with lemma E.1. $\square$

**Expected runtime:** Combining Equation (200) with Equation (143) yields

$$\mathbb{E}[D] \leq \frac{D_{\text{KL}}[Q \| P] + 2 \cdot \log_2 e}{\log_2(4/3)}, \quad (201)$$

which proves Theorem 3.6. In the channel simulation case, taking a pair of correlated random variables $\mathbf{x}, \mathbf{y} \sim P_{\mathbf{x}, \mathbf{y}}$ and setting $Q \leftarrow P_{\mathbf{x} \mid \mathbf{y}}$ and $P \leftarrow P_{\mathbf{x}}$, and taking expectation over $\mathbf{y}$, we get

$$\mathbb{E}[D] \leq \frac{I[\mathbf{x}; \mathbf{y}] + 2 \cdot \log_2 e}{\log_2(4/3)}. \quad (202)$$

**Entropy coding the heap index:** Let $\nu \in \mathcal{T}_{\text{on-sample}}$ now denote the node accepted and returned by Algorithm 5. We will encode $\nu$ using its heap index $H$. As we saw in Appendix D, $H$ can be decomposed into two parts: the depth $D$ and the *residual* $R$:

$$H = 2^D + R, \quad 0 \leq R < 2^D. \quad (203)$$

Then, we have

$$\mathbb{H}[H] \leq \mathbb{H}[D, R] \quad (204)$$

$$= \mathbb{H}[R \mid D] + \mathbb{H}[D]. \quad (205)$$

Dealing with the second term first, we note that the maximum entropy distribution for a random variable $Z$ over the positive integers with the moment constraint $\mathbb{E}[Z] = 1/p$ is the geometric distribution with rate $p$. Thus, based on Equation (202), we set

$$p = \frac{\log_2(4/3)}{I[\mathbf{x}; \mathbf{y}] + 2 \cdot \log_2 e}, \quad (206)$$

from which we get

$$\mathbb{H}[D] \leq \mathbb{H}\left[\text{Geom}\left(p\right)\right] \tag{207}$$

$$= -\log_2 p - \frac{1-p}{p}\log_2(1-p) \tag{208}$$

$$\leq \log_2(I[\mathbf{x};\mathbf{y}] + 2\log_2 e) - \log_2\log_2(4/3) + \log_2 e \tag{209}$$

$$\leq \log_2(I[\mathbf{x};\mathbf{y}] + 1) + \log_2(2\log_2 e) - \log_2\log_2(4/3) + \log_2 e, \tag{210}$$

where Equation (208) follows from the fact that the term $-\frac{1-p}{p}\log_2(1-p)$ is bounded from above by $\log_2 e$, which it attains as $I[\mathbf{x};\mathbf{y}] \to \infty$.

For the $\mathbb{H}[R \mid D]$ term, we begin by noting that there is a particularly natural encoding for $R$ given a particular run of the algorithm. Let $B_1^{(D)}, \ldots, B_{2^D}^{(D)}$ be the bounds associated with the depth-$D$ nodes of $\mathcal{T}_{\text{on-sample}} \mid D$, the split-on-sample BSP tree given $D$. Then, since $B_1^{(D)}, \ldots, B_{2^D}^{(D)}$ form a $P$-essential partition of $\Omega$, we can encode $R$ by encoding the bounds associated with the accepted sample, $B_R^{(D)}$. Here, we can borrow a technique from arithmetic coding (Rissanen & Langdon, 1979): we encode *the largest dyadic interval that is a subset of* $B_R^{(D)}$, which can be achieved using $-\log_2 P(B_R^{(D)}) + 1$ bits. Note, that Algorithm 5 only ever realises a single branch of $\mathcal{T}_{\text{on-sample}}$, hence, in this section only, let $X_{1:\infty}$ denote the infinite sequence of samples along this branch, starting with $X_1 \sim P$ at the root. Furthermore, from here onwards, let $B_d$ and $T_d$ denote the bounds and arrival time associated with $X_d$ in the sequence, respectively. Then, by the above reasoning, we have

$$\mathbb{H}[R \mid D] \leq \mathbb{E}_{X_{1:\infty},T_{1:\infty}}\left[\mathbb{E}_D[-\log_2 P(B_D)]\right] + 1 \tag{211}$$

$$= -\sum_{k=1}^{\infty}\mathbb{E}_{X_{1:k-1},T_{1:k}}\left[\mathbb{P}[D = k \mid X_{1:k-1}, T_{1:k}]\log_2 P(B_k)\right] + 1. \tag{212}$$

We now develop a bound on $P(B_k)$. To this end, note, that by the definition of GPRS,

$$\mathbb{P}\left[D \geq k+1 \mid X_{1:k-1}, T_{1:k}, D \geq k\right] = \mathbb{P}\left[\mathbf{N}_{\tilde{\Pi}}([T_{k-1}, T_k] \times B_k) = 0 \mid X_{1:k-1}, T_{1:k-1}, D \geq k\right].$$

Hence,

$$\mathbb{P}\left[D = k \mid X_{1:k-1}, T_{1:k}, D \geq k\right] = 1 - \mathbb{P}\left[D \geq k+1 \mid X_{1:k-1}, T_{1:k}, D \geq k\right] \tag{213}$$

$$= \mathbb{P}\left[\mathbf{N}_{\tilde{\Pi}}([T_{k-1}, T_k] \times B_k) \geq 1 \mid X_{1:k-1}, T_{1:k-1}, D \geq k\right] \tag{214}$$

$$\leq \mathbb{E}\left[\mathbf{N}_{\tilde{\Pi}}([T_{k-1}, T_k] \times B_k) \mid X_{1:k-1}, T_{1:k-1}, D \geq k\right] \tag{215}$$

$$\leq \mathbb{E}\left[\mathbf{N}_{\Pi}([T_{k-1}, T_k] \times B_k) \mid X_{1:k-1}, T_{1:k-1}, D \geq k\right] \tag{216}$$

$$= P(B_k)(T_k - T_{k-1}), \tag{217}$$

where eq. (215) follows from Markov's inequality, and eq. (216) follows from the fact that the mean measure of $\tilde{\Pi}$ is just a restricted version of $\Pi$'s mean measure. For convenience, we now write $\Delta_k = T_k - T_{k-1}$. Then, rearranging the above inequality, we get

$$-\log_2 P(B_k) \leq -\log_2 \mathbb{P}\left[D = k \mid X_{1:k-1}, T_{1:k-1}, \Delta_k, D \geq k\right] + \log_2 \Delta_k. \tag{218}$$

Substituting this into eq. (212), we get

$$\mathbb{H}[R \mid D] \tag{219}$$

$$\leq -\sum_{k=1}^{\infty}\mathbb{E}_{X_{1:k-1},T_{1:k-1},\Delta_k}\left[\mathbb{P}[D = k \mid X_{1:k-1}, T_{1:k-1}, \Delta_k]\log_2 \mathbb{P}[D = k \mid X_{1:k-1}, T_{1:k-1}, \Delta_k]\right]$$

$$+ \sum_{k=1}^{\infty}\mathbb{E}_{X_{1:k-1},T_{1:k-1},\Delta_k}\left[\mathbb{P}[D = k \mid X_{1:k-1}, T_{1:k-1}, \Delta_k]\log_2 \Delta_k\right] + 1. \tag{220}$$

We deal with the above two sums separately. In the first sum, we apply Jensen's inequality on the concave function $x \mapsto -x \log_2 x$ in each summand, from which we get

$$-\sum_{k=1}^{\infty} \mathbb{E}_{X_{1:k-1}, T_{1:k-1}, \Delta_k} \left[ \mathbb{P}[D = k \mid X_{1:k-1}, T_{1:k-1}, \Delta_k] \log_2 \mathbb{P}[D = k \mid X_{1:k-1}, T_{1:k-1}, \Delta_k] \right]$$

$$\leq -\sum_{k=1}^{\infty} \mathbb{P}[D = k] \log_2 \mathbb{P}[D = k] \tag{221}$$

$$= \mathbb{H}[D]. \tag{222}$$

For the second sum, consider the $k$th summand first:

$$\mathbb{P}[D = k \mid X_{1:k-1}, T_{1:k-1}, \Delta_k, D \geq k] \log_2 \Delta_k \tag{223}$$

$$= \int_{B_k} \frac{d\mathbb{P}[D = k, X_k = x \mid X_{1:k-1}, T_{1:k-1}, \Delta_k, D \geq k]}{dP} \log_2 \Delta \, dP(x).$$

$$= \frac{1}{P(B_k)} \int_{B_k} \mathbf{1}[r(x) \geq \sigma^{-1}(T_k)] \log_2 \Delta \, dP(x).$$

To continue, we now examine the integrand:

$$r(x) \geq \sigma^{-1}(T_k) \overset{\text{eq. (132)}}{\geq} \mathbb{P}[\tilde{T} \geq T_k] \cdot T_k \geq \mathbb{P}[\tilde{T} \geq T_k] \cdot \Delta_k \tag{224}$$

from which

$$\log_2 \Delta_k \leq \log_2 r(x) - \log_2 \mathbb{P}[\tilde{T} \geq T_k]. \tag{225}$$

Substituting this back, we get

$$\mathbb{P}[D = k \mid X_{1:k-1}, T_{1:k-1}, \Delta_k, D \geq k] \log_2 \Delta_k \tag{226}$$

$$\leq \frac{1}{P(B_k)} \int_{B_k} \mathbf{1}[r(x) \geq \sigma^{-1}(T_k)] \log_2 r(x) \, dP(x)$$

$$- \mathbb{P}[D = k \mid X_{1:k-1}, T_{1:k-1}, \Delta_k, D \geq k] \log_2 \mathbb{P}[\tilde{T} \geq T_k]. \tag{227}$$

Therefore, for the second summand, we get

$$\sum_{k=1}^{\infty} \mathbb{E}_{X_{1:k-1}, T_{1:k-1}, \Delta_k} \left[ \mathbb{P}[D = k \mid X_{1:k-1}, T_{1:k-1}, \Delta_k] \log_2 \Delta_k \right] \tag{228}$$

$$\leq \sum_{k=1}^{\infty} \mathbb{E}_{X \sim P} \left[ \frac{d\mathbb{P}[D = k, X_k = X]}{dP} \log_2 r(X) \right] - \sum_{k=1}^{\infty} \mathbb{P}[D = k] \log_2 \mathbb{P}[\tilde{T} \geq T_k]. \tag{229}$$

For the first term, by Fubini's theorem and by the correctness of GPRS, we get

$$\sum_{k=1}^{\infty} \mathbb{E}_{X \sim P} \left[ \frac{d\mathbb{P}[D = k, X_k = X]}{dP} \log_2 r(X) \right] = \mathbb{E}_{X \sim P} \left[ \sum_{k=1}^{\infty} \frac{d\mathbb{P}[D = k, X_k = X]}{dP} \log_2 r(X) \right] \tag{230}$$

$$= \mathbb{E}_{X \sim P} \left[ r(X) \log_2 r(X) \right] \tag{231}$$

$$= D_{\text{KL}}[Q \| P]. \tag{232}$$

For the second term, note that

$$\mathbb{P}[D = k] = \mathbb{P}[\tilde{T} = T_k]. \tag{233}$$

Hence,

$$-\sum_{k=1}^{\infty} \mathbb{P}[D = k] \log_2 \mathbb{P}[\tilde{T} \geq T_k] = -\sum_{k=1}^{\infty} \mathbb{P}[D = k] \log_2 \mathbb{P}[D \geq k] \tag{234}$$

$$= \sum_{k=1}^{\infty} (\mathbb{P}[D \geq k] - \mathbb{P}[D \geq k + 1]) \log_2 \frac{1}{\mathbb{P}[D \geq k]} \tag{235}$$

$$\leq -\int_1^0 \log_2 p \, dp \tag{236}$$

$$= \log_2 e. \tag{237}$$

Thus, we finally get that

$$\mathbb{H}[R \mid D] \leq D_{\mathrm{KL}}[Q\|P] + \mathbb{H}[D] + \log_2 e + 1. \tag{238}$$

In the average case, this yields

$$\mathbb{H}[R \mid D] \leq I[\mathbf{x};\mathbf{y}] + \mathbb{H}[D] + \log_2 e + 1. \tag{239}$$

Thus, we finally get that the joint entropy is bounded above by

$$\mathbb{H}[R, D] \leq I[\mathbf{x};\mathbf{y}] + 2\log_2(I[\mathbf{x};\mathbf{y}] + 1) \tag{240}$$
$$+ 2\left(\log_2(2\log_2 e) - \log_2\log_2(4/3) + \log_2 e\right) + \log_2 e + 1 \tag{241}$$
$$< I[\mathbf{x};\mathbf{y}] + 2\log_2(I[\mathbf{x};\mathbf{y}] + 1) + 11, \tag{242}$$

which finally proves Theorem 3.7.

# F    General GPRS with Binary Search

We finally present a generalized version of branch-and-bound GPRS (Algorithm 6) for more general spaces and remove the requirement that $r$ be unimodal.

**Decomposing a Poisson process into a mixture of processes:** Let $\Pi$ be a process over $\mathbb{R}^+ \times \Omega$ as before, with spatial measure $P$, and let $Q$ be a target measure, $\varphi = \sigma \circ r$ and $U = \{(t, x) \in \mathbb{R}^+ \times \Omega \mid t \leq \varphi(x)\}$ as before. Let $\tilde{\Pi}$ be $\tilde{\Pi} = \Pi \cap U$ restricted under $\varphi$ and $(\tilde{T}_1, \tilde{X}_1)$ its first arrival. Let $\{L, R\}$ form an $P$-essential partiton of $\Omega$, i.e. $L \cap R = \emptyset$ and $P(L \cup R) = 1$, and let $\Pi_L = \Pi \cap \mathbb{R}^+ \times L$ and $\Pi_R = \Pi \cap \mathbb{R}^+ \times R$ be the restriction of $\Pi$ to $L$ and $R$, respectively. Let $\tilde{\Pi}_L = \Pi_L \cap U$ and $\tilde{\Pi}_R = \Pi_R \cap U$ be the restrictions of the two processes under $\varphi$ as well. Let $\tilde{\mu}_L(t)$ and $\tilde{\mu}_R(t)$ be the mean measures of these processes. Thus, the first arrival times in these processes have survival functions

$$\mathbb{P}[\tilde{T}_1^L \geq t] = e^{-\tilde{\mu}_L(t)} \tag{243}$$
$$\mathbb{P}[\tilde{T}_1^R \geq t] = e^{-\tilde{\mu}_R(t)}. \tag{244}$$

Note, that by the superposition theorem, $\Pi_L \cup \Pi_R = \Pi$, hence the first arrival $(\tilde{T}_1, \tilde{X}_1)$ occurs in either $\Pi_L$ or $\Pi_R$. Assume now that we have already searched through the points of $\Pi$ up to time $\tau$ without finding the first arrival. At this point, we can ask: will the first arrival occur in $\Pi_L$, given that $\tilde{T}_1 \geq \tau$? Using Bayes' rule, we find

$$\mathbb{P}[\tilde{X}_1 \in L \mid \tilde{T}_1 \geq \tau] = \frac{\mathbb{P}[\tilde{X}_1 \in L, \tilde{T}_1 \geq \tau]}{\mathbb{P}[\tilde{T}_1 \geq \tau]}. \tag{245}$$

More generally, assume that the first arrival of $\Pi$ occurs in some set $A \subseteq \Omega$, and we know that the first arrival time is larger than $\tau$. Then, what is the probability that the first arrival occurs in some set $B \subseteq A$? Similarly to the above, we find

$$\mathbb{P}[\tilde{X}_1 \in B \mid \tilde{T}_1 \geq \tau, \tilde{X}_1 \in A] = \frac{\mathbb{P}[\tilde{X}_1 \in B, \tilde{T}_1 \geq \tau, \tilde{X}_1 \in A]}{\mathbb{P}[\tilde{T}_1 \geq \tau, \tilde{X}_1 \in A]} \tag{246}$$
$$= \frac{\mathbb{P}[\tilde{T}_1 \geq \tau, \tilde{X}_1 \in B]}{\mathbb{P}[\tilde{T}_1 \geq \tau, \tilde{X}_1 \in A]}. \tag{247}$$

Let $(\tilde{T}_L, \tilde{X}_L)$ be the first arrival of $\Pi_L$. Then, the crucial observation is that

$$(\tilde{T}_L, \tilde{X}_L) \overset{d}{=} \tilde{T}_1, \tilde{X}_1 \mid \tilde{X}_1 \in L. \tag{248}$$

This enables us to search for the first arrival of $\Pi$ under the graph of $\varphi$ using an extended BSP tree construction. At each node, if we reject, we draw a Bernoulli random variable $b$ with mean equal to the probability that the first arrival occurs within the bounds associated with the right child node. Then, if $b = 1$, we continue the search along the right branch. Otherwise, we search along the left branch.

Note, however, that in a restricted process $\Pi_A$, the spatial measure no longer integrates to 1. Furthermore, our target Radon-Nikodym derivative is $r(x) \cdot \mathbf{1}[x \in A]/Q(A)$. This means we need to change

the graph $\varphi$ to some new graph $\varphi_A$ to ensure that the spatial distribution of the returned sample is still correct. Therefore, for a set $A$ we define the restricted versions of previous quantities:

$$\tilde{\mu}_A(t) \stackrel{def}{=} \int_0^t \int_A \mathbf{1}[\tau \leq \varphi_A(x)] \, dP(x) \, dt \tag{249}$$

$$w_P(h \mid A) \stackrel{def}{=} \int_A \mathbf{1}[h \leq r(x)] \, dP(x) \tag{250}$$

Then, via analogous arguments to the ones in Appendix A, we find

$$\frac{d\mathbb{P}[\tilde{T}_A = t, \tilde{X}_A = x]}{d(\lambda \times P)} = \mathbf{1}[x \in A]\mathbf{1}[t \leq \varphi_A(x)]\mathbb{P}[\tilde{T}_A \geq t] \tag{251}$$

$$\frac{d\mathbb{P}[\tilde{X}_A = x]}{dP} = \mathbf{1}[x \in A] \int_0^{\varphi_A(x)} \mathbb{P}[\tilde{T}_A \geq t] \, dt \tag{252}$$

$$\varphi_A = \sigma_A \circ r. \tag{253}$$

Similarly, setting $\frac{d\mathbb{P}[\tilde{X}_A = x]}{dP} = \mathbf{1}[x \in A] \cdot r(x)/Q(A)$, and setting $\tau = \sigma_A(r(x))$, we get

$$\sigma_A^{-1}(\tau) = Q(A) \int_0^\tau \mathbb{P}[\tilde{T}_A \geq t] \, dt \tag{254}$$

$$\Rightarrow \quad \left(\sigma_A^{-1}\right)'(\tau) = \mathbb{P}[\tilde{T}_A \geq \tau] \tag{255}$$

$$= \mathbb{P}[\tilde{T} \geq \tau, \tilde{X} \in A]. \tag{256}$$

From this, again using similar arguments to the ones in Appendix A, we find

$$\left(\sigma_A^{-1}\right)'(\tau) = w_Q(\sigma_A^{-1}(t) \mid A) - \sigma_A^{-1}(t) \cdot w_P(\sigma_A^{-1}(t) \mid A). \tag{257}$$

## G   Necessary Quantities for Implementing GPRS in Practice

Ultimately, given a target-proposal pair $(Q, P)$ with density ratio $r$, we would want an easy-to-evaluate expression for the appropriate stretch function $\sigma$ or $\sigma^{-1}$ to plug directly into our algorithms. Computing $\sigma$ requires computing the integral in Equation (5) and finding $\sigma^{-1}$ requires solving the non-linear ODE in Equation (4), neither of which is usually possible in practice. Hence, we usually resort to computing $\sigma^{-1}$ numerically by using an ODE solver for Equation (4).

In any case, we need to compute $w_P$ and $w_Q$, which are analytically tractable in the practically relevant cases. Hence, in this section, we give closed-form expressions for $w_P$ and $w_Q$ for all the examples we consider and give closed-form expressions for $\sigma$ and $\sigma^{-1}$ for some of them. If we do not give a closed-form expression of $\sigma$, we use numerical integration to compute $\sigma^{-1}$ instead.

**Note on notation:** In this section only, we denote the natural logarithm as $\ln$ and exponentiation with respect to the natural base as $\exp$.

### G.1   Uniform-Uniform Case

Let $P$ be the uniform distribution over an arbitrary space $\Omega$ and $Q$ a uniform distribution supported on some subset $\mathcal{X} \subset \Omega$, with $P(\mathcal{X}) = C$ for some $C \leq 1$ Then,

$$r(x) = \frac{1}{C} \cdot \mathbf{1}[x \in \mathcal{X}] \tag{258}$$

$$w_P(h) = C \tag{259}$$

$$w_Q(h) = 1 \tag{260}$$

$$\sigma(h) = -\frac{1}{C} \log\left(1 - C \cdot h\right) \tag{261}$$

$$\sigma^{-1}(t) = \frac{1}{C}\left(1 - \exp(-C \cdot h)\right). \tag{262}$$

Note that using GPRS in the uniform-uniform case is somewhat overkill, as in this case, it is simply equivalent to standard rejection sampling.

## G.2 Triangular-Uniform Case

Let $P = \mathrm{Unif}(0,1)$ and for some numbers $0 < a < c < b < 1$, let $Q$ be the triangular distribution, defined by the PDF

$$q(x) = \begin{cases} 0 & \text{if } x < a \\ \frac{2(x-a)}{(b-a)(c-a)} & \text{if } a \le x < c \\ \frac{2}{b-a} & \text{if } x = c \\ \frac{2(b-x)}{(b-a)(b-c)} & \text{if } c < x \le b \\ 0 & \text{if } b < x. \end{cases} \tag{263}$$

In this parameterization, the distribution has support on $(a, b)$. Moreover, it is linearly increasing on $(a, c)$ and linearly decreasing on $(c, b)$, with the mode located at $c$. For convenience, let $\ell = b - a$. Then,

$$r(x) = q(x) \tag{264}$$

$$D_{\mathrm{KL}}[Q\|P] = \log_2\left(\frac{2}{\ell}\right) - \frac{1}{2}\log_2 e \tag{265}$$

$$D_\infty[Q\|P] = 1 - \log_2 \ell \tag{266}$$

$$D_\alpha[Q\|P] = -\log_2 \ell + \frac{\alpha - \log_2(1+\alpha)}{\alpha - 1} \tag{267}$$

$$w_P(h) = \ell - \frac{\ell^2}{2}\cdot h \tag{268}$$

$$w_Q(h) = 1 - \frac{\ell^2}{4}\cdot h^2 \tag{269}$$

$$\sigma(h) = \frac{2h}{2 - \ell \cdot h} \tag{270}$$

$$\sigma^{-1}(t) = \frac{2t}{2 + \ell \cdot t} \tag{271}$$

Additionally, in this case we find

$$\mathbb{P}[\tilde{T} \ge t] = \frac{4}{(2 + \ell \cdot t)^2} \tag{272}$$

$$\mathbb{P}[\tilde{T} \in dt] = \frac{8 \cdot \ell}{(2 + \ell \cdot t)^3} \tag{273}$$

$$\varphi_{\tilde{T}}(s) = \frac{2G_{1,3}^{3,1}\left(\frac{s^2}{l^2}\,\middle|\, \begin{matrix} 0 \\ 0, 1, 3/2 \end{matrix}\right)}{\sqrt{\pi}} \tag{274}$$

$$+ \frac{2is^2\,\mathrm{sign}(s)\left(l - |s|\left(2\mathrm{Ci}\left(\frac{2|s|}{l}\right)\sin\left(\frac{2|s|}{l}\right) + \left(\pi - 2\mathrm{Si}\left(\frac{2|s|}{l}\right)\right)\cos\left(\frac{2|s|}{l}\right)\right)\right)}{l^2|s|} \tag{275}$$

$$\mathcal{M}_{\tilde{T}}(s) = \frac{\ell^2 + 2\ell s - 4s^2 e^{-2s/\ell}\mathrm{Ei}\left(\frac{2s}{\ell}\right)}{\ell^2} \quad \text{for } s \le 0, \tag{276}$$

where $\varphi_{\tilde{T}}$ and $\mathcal{M}_{\tilde{T}}$ denote the characteristic and the moment-generating functions of $\tilde{T}$, respectively. Furthermore, Si, Ci and Ei are the sine, cosine and exponential integrals, defined as

$$\mathrm{Si}(x) = \int_0^x t^{-1}\sin(t)\,dt \tag{277}$$

$$\mathrm{Ci}(x) = -\int_{-x}^\infty t^{-1}\cos(t)\,dt \tag{278}$$

$$\mathrm{Ei}(x) = \int_{-\infty}^x t^{-1}e^t\,dt. \tag{279}$$

Finally, $G_{1,3}^{3,1}$ is the Meijer-G function (Sections 5.1-3; Bateman & Erdélyi, 1953) Interestingly, we can see that in this case, while the expectation of $\tilde{T}$ is finite, we find that $\mathbb{V}[\tilde{T}] = \infty$. Thus, by Equation (72), we find that $\mathbb{V}[N] = \infty$ as well, where $N$ is the number of samples simulated by Algorithm 3.

### G.3 Finite Discrete-Discrete / Piecewise Constant Case

Without loss of generality, let $Q$ be a discrete distribution over a finite set with $K$ elements defined by the probability vector $(r_1 < r_2 < \ldots < r_K)$ and $P = \mathrm{Unif}([1:K])$ the uniform distribution on $K$ elements. Note that any discrete distribution pair can be rearranged into this setup. Now, we can embed this distribution into $[0, 1]$ by mapping $P$ to $\hat{P} = \mathrm{Unif}(0, 1)$ the uniform distribution over $[0, 1]$ and mapping $Q$ to $\hat{Q}$ with density

$$\hat{q}(x) = \sum_{k=1}^{K} \mathbf{1}\left[\lfloor K \cdot x \rfloor = k - 1\right] \cdot K \cdot r_k. \tag{280}$$

Then, we can draw a sample $x$ from $Q$ by simulating a sample $\hat{x} \sim \hat{Q}$ and computing $x = K \cdot \lfloor \hat{x}/K \rfloor + 1$. Hence, we can instead reduce the problem to sampling from a piecewise constant distribution. Thus, let us now instead present the more general case of arbitrary piecewise constant distributions over $[0, 1]$, with $\tilde{Q}$ defined by probabilities $(q_1 < q_2 < \ldots < q_K)$ and corresponding piece widths $(w_1 < w_2 < \ldots < w_K)$. We require, that $\sum_{k=1} q_k = \sum_{k=1}^{K} w_k = 1$. Then, the density is

$$\tilde{q}(x) = \sum_{k=1}^{K} \mathbf{1}\left[\sum_{j=1}^{k-1} w_j \le x \le \sum_{j=1}^{k} w_j\right] \cdot \frac{q_k}{w_k} \tag{281}$$

Define $r_k = q_k/w_k$. Then,

$$w_P(h) = \sum_{k=1}^{K} \mathbf{1}[h \le r_k] \cdot w_k \tag{282}$$

$$w_Q(h) = \sum_{k=1}^{K} \mathbf{1}[h \le r_k] \cdot w_k \cdot r_k \tag{283}$$

$$\sigma(h) = \sum_{k=1}^{K} \mathbf{1}[h \ge r_{k-1}] \cdot \frac{1}{B_k} \cdot \ln\left(\frac{A_k - B_k r_{k-1}}{A_k - B_k \min\{r_k, h\}}\right) \tag{284}$$

where we defined

$$A_k = \sum_{j=k}^{K} w_j r_j = \sum_{j=k}^{K} q_j \tag{285}$$

$$B_k = \sum_{j=k}^{K} w_j \tag{286}$$

### G.4 Diagonal Gaussian-Gaussian Case

Without loss of generality, let $Q = \mathcal{N}(\mu, \sigma^2 I)$ and $P = \mathcal{N}(0, I)$ be $d$-dimensional Gaussian distributions with diagonal covariance. As a slight abuse of notation, let $\mathcal{N}(x \mid \mu, \sigma^2 I)$ denote the probability density function of a Gaussian random variable with mean $\mu$ and covariance $\sigma^2 I$ evaluated

at $x$. Then, when $\sigma^2 < 1$, we have

$$r(x) = Z \cdot \mathcal{N}\left(x \mid m, s^2 I\right) \tag{287}$$

$$m = \frac{\mu}{1 - \sigma^2} \tag{288}$$

$$s^2 = \frac{\sigma^2}{1 - \sigma^2} \tag{289}$$

$$Z = \frac{(1 - \sigma^2)^{-d}}{\mathcal{N}(\mu \mid 0, (1 - \sigma^2)I)} \tag{290}$$

$$w_P(h) = \mathbb{P}\left[\chi^2\left(d, \|m\|^2\right) \leq -2s^2 \ln h + C\right] \tag{291}$$

$$w_Q(h) = \mathbb{P}\left[\chi^2\left(d, \left\|\frac{m - \mu}{\sigma}\right\|^2\right) \leq \frac{-2s^2 \ln h + C}{\sigma^2}\right] \tag{292}$$

$$C = s^2\left(2 \ln Z - d \ln(2\pi s^2)\right). \tag{293}$$

Unfortunately, in this case, it is unlikely that we can solve for the stretch function analytically, so in our experiments, we solved for it numerically using Equation (4).

### G.5 One-dimensional Laplace-Laplace Case

Without loss of generality, let $Q = \mathcal{L}(\mu, b)$ and $P = \mathcal{L}(0, 1)$ be Laplace distributions. As a slight abuse of notation, let $\mathcal{L}(x \mid \mu, b)$ denote the probability density function of a Laplace random variable with mean $\mu$ and scale $b$ evaluated at $x$. Then, for $b < 1$ we have

$$r(x) = \frac{1}{b} \exp\left(-\frac{|x - \mu|}{b} + |x|\right) \tag{294}$$

$$w_P(h) = \begin{cases} 1 - \exp\left(\frac{b \ln(bh)}{1-b}\right) \cosh\left(\frac{\mu}{1-b}\right) & \text{if } \ln h \leq -\frac{|\mu|}{b} - \ln b \\ \exp\left(\frac{b^2 \ln(bh) - |\mu|}{1-b^2}\right) \sinh\left(-\frac{b(\ln(bh) - |\mu|)}{1-b^2}\right) \\ \quad = \frac{1}{2}\left((bh)^{\frac{-b}{1+b}} \exp\left(-\frac{|\mu|}{1+b}\right) - (bh)^{\frac{b}{1-b}} \exp\left(-\frac{|\mu|}{1-b}\right)\right) & \text{if } \ln h > -\frac{|\mu|}{b} - \ln b \end{cases} \tag{295}$$

$$w_Q(h) = \begin{cases} 1 - \exp\left(\frac{\ln(hb)}{1-b}\right) \cosh\left(\frac{\mu}{1-b}\right) & \text{if } \ln h \leq -\frac{|\mu|}{b} - \ln b \\ 1 - \exp\left(\frac{\ln(hb) - |\mu|}{1-b^2}\right) \cosh\left(\frac{b(\ln(hb) - |\mu|)}{1-b^2}\right) \\ \quad = 1 - \frac{1}{2}(bh)^{\frac{1}{1-b}} \exp\left(-\frac{|\mu|}{1-b}\right) - \frac{1}{2}(bh)^{\frac{1}{1+b}} \exp\left(-\frac{|\mu|}{1+b}\right) & \text{if } \ln h > -\frac{|\mu|}{b} - \ln b, \end{cases} \tag{296}$$

from which

$$(\sigma^{-1})' = \begin{cases} 1 - \sigma^{-1} + \underbrace{\left(b^{\frac{b}{1-b}} - b^{\frac{1}{1-b}}\right)}_{\geq 0 \text{ on } [0,1]} \cosh\left(\frac{\mu}{1-b}\right) (\sigma^{-1})^{\frac{1}{1-b}} & \text{if } \ln h \leq -\frac{|\mu|}{b} - \ln b \\ 1 + \frac{1}{2}\left(b^{\frac{b}{1-b}} - b^{\frac{1}{1-b}}\right) \exp\left(-\frac{|\mu|}{1-b}\right) (\sigma^{-1})^{\frac{1}{1-b}} \\ \quad - \frac{1}{2}\left(b^{-\frac{b}{1+b}} + b^{\frac{1}{1+b}}\right) \exp\left(-\frac{|\mu|}{1+b}\right) (\sigma^{-1})^{\frac{1}{1+b}} & \text{otherwise} \end{cases} \tag{297}$$

Similarly to the Gaussian case, we resorted to numerical integration using Equation (4) to solve for $\sigma^{-1}$.

## H Experimental Details

**Comparing Algorithm 3 versus Global-bound A\* coding:** We use a setup similar to the one used by Theis & Yosri (2022). Concretely, we assume the following model for correlated random variables $\mathbf{x}, \mu$:

$$P_\mu = \mathcal{N}(0, 1) \tag{298}$$

$$P_{\mathbf{x}|\mu} = \mathcal{N}(\mu, \sigma^2). \tag{299}$$

From this, we find that the marginal on $\mathbf{x}$ must be $P_{\mathbf{x}} = \mathcal{N}(0, \sigma^2 + 1)$. The mutual information is $I[\mathbf{x}; \mu] = \frac{1}{2} \log_2 \left(1 + \sigma^2\right)$ bits, which is what we plot as $I[\mathbf{x}; \mu]$ in the top two panels in Figure 2.

For the bottom panel in Figure 2, we fixed a standard Gaussian prior $P = \mathcal{N}(0, 1)$, fixed $K = D_{\mathrm{KL}}[Q\|P]$ and linearly increased $R = D_{\infty}[Q\|P]$. To find a target that achieves the desired values for these given divergences, we set its mean and variances as

$$\sigma^2 = \exp\left(W(A \cdot \exp(B))\right) - B \tag{300}$$

$$\mu = 2K - \sigma^2 + \ln \sigma^2 + 1 \tag{301}$$

$$A = 2\ln R - 2K - 1 \tag{302}$$

$$B = 2\ln R - 1, \tag{303}$$

where $W$ is the principal branch of the Lambert $W$-function (Corless et al., 1996).

We can derive this formula by assuming we wish to find $\mu$ and $\sigma^2$ such that for fixed numbers $K$ and $R$, and $q(x) = \mathcal{N}(x \mid \mu, \sigma^2), p(x) = \mathcal{N}(x \mid 0, 1)$. Then, we have that

$$D_{\mathrm{KL}}[q\|p] = K \quad \text{and} \quad \sup_{x \in \mathbb{R}} \left\{\frac{q(x)}{p(x)}\right\} = R. \tag{304}$$

We know that

$$K = D_{\mathrm{KL}}[q\|p] = \frac{1}{2} \left[\mu^2 + \sigma^2 - \log \sigma^2 - 1\right]$$

$$\log R = \log \sup_{x \in \mathbb{R}} \left\{\frac{q(x)}{p(x)}\right\} = \frac{\mu^2}{2(1 - \sigma^2)} - \log \sigma. \tag{305}$$

From these, we get that

$$\mu^2 = 2K - \sigma^2 + \log \sigma^2 + 1$$

$$\mu^2 = 2(1 - \sigma^2)(\log R + \log \sigma) \tag{306}$$

Setting these equal to each other

$$\begin{aligned}
2K - \sigma^2 + \log \sigma^2 + 1 &= 2\log R + \log \sigma^2 - 2\sigma^2 \log R - \sigma^2 \log \sigma^2 \\
\sigma^2 \log \sigma^2 - \sigma^2 + 2\sigma^2 \log R &= 2\log R - 2K - 1 \\
\sigma^2 \log \sigma^2 + \sigma^2 (2\log R - 1) &= A \\
\sigma^2 (\log \sigma^2 + B) &= A \\
\sigma^2 \log(\sigma^2 e^B) &= A \\
e^B \sigma^2 \log(\sigma^2 e^B) &= A e^B \\
e^{\log(\sigma^2 e^B)} \log(\sigma^2 e^B) &= A e^B \\
\log(\sigma^2 e^B) &= W(A e^B) \\
\sigma^2 &= e^{W(A e^B) - B},
\end{aligned} \tag{307}$$

where we made the substitutions $A = 2\log R - 2K - 1$ and $B = 2\log R - 1$.

# I Rejection sampling index entropy lower bound

Assume that we have a pair of correlated random variables $\mathbf{x}, \mathbf{y} \sim P_{\mathbf{x}, \mathbf{y}}$ and Alice and Bob wish to realize a channel simulation protocol using standard rejection sampling as given by, e.g. Algorithm 2. Thus, when Alice receives a source symbol $\mathbf{y} \sim P_{\mathbf{y}}$, she sets $Q = P_{\mathbf{x}|\mathbf{y}}$ as the target and $P = P_{\mathbf{x}}$ as the proposal for the rejection sampler. Let $N$ denote the index of Alice's accepted sample, which is also the number of samples she needs to draw before her algorithm terminates. Since each acceptance decision is an independent Bernoulli trial in standard rejection sampling, $N$ follows a geometric distribution whose mean equals the upper bound $M$ used for the density ratio (Maddison, 2016). The lower bound on the optimal coding cost for $N$ is given by its entropy

$$\mathbb{H}[N] = -(M - 1)\log_2\left(1 - \frac{1}{M}\right) + \log_2 M \geq \log_2 M, \tag{308}$$

where the inequality follows by omitting the first term, which is guaranteed to be positive since $x \mapsto -x \log_2 x$ is positive on $(0, 1)$. Hence, by using the optimal upper bound on the density ratio $M^* = \exp_2(D_\infty[Q\|P])$ and plugging it into the formula above, we find that

$$D_\infty[Q\|P] \le \mathbb{H}[N]. \tag{309}$$

Now, taking expectation over $\mathbf{y}$, we find

$$\mathbb{E}_{\mathbf{y} \sim P_\mathbf{y}} \left[ D_\infty[P_{\mathbf{x}|\mathbf{y}}\|P_\mathbf{x}] \right] \le \mathbb{H}[N]. \tag{310}$$

