# OpenReview forum: "Greedy Poisson Rejection Sampling"
_NeurIPS.cc/2023/Conference — NeurIPS 2023 poster_

### Official Review · Reviewer_tLU5 · 2023-06-24

**Soundness:** 4 excellent
**Presentation:** 3 good
**Contribution:** 4 excellent
**Rating:** 9
**Confidence:** 5

**Summary:**

The paper proposes a new relative entropy coding algorithm for compression without quantization. Compression without quantization is an exciting line of research that tries to eliminate training-test mismatches in learned compression by avoiding discrete representations altogether. Consequently, one can losslessly compress data in a continuous latent variable model, such as a VAE, without quantizing the latent representation. The downside of existing algorithms for relative entropy coding is that their runtimes are intolerably slow. The paper makes a big step towards more runtime efficiency by designing new efficient rejection sampling approaches for transmitting samples from a variational posterior, especially in cases where the likelihood ratio between prior and posterior is uni-modal.

**Strengths:**

+ Relative entropy coding is a somewhat under-appreciated but up-and-coming field of research at the intersection between information theory and machine learning. There are only a few papers in this domain. (For example, recent advances made compression in diffusion models possible [Theis et al., 2022].)
+ The paper significantly improves runtime efficiency for these algorithms, paving the way toward its scalable deployment. Compared to runtime complexities exponential in the Renyi divergence, it achieves a runtime *linear* in the KL divergence between the target and proposal distributions. Notably, the involved algorithm is conceptually much simpler (and in practice faster) than relative entropy coding with A* coding, which achieves the same asymptotic scaling.
+ Beyond its base version, the paper also proposes two computationally much more efficient extensions: parallel GPRS and branch&bound GPRS. Both approaches rely on non-trivial mathematical insights, e.g., additivity of a Poisson process.
+ Moreover, in one-dimensional channel simulation problems where the density ratio is unimodal, the algorithm displays the theoretically-optimal runtime (unlike existing approaches such as A* sampling)
+ The paper shows a nice combination of solid empirical results and theory. For example, most claims on the runtime complexities and theoretical guarantees (finite runtime) are proven rigorously.

**Weaknesses:**

See questions below. There are no immediate weaknesses evident, but some clarifying questions need to be addressed.

**Questions:**

* Could the authors discuss in depth the commonalities and differences to a recent paper with a similar title, “adaptive greedy rejection sampling” (https://arxiv.org/pdf/2304.10407.pdf) that the authors already mentioned?
* The presentation at the beginning of Section 3 is very dense. For example, for an arbitrary P and Q, how is w_p approximated—analytically or based on samples? How does that affect the numerical solvability of Eq. 2?
* It appears that the stretching function sigma is learned anew for every datum. How does this aspect affect the runtime, and is it respected in the scaling analysis? A discussion would be adequate.
* Parallelization and branch & bound: it appears that the parallelization affects the cost to *encode* data only, but not to *decode* data. Provided this is correct, could the authors explain why the encoding step is a bigger concern than decoding? In practice (e.g., video streaming), one mostly cares about reducing *decoding* complexity.
* Could the authors also comment on how they propose to simulate from the Poisson process restricted to subregions? It is unclear why this saves in runtime performance since sampling from a spatially-restricted Poisson process again amounts to rejection sampling.

Writing suggestion:
* It took me a long time to understand why a temporal poisson process construction was used, where the original communication problem is time-independent. Maybe just say early on that including a temporal poisson process replaces rejection sampling with a deterministic criterion, a trick that was previously proposed for numerical integration (Maddison).

**Limitations:**

Limitations in terms of scaling and speed are sufficiently addressed. Societal impacts are a less important topic for general-purpose data transmission protocols.

---

> ### Author Rebuttal · Authors · 2023-08-09
>
> We thank the reviewer for their glowing review of our work; we are delighted that the reviewer shares our excitement for relative entropy coding/channel simulation!
>
> We answer the reviewer's questions below and will gladly answer any further questions.
> > Could the authors discuss in depth the commonalities and differences to a recent paper with a similar title, "adaptive greedy rejection sampling"?
>
> Of course! The "obvious" commonality between GPRS and adaptive greedy rejection sampling (AGRS) is that they are both rejection samplers for channel simulation, i.e. they encode exact samples with optimal expected codelength, but their runtime is stochastic. This contrasts with the depth-limited version of A* sampling, an importance sampler whose runtime is fixed but only returns approximate samples. Moreover, GPRS and AGRS are greedy algorithms, though they differ in _how_ they are greedy. As we explain in lines 207-210, we call GPRS greedy because it performs a greedy search over the points of a Poisson process instead of the non-greedy search performed by A* sampling/coding. On the other hand, AGRS is greedy in the sense defined by Harsha et al. (2009): it greedily maximizes the acceptance probability of a proposed sample under a certain correctness constraint; see Section IV in Harsha et al. (2009) for the precise explanation. These two notions of greediness are likely related, and we are actively investigating this link.
>
> Furthermore, Flamich & Theis (2023) also propose a space partitioning method that empirically appears to achieve an exponential speedup similar to GPRS's branch-and-bound variant. However, the authors' space partition differs significantly from ours in several ways: The authors' method is based on dithered quantization (DQ; Agustsson & Theis, 2020) and hybrid coding (Theis & Yosri, 2022), while we based GPRS's branch-and-bound variant on a bisection search method, inspired by the AS* and AD* variants of A* coding. Moreover, our method comes with theoretical guarantees on the correctness and runtime of the algorithm, while Flamich & Theis do not even prove the correctness of their space partitioning variant, let alone provide guarantees on the runtime; all their results are empirical. That said, it is interesting whether their DQ-based space partitioning method could be adapted to be used by GPRS too.
>
> > ... for an arbitrary P and Q, how is w_p approximated—analytically or based on samples? How does that affect the numerical solvability of Eq. 2?
>
> This is a good question; unfortunately, we lack a good answer for the general case. Appendix G provides analytic solutions for $w_P$ and $w_Q$ for the most relevant practical cases, i.e., the uniform, Gaussian and Laplace distributions. Hence, numerical integration is fine in these cases. Evaluating $w_P$ and $w_Q$ even approximately and studying its effects on the numerical integration in more general cases is an interesting future direction for research. As a first step, we are currently investigating if there is a natural family of distributions (e.g. natural exponential families) for which $w_P$ and $w_Q$ can always be analytically computed.
>
> In the camera-ready version, we will add a clarifying discussion on this and note the challenges the reviewer highlighted as interesting future research directions.
> > It appears that the stretching function sigma is learned anew for every datum. How does this aspect affect the runtime, and is it respected in the scaling analysis?
>
> We are not quite sure what the reviewer means by "datum". If they mean target distribution, then this is correct; we need to perform the numerical integration during each encoding procedure; unfortunately, there doesn't seem to be a good way to precompute $\sigma$. In our runtime analysis, we assumed that we could evaluate $\sigma$ in $\mathcal{O}(1)$ time, and we used the number of samples simulated by the algorithm as the proxy for the algorithm's runtime, as it is the dominant factor. However, reducing the complexity of the numerical integration is an important practical concern, and we will note this as future work in the camera-ready version.
> > Parallelization and branch & bound: it appears that the parallelization affects the cost to encode data only, but not to decode data. Provided this is correct, could the authors explain why the encoding step is a bigger concern than decoding?
>
> In practice, we use the step count as the random seed (mixed with a global seed) for the proposed samples. Hence, the decoder only needs to set the step count they received as their PRNG seed and simulate a single sample from the proposal distribution. In the camera-ready version, we will add a section to the appendix discussing the basic implementation details for GPRS.
> > Could the authors also comment on how they propose to simulate from the Poisson process restricted to subregions? It is unclear why this saves in runtime performance since sampling from a spatially-restricted Poisson process again amounts to rejection sampling.
>
> Great question! The reviewer is correct in the general case; simulating from arbitrarily truncated distributions is computationally hard. However, we only truncate to intervals in the one-dimensional branch-and-bound variant of GPRS. We can use generalised inverse transform sampling in this case: Suppose we have a real-valued random variable $X$ with CDF $F$ and inverse CDF $F^{-1}$. Then, we can simulate $X$ truncated to $[a, b]$ by simulating $U \sim \mathrm{Uniform}(F(a), F(b))$ and computing $X = F^{-1}(U)$.
>
> ## References
>
> - Agustsson, E., & Theis, L. (2020). Universally quantized neural compression. In NeurIPS 2020
>
> - Flamich, G., & Theis, L. (2023). Adaptive Greedy Rejection Sampling. arXiv preprint
>
> - Harsha, P., Jain, R., McAllester, D., & Radhakrishnan, J. (2009). The Communication Complexity of Correlation. IEEE Transactions on Information Theory
>
> - Theis, L., & Ahmed, N. Y. (2022). Algorithms for the communication of samples. In ICML

---

> > ### Comment · Reviewer_tLU5 · 2023-08-16
> > **Thanks for your response**
> >
> > I have read the authors' response. My rating remains high. I hope the authors use my feedback not only to respond to me as a reviewer but also adopt changes to improve the paper's clarity in terms of practical aspects of their algorithm, e.g., when it comes to the stretching function and calculating w_p. Overall, I congratulate the authors on their nice work.

---

### Official Review · Reviewer_prqL · 2023-06-28

**Soundness:** 3 good
**Presentation:** 2 fair
**Contribution:** 2 fair
**Rating:** 5
**Confidence:** 2

**Summary:**

The paper addresses the problem of representing a target distribution using the least possible number of bits. The authors refined the idea of encoding a sample from the target distribution as the first sample from the proposal distribution that passes a rejection sampling condition. The refined approach is shown to achieve runtime optimality.

**Strengths:**

The idea of using a Poisson process to encode a target distribution is powerful. The strategy is well known but further investigations and attempts to make it more feasible could have an impact in various domains.

**Weaknesses:**

The authors' technical contribution. could have been outlined more precisely. It is hard to understand why the new strategy is expected to be better than the A* algorithm or naive rejection sampling. The authors should have explained why "a criterion that does depend on the time variable" is better.

In the introduction, the authors say that the distribution is assumed to be unimodal and 1 dimensional. This is a strong assumption and it is unclear why the proposed method requires it. Similar assumptions are made in Flemich 22. The authors may have specified whether these are also required in Maddison 16.

The authors should have found a way of testing the algorithm on real-world data.

The text could be more curated and structured.



**Questions:**

- What are the requirements the target and proposal distributions should fulfil? Why do the distributions need to be 1-dimensional? Is this only for theoretical purposes (i.e. to estimate the runtime bounds)? Or would it be also a practical constraint?
- How expansive is it to solve the ODE for computing \sigma? Should this be included in the total runtime?
- How is the density ratio estimated? Is the method feasible if the target distribution is unavailable?
- What are the key differences between the proposed algorithm, A*, and the approach described in Maddison 16? Why is rejection sampling supposed to be better than importance sampling? It would be good to include an intuitive explanation of why and when GPRS is faster.


**Limitations:**

The authors do not discuss the limitation of the proposed approach. It is unclear what assumptions are required for the proofs. The theoretical runtime of related methods is not reported.

---

> ### Author Rebuttal · Authors · 2023-08-07
>
> We thank the reviewer for their comments and feedback on our work. We would like to begin by clarifying a crucial point that, in our experience, is the most common source of confusion regarding channel simulation.
> > The paper addresses the problem of representing a target distribution using the least possible number of bits.
>
> This is not quite correct; in channel simulation, our goal is **not** to encode the target distribution so that the decoder can recover it. Please see lines 25-27 and 136-139 in our paper that outline this. As the reviewer notes in the second sentence of their summary, we only seek to encode a sample from the target distribution.
>
> We now address the reviewer's concerns and questions.
> > The authors' technical contribution. could have been outlined more precisely.
>
> We outline our technical and empirical contributions on lines 56-67, visually depict all GPRS variants in Figure 1, sketch GPRS's construction in Section 3, and construct and analyse it rigorously in the Appendices. Did the reviewer find these too vague or mathematically imprecise or simply that the writing needed to be improved? We are happy to incorporate any concrete feedback the reviewer has!
> > It is hard to understand why the new strategy is expected to be better than the A* algorithm or naive rejection sampling.
>
> This is a tricky question that would take more space to answer than we have available for this rebuttal; if the reviewer is interested, we are happy to follow up with more detail!
>
> Note that the expected runtimes (in terms of the number of samples simulated) of GPRS and naive rejection sampling (NRS) are all equal to $2^{D_\infty[Q || P]}$ and the only thing that differs is the codelength. Here, GPRS performs better than NRS because by only accepting samples below the graph of $\phi$ and rejecting everything above, GPRS concentrates the acceptance probability of samples in the earliest step possible. We see this best in the discrete case, where we might draw the same sample $X_k = X_{k+1}$ from the proposal twice in a row. In this case, GPRS will always accept $X_k$, whereas NRS might reject $X_k$ but accept $X_{k+1}$.
>
> At a high level, GPRS's branch-and-bound variant is better than A* coding's because it immediately terminates once it finds the sample it will accept. On the other hand, when A* sampling finds the sample it will eventually accept, it does some "extra useless work" by checking more samples due to its non-greedy acceptance criterion.
>
> In the camera-ready version, we will add an explanation of why GPRS outperforms NRS and A* coding based on our answers above.
>
> > The authors should have found a way of testing the algorithm on real-world data.
>
> We developed a rejection sampler; what kind of test using real-world data does the reviewer have in mind?
> > What are the requirements the target and proposal distributions should fulfil? Why do the distributions need to be 1-dimensional? Is this only for theoretical purposes (i.e. to estimate the runtime bounds)? Or would it be also a practical constraint?
>
> As we mentioned in lines 82-84, we can define the standard version of GPRS over very general probability spaces. We only require that $dQ/dP$ be bounded, meaning that GPRS is theoretically as widely applicable as NRS. On the other hand, the branch-and-bound version requires 1-dimensional distributions over $\mathbb{R}$ because it can be considered a stochastic bisection search, which is naturally defined over $\mathbb{R}$ and extending it to more general spaces is highly non-trivial.
> > How expansive is it to solve the ODE for computing \sigma? Should this be included in the total runtime?
>
> Our analysis in the paper assumes that we can evaluate $\sigma$ in $\mathcal{O}(1)$ time; we will clarify this in the camera-ready version. Of course, in practice, we need to ensure that we can cheaply do the numerical integration, but we leave this for future work.
> > How is the density ratio estimated? Is the method feasible if the target distribution is unavailable?
>
> GPRS wouldn't work very well as a general-purpose sampling algorithm in its current form. In this paper, we are only interested in applying it to channel simulation, where it is reasonable to assume that the encoder knows the density ratio exactly.
> > What are the key differences between the proposed algorithm, A*, and the approach described in Maddison 16?
>
> The approach proposed in Maddison 16 _is_ A* sampling, and channel simulation is not discussed. For a comparison of GPRS and A* _coding_, please see lines 207-214 and 291-300 in our paper.
> > Why is rejection sampling supposed to be better than importance sampling?
>
> The answer to this question depends on what problem we are applying the sampler to and what the reviewer means by "better". Rejection sampling yields an exact sample from the desired target distribution, but its termination time is stochastic. In contrast, importance sampling has a fixed runtime but only produces an approximate sample.
> > It would be good to include an intuitive explanation of why and when GPRS is faster.
>
> See our response above for a brief discussion regarding the speed comparison of the algorithms.
> > The authors do not discuss the limitation of the proposed approach. It is unclear what assumptions are required for the proofs.
>
> As noted in the paper, we intentionally left a lot of discussion in the main text informal and only sketched the constructions of the different variants of GPRS in Section 3 to illustrate the important, high-level ideas. In each section, we refer the reader to the appropriate parts of the appendix for the fully rigorous development of each result. Could the reviewer elaborate on what they found lacking and needing improvement, please?
>
> _We will gladly answer any further questions from the reviewer. On the other hand, should we have answered the reviewer's concerns adequately, we kindly ask the reviewer to consider raising their score._

---

> > ### Comment · Reviewer_prqL · 2023-08-17
> > **Thank you for your answers**
> >
> > After reading the authors' answers and the other reviewers' comments, I am ready to support acceptance. I have raised my score accordingly. I still have concerns about the practical application of the scheme but recognise this is a theoretical work. Real data evaluation is probably not needed.

---

### Official Review · Reviewer_4NF6 · 2023-07-03

**Soundness:** 4 excellent
**Presentation:** 3 good
**Contribution:** 3 good
**Rating:** 6
**Confidence:** 1

**Summary:**

This paper focuses on the channel simulation problem, which finds applications in stochastic lossy compression. In contrast to the importance sampling approach A* coding (Flamich et.al. 2022) for channel simulation with Poisson processes, this work adopts a rejection sampling method. They demonstrate that the standard sampling approach may yield suboptimal code lengths. As an alternative, the authors propose a novel greedy rejection sampling algorithm called the Greedy Rejection Sampling (GPRS) algorithm. This algorithm composites the density ratio $r = \frac{dQ}{dP}$ with an invertible function referred to as the "stretch" function $\sigma$ that utilizes the temporal structure of the Poisson process without changing the target distribution.  Theoretical findings establish the correctness and optimality of the proposed GPRS algorithm in terms of code length. Furthermore, the expected runtime of GPRS matches the runtime of standard rejection sampling.

In addition to the vanilla GPRS, the authors introduce parallel and Branch-and-bound variants to further enhance the runtime of the algorithm. Specifically, the Branch-and-bound GPRS demonstrates a provable runtime improvement over A* coding when the density ratio exhibits unimodal characteristics. To validate the theoretical results, the authors conduct several toy experiments.






**Strengths:**

- Mathematical derivations look rigorous.
- This paper is well-written and easy to follow.
- Theoretical results seem to be compelling. The authors successfully demonstrate, through rigorous analysis, that the GPRS algorithm outperforms both standard rejection sampling and A* coding in terms of either runtime or code length in some cases.

**Weaknesses:**

- Some baselines and closely related works are not compared theoretically or empirically (See the first two bullets in Questions).

**Questions:**

- GPRS is thoroughly compared to A* coding both theoretically and experimentally. However, it is unclear why the baseline standard rejection sampling (Algorithm 2) is not empirically compared in Section 4 to validate the theoretical assertion that GPRS enhances the code length efficiency compared to standard RS.
- Additionally, while the more recent works dithered quantization methods (DQ) and greedy rejection coding (GRC) are briefly mentioned in the related works section, there is a lack of theoretical or empirical comparisons with GPRS. Although these methods may have distinct formulations and proof techniques, conducting some comparisons would be valuable in justifying the significance of GPRS and its potential advantages over alternative approaches.
- GPRS needs to numerically solve an ODE for $\sigma^{-1}$ before the time steps. What about the extra cost in runtime?

---

> ### Author Rebuttal · Authors · 2023-08-05
>
> We thank the reviewer for their nice comments; we address the reviewer's questions below.
>
> > GPRS is thoroughly compared to A* coding both theoretically and experimentally. However, it is unclear why the baseline standard rejection sampling (Algorithm 2) is not empirically compared in Section 4 to validate the theoretical assertion that GPRS enhances the code length efficiency compared to standard RS.
>
> We omitted standard RS from the codelength comparison because its codelength is provably suboptimal (as we outline in Problem 1 in Section 2.2 and show in Appendix I), and hence including it in Figure 2 would be uninformative. On the other hand, Figure 2 does more than verify our theorems regarding the expected runtime of the different algorithms; it also demonstrates that our upper bound on GPRS's codelength appears quite tight, and empirically it tends to heavily concentrate around its mean and that the mean appears to be quite robust in that it closely lines up with the median.
>
> In the camera-ready version of the paper, we will add clarifying sentences based on our response above in the experiments section (Section 4).
>
> > Additionally, while the more recent works dithered quantization methods (DQ) and greedy rejection coding (GRC) are briefly mentioned in the related works section, there is a lack of theoretical or empirical comparisons with GPRS. Although these methods may have distinct formulations and proof techniques, conducting some comparisons would be valuable in justifying the significance of GPRS and its potential advantages over alternative approaches.
>
> **Regarding GRC:** We agree with the reviewer that in the current version of the paper, the comparison between GRC and GPRS in terms of high-level theoretical properties and empirical performance is too loose. We decided to omit GRC from Figure 2 because we felt it made the figure too cluttered; however, in the camera-ready version, we will add another section in the appendix with more empirical comparisons and specifically comparisons to different variants of GRC there. Furthermore, we will extend the discussion of GRC in the related works section and compare their theoretical properties more precisely by contrasting the exact bounds on their runtime, for example.
>
> **Regarding DQ:** Unfortunately, we are uncertain how to compare DQ with GPRS best as its formulation is significantly different. In fact, there isn't currently a complexity-theoretic framework that would cover all channel simulation protocols. We are actively working on this as part of our research agenda. In our paper, since we developed a rejection sampler, the number of samples simulated by the algorithm is a natural proxy for the computational time complexity that lends itself to theoretical analysis. We assume all other aspects (such as simulating a single sample from the proposal or evaluating sigma) take $\mathcal{O}(1)$ time. However, this proxy is inappropriate for DQ, as it is not a rejection or importance sampler and only needs to simulate one sample from the proposal. Furthermore, DQ only applies to cases where both the target and proposal distributions are uniform, and in this case, it enjoys a clear advantage over GPRS.
>
> In the camera-ready version, we will clarify that we are using the number of samples simulated by GPRS as a proxy for the algorithms' time complexity and assume that all other operations can be carried out in constant time. Based on our answer above, we will also discuss the difficulties of comparing DQ with GPRS.
>
> > GPRS needs to numerically solve an ODE for $\sigma^{-1}$ before the time steps. What about the extra cost in runtime?
>
> This is an important practical question, and currently, we lack a good answer to it. As discussed above, in our runtime analysis, we assumed that evaluating $\sigma$ or its inverse could be done in constant time and disregarded practical implementation details as it was not a hindrance to carrying out our experiments (we used Scipy's `odeint` function to solve the ODE which was sufficiently fast in practice). That said, developing a cheap approximation to $\sigma$ or its inverse is an important future direction, and we are actively working on it.
>
> Thus, in the camera-ready version, we will discuss the practical considerations for the numerical integration and point out that analyzing the complexity of evaluating $\sigma$ and developing cheap-to-evaluate approximations is an exciting future research direction.

---

> > ### Comment · Reviewer_4NF6 · 2023-08-12
> >
> > Thank you for the response. Upon reading both the main paper and the authors' rebuttal, I am inclined to support its acceptance. However, I would like to acknowledge that my evaluation is limited because this paper is out of my expertise.

---

### Official Review · Reviewer_HQuL · 2023-07-07

**Soundness:** 3 good
**Presentation:** 4 excellent
**Contribution:** 3 good
**Rating:** 6
**Confidence:** 3

**Summary:**

This paper investigates the problem of one-shot channel simulation, which can be used as lossy compression without involving quantization.
A new rejection sampling algorithm called greedy Poisson rejection sampling (GPRS) is proposed. Then,  a parallelized and a branch-and-bound variant is proposed. Those algorithms are analyzed regrading both correctness and runtime.
Toy experiments on one-dimensional problems show that GPRS compares
favourably against A* coding, the current state-of-the-art channel simulation protocol.

**Strengths:**

The proposed GPRS is properly positioned against previous and concurrent works, and contains sufficient novelty.
The proposed method has much better runtime complexity compared with previous SOTA A* coding.
The manuscript is well written and enough background information is provided for general readers.

**Weaknesses:**

Though this manuscript is a theoretical contribution, it would be better to discuss more about promising application scenarios and current gaps regarding both performance and efficiency, which can encourage more future works in this field and facilitate more applications of similar techniques.

**Questions:**

L331-334, please explain in more detail about the application scenario of efficient channel simulation protocol for multivariate Gaussians.

**Limitations:**

see above.

---

> ### Author Rebuttal · Authors · 2023-08-04
>
> We thank the reviewer for their valuable comments and the concerns they raise, to which we respond below.
>
> > Though this manuscript is a theoretical contribution, it would be better to discuss more about promising application scenarios and current gaps regarding both performance and efficiency, which can encourage more future works in this field and facilitate more applications of similar techniques.
>
> We agree with the reviewer that we need to discuss promising applications for computationally efficient channel simulation/relative entropy coding to put our work into a better context. Indeed, channel simulation already has a few exciting non-trivial applications, such as model compression (Havasi et al., 2018), lossy data compression with perfect realism (Blau & Micali, 2018; Theis et al., 2022), and differential privacy (Shah et al., 2020). Our work in this area is particularly relevant as the practicality of all these works is limited precisely by the undesirable exponential scaling of the runtime of the channel simulation algorithms they use.
>
> Hence, in the camera-ready version of the paper, we will add a paragraph or two explaining and discussing this to help the reader put our work into its appropriate context. Furthermore, we will add more interesting possible future directions in the paper's final section to encourage future works, as the reviewer suggested.
>
> > L331-334, please explain in more detail about the application scenario of efficient channel simulation protocol for multivariate Gaussians.
>
> Of course! In fact, the example we give below is the primary motivation for our work; for a more complete discussion, please refer to (Flamich et al. 2020; 2022) and Theis et al. (2022).
>
> The relevance of multivariate Gaussians is their widespread use in machine learning, in particular for variational inference and diffusion models. Thus, let us assume we wish to develop a neural image compression algorithm using a variational autoencoder (VAE) and channel simulation. We choose a standard Gaussian prior and a mean-field Gaussian variational posterior for our VAE. Note that with a powerful-enough network architecture, this is not a very restrictive choice for the distributions, and the VAE will be able to adapt to the distribution of images very well. Now, for simplicity, we can train this VAE with the standard beta-ELBO on an image dataset to fit the network parameters, or we can also incorporate an adversarial loss term if we are interested in compression with realism constraints.
>
> Once we fit the model, we can compress a new image as follows, assuming that besides sharing common randomness and the VAE's prior $p$, the decoder also knows the VAE's generative network. The encoder receives a new image $x$ and uses the VAE's inference network to obtain the image's latent variational posterior $q(z | x)$. Then, the encoder uses a channel simulation protocol to encode a single multivariate Gaussian sample $z \sim q(z | x)$ using the VAE's prior $p$ as the proposal distribution. The decoder obtains a lossy reconstruction of the image $x$ by decoding the encoder's sample and pushing it through the VAE's generative network.
>
> The practicality of this scheme hinges upon the efficiency of the channel simulation protocol. Previous works have all used inefficient, exponentially scaling encoding algorithms and had to resort to small-scale experiments only or develop tricks to make the encoding procedure feasible. In contrast, we could break up encoding a single multivariate sample into a sequence of one-dimensional samples and apply branch-and-bound GPRS dimensionwise. This would be fast but incur at least 1 bit of overhead per dimension, so the codelength would be somewhat suboptimal. Thus, solving the multivariate Gaussian problem would yield a maximally efficient solution that does not have the dimensionwise codelength overhead.
>
> ## References
> - Blau, Y., & Michaeli, T. (2018). The perception-distortion tradeoff. In Proceedings of the IEEE conference on computer vision and pattern recognition
> - Flamich, G., Havasi, M., & Hernández-Lobato, J. M. (2020). Compressing images by encoding their latent representations with relative entropy coding. In NeurIPS 2020
> - Flamich, G., Markou, S., & Hernández-Lobato, J. M. (2022). Fast relative entropy coding with A* coding. In ICML 2022
> - Havasi, M., Peharz, R., & Hernández-Lobato, J. M. (2018). Minimal random code learning: Getting bits back from compressed model parameters. In ICLR 2018
> - Shah, A., Chen, W. N., Balle, J., Kairouz, P., & Theis, L. (2022). Optimal compression of locally differentially private mechanisms. In AISTATS 2022
> - Theis, L., Salimans, T., Hoffman, M. D., & Mentzer, F. (2022). Lossy compression with Gaussian diffusion.

---

> > ### Comment · Reviewer_HQuL · 2023-08-17
> >
> > I have read all the discussion and decide to keep my score.

---

### Official Review · Reviewer_sb37 · 2023-07-12

**Soundness:** 3 good
**Presentation:** 3 good
**Contribution:** 2 fair
**Rating:** 6
**Confidence:** 3

**Summary:**

This paper proposes a new algorithm for lossy compression, using ideas from Poisson rejection sampling.

Given a sample $y \sim P_y$, Alice wants to communicate the smallest number of bits possible such that Bob can simulate $x \sim P_{x | y}$, when Alice and Bob have access to the distribution $P_{x, y}$ (and shared randomness to allow simulation).

The proposed algorithm achieves an almost optimal codelength by rejection sampling -- by introducing a temporal random variable t, Alice can communicate the hitting time $t$ for which $(t, X_t)$ lies under an appropriately defined function $\phi$. Variations of this algorithm are suggested: one variation is to parallelize using $L$ servers to reduce the runtime by a factor of $L$, and another binary search variant which gives an exponential improvement when the Radon-Nikodym derivative of $dP_{x|y} / dP_{x}$ is bounded.

**Strengths:**

- The paper has several nice ideas from information theory, and seems to be the first that can provably achieve runtimes predicted by Li and El Gamal.

- The algorithm is built on A* coding, but is more robust, in the sense that while the runtime of A* coding depends on the Renyi entropy between $P_{x|y}$ and $P{x}$, the runtime of the proposed algorithm is proportional to the KL divergence between them, which may be much smaller and finite even when the Renyi entropy is infinite.


**Weaknesses:**

- The rejection sampler requires access to likelihoods of the conditional distribution and marginal, and hence it's not clear how much this can be generalized.

- The inverse function in Eqn (2) is solved numerically. How would this be computed for distributions with harder CDFs?

- All experiments are toy experiments on Gaussian, Binomial variables etc.

- The experiments are for scalar random variables, and extending to higher dimensions would be extremely non-trivial. The authors state that extending this to multivariate Gaussians is an open problem.

- The paper tries to establish connections with areas like generative modelling by arguing that source compression is useful in VAEs/ latent diffusion models, but this is probably not correct. The works of Theis et al for example use generative models for compression, not the other way around.

Minor:
- The name GPRS for a communication protocol is probably a poor choice given that it's an existing standard.
- $\mathbb{V}$ in Theorem 3.1 is undefined.



**Questions:**

Questions are listed in the weaknesses section.

My main concern is that this is probably a poor fit for NeurIPS. The theorems are nice, but the impact to a NeurIPS audience is probably extremely limited, given that the algorithms are only feasible for explicit likelihood models and scalar random variables.

**Limitations:**

There needs to be more discussion about the limitations of this work, especially if the authors are going to mention connections between the proposed algorithm and compression of neural networks. The connections to generative modelling are extremely weak.

---

> ### Author Rebuttal · Authors · 2023-08-04
>
> We thank the reviewer for their nice comments and valuable feedback and attempt to address the concerns they raise below.
>
> > The rejection sampler requires access to likelihoods of the conditional distribution and marginal, and hence it's not clear how much this can be generalized.
>
> We believe the reviewer means that in our definition of GPRS, we require exact knowledge of $dQ/dP$ (i.e. an unnormalized version will not suffice). Furthermore, we need to be able to compute the $w_P$ and $w_Q$ quantities just to numerically evaluate $\sigma^{-1}$, which makes applying GPRS difficult to more complex problems. These concerns are completely fair, and we agree that GPRS, in its current form, would be a terrible general-purpose sampling algorithm! However, we are interested in applying GPRS to solve channel simulation, which we would use as a core component in a neural data compression pipeline; where the most frequently used channel distributions are uniform, Gaussian and Laplace distributions.
>
> We will improve our presentation in the camera-ready version in two ways based on the reviewer's comment:
> 1. We will more clearly delineate between the relevance of using GPRS as a general-purpose rejection sampler (for which it is terrible) and for channel simulation (for which it yields theoretically optimal and SOTA results)
> 2. It is an exciting question if we could GPRS extended to give a general-purpose sampling algorithm by lifting the requirements on the exact knowledge of $dQ/dP$ and being able to compute $w_P$ and $w_Q$ analytically.
>
> > The inverse function in Eqn (2) is solved numerically. How would this be computed for distributions with harder CDFs?
>
> We believe here the author is asking about cases where even if we know the derivative exactly and can compute $w_P$ and $w_Q$ analytically, evaluating them might still be computationally expensive. This is a very good question; sadly, we lack a good general answer. Thankfully, this is not an issue in the practically relevant cases (i.e. uniform, Gaussian, Laplace); these quantities are all cheap to evaluate (see Appendix G for the precise analytic formulae).
>
> Thus, based on the reviewer's comment, we will discuss the concerns mentioned above regarding the practicality of the numerical integration in the camera-ready version of the paper.
>
> > The experiments are for scalar random variables, and extending to higher dimensions would be extremely non-trivial. The authors state that extending this to multivariate Gaussians is an open problem.
>
> This is correct, though we'd like to specify that standard GPRS can also be applied to multivariate Gaussians. The open problem is concerned with finding a fast algorithm with $\mathcal{O}(D_{KL}[Q || P])$ runtime for multivariate Gaussians.
>
> > The paper tries to establish connections with areas like generative modelling by arguing that source compression is useful in VAEs/ latent diffusion models, but this is probably not correct. The works of Theis et al for example use generative models for compression, not the other way around.
>
> We think this might be a misunderstanding; our work is fully in line with the work of Theis et al. (2022); in particular, we could apply "standard" GPRS to obtain a similar diffusion-model-based lossy compression algorithm. Could the reviewer please elaborate on which of our arguments they are referring to and believe to be incorrect?
>
> > The name GPRS for a communication protocol is probably a poor choice given that it's an existing standard.
>
> We thank the reviewer for pointing this out! Fortunately, we believe that the two concepts are sufficiently far apart (one being a rejection sampler, the other being a cellular communication standard) that the chances of much confusion arising should be minimal.
>
> > $\mathbb{V}$ in Theorem 3.1 is undefined.
>
> We thank the reviewer for noting this, $\mathbb{V}$ simply denotes the variance of a random variable in analogy to the expectation operator $\mathbb{E}$. We will clarify this in the manuscript.
>
> > My main concern is that this is probably a poor fit for NeurIPS. The theorems are nice, but the impact to a NeurIPS audience is probably extremely limited, given that the algorithms are only feasible for explicit likelihood models and scalar random variables.
>
> We appreciate the concern of the reviewer. However, we argue that our work is, in fact, relevant to the learned data compression community at NeurIPS, as evidenced by many related papers at NeurIPS, ICML, and ICLR, such as Havasi et al. (2018), Flamich et al. (2020), Agustsson & Theis (2020) and Theis et al. (2022). All of these works use channel simulation/relative entropy coding. Their primary limitation is the exponential scaling of the runtime of their algorithms, which could be improved by using a faster algorithm. Thus, our paper presents a significant step towards making all the abovementioned methods more practical.
>
> However, we must communicate this point more clearly in the paper. Hence, in the camera-ready version, we will include a paragraph or two explaining and discussing this to help the reader put our work into its appropriate context.
>
> ## References
>
>  - Agustsson, E., & Theis, L. (2020). Universally quantized neural compression. In NeurIPS 2020
>  - Flamich, G., Havasi, M., & Hernández-Lobato, J. M. (2020). Compressing images by encoding their latent representations with relative entropy coding. In NeurIPS 2020
>  - Havasi, M., Peharz, R., & Hernández-Lobato, J. M. (2018). Minimal random code learning: Getting bits back from compressed model parameters. In ICLR 2018.
>  - Theis, L. & Yosri, N. (2022) Algorithms for the communication of samples. In ICML 2022.
>  - Theis, L., Salimans, T., Hoffman, M. D., & Mentzer, F. (2022). Lossy compression with Gaussian diffusion.

---

### Author Rebuttal · Authors · 2023-08-09

We thank all the reviewers for their valuable feedback on our paper, which will help us improve it significantly. We are delighted that all reviewers agree that our contributions are significant, that most reviewers (sb37, HQuL, 4NF6 and tLU5) found our exposition well-written and easy to follow, and that Reviewers prqL and tLU5 highlighted the potential impact of our work.

However, the reviewers also identified several aspects of the paper that need improvement. Thus, we highlight the most common concerns and the most significant changes we propose for the camera-ready version of our paper.

## Delineate GPRS as a sampling algorithm from its use in channel simulation

We presented GPRS in its full generality and did not comment much on its limitations as a general sampling algorithm, detached from its intended use for channel simulation. Hence, Reviewers sb37, prqL and tLU5 have rightly raised concerns regarding cases that often arise in practice when applying a sampling algorithm, e.g. that the Radon-Nikodym derivative is only known up to a constant or that the functions $w_P$ and $w_Q$ could be troublesome to evaluate.

This question is especially interesting because A* coding, based on the A* sampling algorithm, is more generally applicable in practice just as a sampling algorithm.

On the other hand, since we are ultimately motivated by applying our GPRS to neural compression via channel simulation, the set of distributions of practical interest is much smaller, consisting of the uniform, Gaussian and Laplace distributions. This is because, in neural compression, the data distribution (e.g. the distribution of natural images) is usually approximated by a latent representation following a simple distribution (e.g. Gaussian) that is then transformed by powerful neural networks to match the distribution statistics (e.g. in the case of VAEs and diffusion models).

Hence, in the camera-ready version of the paper, we will add clarifying discussion regarding the limitations of applying GPRS as a general-purpose sampling algorithm and note that extending it to more general cases is future work.

## Applications of GPRS
Reviewers sb37, HQuL and prqL expressed concern that the motivation or application of our method (and perhaps channel simulation in general) needs to be communicated more clearly.

Developing a fast channel simulation algorithm is of great practical interest, as there have been several papers at NeurIPS, ICML, and ICLR, such as Havasi et al. (2018), Flamich et al. (2020), Agustsson & Theis (2020) and Theis et al. (2022) utilizing it. All of these works use channel simulation/relative entropy coding. Their primary limitation is the exponential scaling of the runtime of their algorithms, which could be improved by using a faster algorithm. Thus, our paper presents a significant step towards making all the abovementioned methods more practical.

Hence, in the camera-ready version, we will include a paragraph or two explaining and discussing this to help the reader put our work into its appropriate context.

## Concerns regarding the implementation and numerics
Related to the first point, Reviewers sb37,  4NF6, prqL and tLU5 expressed concerns regarding the implementation details and numerics of GPRS, such as:
- how $w_P$ and $w_Q$ can be computed or approximated
- the numerical stability and time complexity of evaluating $\sigma$ or $\sigma^{-1}$
- the intricacies of the encoding and decoding procedures

While there are many subtle details, here we would like to emphasize that:
 - for the practically relevant channel simulation cases $w_P$ and $w_Q$ have analytic forms which are stated in Appendix G
 - for the runtime complexity analysis we assumed that $\sigma$ or its inverse can be evaluated in $\mathcal{O}(1)$ time
 - the time complexity of the numerical integration is essentially independent of the number of samples simulated, and we found it to be negligible in our experiments, as the number of samples GPRS simulates heavily dominates the time complexity anyways

To address all the above and more, in the camera-ready version of the paper, we will add another section to the Appendix to detail all relevant considerations for the practical implementation of each GPRS variant.

__We thank the reviewers again for their time reviewing our work and providing important feedback to improve our work. We will gladly address any further questions the reviewers might have during the discussion phase. Should we have answered a reviewer's concerns adequately, we kindly invite them to consider raising their score.__

## References
- Agustsson, E., & Theis, L. (2020). Universally quantized neural compression. In NeurIPS 2020
- Flamich, G., Havasi, M., & Hernández-Lobato, J. M. (2020). Compressing images by encoding their latent representations with relative entropy coding. In NeurIPS 2020
- Havasi, M., Peharz, R., & Hernández-Lobato, J. M. (2018). Minimal random code learning: Getting bits back from compressed model parameters. In ICLR 2018.
- Theis, L. & Yosri, N. (2022) Algorithms for the communication of samples. In ICML 2022.
- Theis, L., Salimans, T., Hoffman, M. D., & Mentzer, F. (2022). Lossy compression with Gaussian diffusion.

---

### Decision · Program_Chairs · 2023-09-21

**Decision:**

Accept (poster)

**Comment:**

The authors solve the problem of 1-shot channel simulation (a fundamental data compression problem) in the 1-dimensional case where the target-proposal density ratio is unimodal, by giving a provable algorithm based on Poisson processes with optimal runtime. Empirical results show improvement over SOTA algorithms.
All reviewers recommend acceptance after discussion. I recommend acceptance.